# Unifying Low Dimensional Spectra in Deep Learning

**Connall Garrod** [1]    **Jonathan P. Keating** [1]

## Abstract

Low-dimensional structures appear ubiquitously in the eigenspectra of deep-learning matrices in classification networks trained in the overparameterized regime. While theoretical advances have aimed to explain this phenomenology, they typically succeed only in capturing subsets of the full behavior or rely on assumptions that cannot hold in practice. In this work, we provide an analytic explanation for the bulk–outlier structure of several canonical deep-learning matrices, including the Hessian, gradients, and weights. We achieve this using unconstrained feature models (UFMs), a now-common tool for studying the emergence of deep neural collapse (DNC). We show that DNC is the source of these low-dimensional eigenspectra: in each case, the eigenvalues and eigenvectors can be constructed from feature means, the characterizing objects of DNC. This provides a unifying analytic explanation for a wide range of spectral phenomena in deep learning and goes beyond empirical characterizations—which typically focus on eigenvalues—by providing a detailed analysis of eigenvectors. We prove that our results hold for both linear and ReLU networks and provide numerical validation in both the modeling context and standard deep-network architectures on canonical datasets.

## 1. Introduction

Deep classification networks demonstrate low-dimensional spectra when trained past the interpolation threshold. Evidence began with analyses of Hessians (LeCun et al., 2012; Dauphin et al., 2014; Sagun et al., 2016; 2017; Papyan, 2020), where eigenvalue histograms revealed that most eigenvalues cluster in a "bulk" near zero, with a few separate outliers. Notably, the number of outliers often matches the

number of classes, $K$ (Sagun et al., 2017). Later, Papyan (2020) identified an additional "mini-bulk" consisting of $K(K-1)$ outliers, yielding a total of $K^2$ outliers. Similar bulk–outlier patterns have been reported for the covariance of gradients (Jastrzebski et al., 2020), the Fisher information matrix (Li et al., 2020), the spectrum of backpropagated errors (Oymak et al., 2019), weight matrices (Mahoney & Martin, 2019), and layer-wise Hessians (Sankar et al., 2021). It has also been noted that gradients tend to align with the Hessian's top-$K$ outlier eigenspace (Gur-Ari et al., 2018).

Papyan et al. (2020) identified further structure in trained classifiers—neural collapse (NC)—in which penultimate-layer features concentrate around their class means, and these means, together with the final-layer weights, form simplex equiangular tight frames (ETFs). Later work showed that this phenomenon extends beyond the final layer, leading to the notion of deep neural collapse (DNC) (He & Su, 2022; Rangamani et al., 2023; Parker et al., 2023).

Despite the Hessian's central role in generalization (Wu et al., 2017; Li et al., 2020; Foret et al., 2020), optimization (Gur-Ari et al., 2018; Cosson et al., 2022; Li et al., 2022), and robustness (Yao et al., 2018; Moosavi-Dezfooli et al., 2018; Zhao et al., 2020), it remains notoriously difficult to analyze theoretically. Many works attempt to reproduce its empirical properties (Choromanska et al., 2015; Pennington & Worah, 2018; Granziol, 2020; Baskerville et al., 2021; Liao & Mahoney, 2021), but they capture only subsets of the observed behavior or rely on unrealistic assumptions. Consequently, despite the importance of the Hessian in deep learning theory, the reason it exhibits the specific structure documented by Papyan (2020) remains open.

Additionally, these low-dimensional structures have not yet been fully unified. A few isolated results exist—for example, Ben Arous et al. (2024) links low dimensionality in the gradients, Hessian, and Fisher information matrix, but only for a two-layer network. The connection to DNC has also been largely overlooked: the only explicit mention is a remark about the link between NC and Hessian spectra in Papyan et al. (2020), without mathematical justification or acknowledgment of the importance of DNC. This gap matters because DNC is more intuitive and geometrically interpretable than the Hessian. Establishing that DNC shapes Hessian behavior would provide a new lens on curvature

---

[1] Mathematical Institute, University of Oxford. Correspondence to: Connall Garrod <connall.garrod@maths.ox.ac.uk>.

*Proceedings of the 43$^{rd}$ International Conference on Machine Learning*, Seoul, South Korea. PMLR 306, 2026. Copyright 2026 by the author(s).

and a stronger basis for understanding, diagnosing, and improving neural network training in practice.

## 1.1. Contributions

We provide the first mathematical explanation that simultaneously recovers and unifies all empirically observed details of the bulk–outlier structure across a variety of deep-learning matrices. We achieve this using the deep unconstrained feature model (UFM), a popular and well-supported tool in the study of overparameterized classification networks (see App. A). Our main contributions are as follows.

**Low-Rank Spectra Emerge in Layer-wise Matrices.** We prove that the layer-wise Hessians and gradients of the deep UFM exhibit the same spectral structure observed in the full objects (Gur-Ari et al., 2018; Papyan, 2020). This provides the first analytic model to reproduce all low-dimensional spectral phenomena simultaneously. We also go beyond prior empirical observations by revealing previously unreported structure in the corresponding eigenvectors.

**DNC Causes Low-Dimensional Spectra.** We show that DNC is the source of this low-dimensional structure: the eigenvalues and eigenvectors can be expressed entirely in terms of the feature means. Thus, DNC provides a unified mathematical explanation for many seemingly disparate empirical observations in deep-matrix spectra.

**Global Matrices Inherit Their Structure.** Under minimal assumptions, we further prove that the full Hessian inherits the same eigenspectrum as the layer-wise Hessians. This elevates the impact of DNC from layer-wise objects near the end of the network to the full-network structure, and consequently highlights its relevance to key notions in deep learning such as local flatness.

Our analysis applies to both linear and ReLU layers, and we validate the theory empirically on standard deep classification architectures. Taken together, these results support our central claim: DNC provides a unifying explanation for a wide variety of low-dimensional spectral phenomena observed in deep learning.

Our work clarifies open questions about how flatness and DNC interconnect, and it opens the door to more systematic design choices that leverage the importance of flatness for deployment-relevant properties by shaping DNC. What's more, recent work has argued that the effect of flatness on generalization can be measured using only the final-layer Hessian (Han et al., 2025; Walter et al., 2025), and that layer-wise Hessians can serve as a diagnostic tool (Bolshim & Kugaevskikh, 2025). Our work fully characterizes these objects in the overparameterized limit.

## 1.2. Related Works

Here we review the most closely related works. See App. A for a full review of the UFM, additional references, and a detailed justification for its use as a modeling tool. The UFM was originally introduced by Mixon et al. (2020) to explore NC, and has since been generalized to the deep UFM to analyze DNC. This line of work has focused primarily on mean squared error (MSE) loss, including linear networks (Dang et al., 2023), binary classification (Súkeník et al., 2023), two separate layers (Tirer & Bruna, 2022), and more general settings (Súkeník et al., 2024). These works provide meaningful insight into DNC, but they do not consider deep-learning matrices more broadly, including the practically important Hessian. Hessian spectra have been analyzed using random matrix theory (Pennington & Bahri, 2017; Pennington & Worah, 2018; Liao & Mahoney, 2021; Baskerville et al., 2022; Granziol et al., 2022), spin-glass analogies (Dauphin et al., 2014; Choromanska et al., 2015; Baskerville et al., 2021), neural tangent kernel limits (Fan & Wang, 2020; Jacot et al., 2020), and decoupling conjectures (Wu et al., 2020). Some of these works leverage properties of the feature and weight matrices in their Hessian analyses, but they model only subsets of the full behavior and do not incorporate NC/DNC. Overall, our results are significantly more general: *(i)* they establish the causal role of DNC in the emergence of low-rank spectra in the Hessian; *(ii)* they capture all schematic details observed in the Hessian; and *(iii)* they provide analytic expressions for both eigenvalues and eigenvectors. Gradient alignment with outlier eigenspaces has been investigated by Gur-Ari et al. (2018), Ben Arous et al. (2024), and Song et al. (2024), which we connect to DNC in Section 3.3.

## 2. Background

We consider $K$-class classification with $n$ samples per class. We denote the $i^{\text{th}}$ data point in the $c^{\text{th}}$ class by $x_{ic} \in \mathbb{R}^{d_0}$, with corresponding one-hot encoded labels $y_c \in \mathbb{R}^K$. A deep neural network $f(x; \theta) : \mathbb{R}^{d_0} \to \mathbb{R}^K$, parameterized by $\theta \in \mathbb{R}^p$, is used to fit the data

$$f(x; \theta) = W_L \sigma(W_{L-1} \sigma(...\sigma(W_1 h(x; \bar{\theta}))...)), \quad (1)$$

where $\theta = \{W_L, ..., W_1, \bar{\theta}\}$, with weight matrices $W_L \in \mathbb{R}^{K \times d}, W_1, ..., W_{L-1} \in \mathbb{R}^{d \times d}$. The function $h(x; \bar{\theta}) : \mathbb{R}^{d_0} \to \mathbb{R}^d$ is a highly expressive feature map, and $\sigma : \mathbb{R} \to \mathbb{R}$ is an activation function. The feature vectors at each layer are defined as $h_{ic} = h(x_{ic}; \bar{\theta})$, $h_{ic}^{(2)} = W_1 h_{ic}$, and $h_{ic}^{(l)} = W_{l-1} \sigma(h_{ic}^{(l-1)})$, for $l = 3, ..., L$. We define the matrices $H_l = [h_{11}^{(l)}, h_{21}^{(l)}, ..., h_{n1}^{(l)}, h_{12}^{(l)}, ..., h_{nK}^{(l)}] \in \mathbb{R}^{d \times Kn}$, whose columns are ordered by class. We also define the class feature means as $\mu_c^{(l)} = \text{Av}_i\{h_{ic}^{(l)}\}$, the global feature means as $\mu_G^{(l)} = \text{Av}_c\{\mu_c^{(l)}\}$ and the feature mean matrices as $\bar{H}_l = [\mu_1^{(l)}, \mu_2^{(l)}, ..., \mu_K^{(l)}] \in \mathbb{R}^{d \times K}$. Av denotes taking an

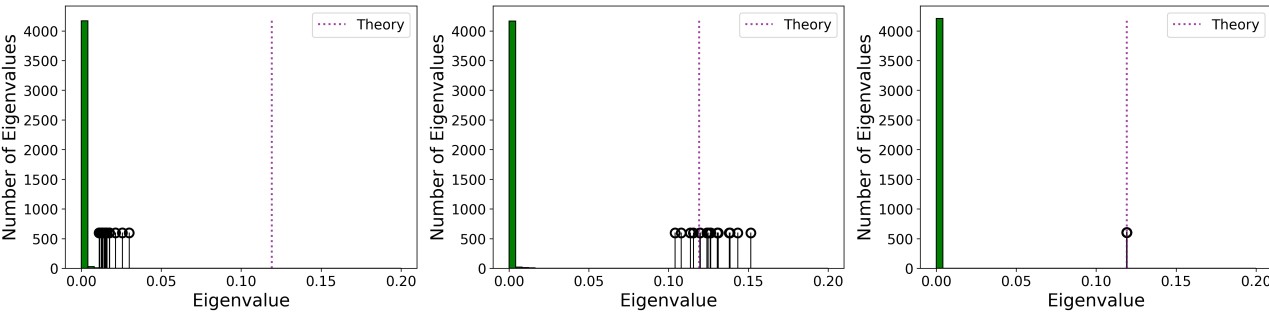

*Figure 1.* Histograms of the spectrum of Hess$_l$ for a deep linear UFM at an intermediate layer $l = 3$, shown at epochs: 0, 2,500 and 50,000, for parameters $L = 4, K = 4$. The top $K^2 = 16$ outlier eigenvalues are plotted as spikes.

average. We denote normalized vectors by $\hat{v} = v/\|v\|_2$. $1_n$ is the $n$-dimensional all-ones vector and $\otimes$ is the Kronecker product.

In the context of MSE loss, DNC refers to the following phenomena observed in overparameterized DNNs as training progresses (Súkeník et al., 2023).

**Definition 2.1** (Deep Neural Collapse). A layer $l$ has DNC structure if the following conditions hold

(i) **DNC1:** Feature vectors collapse to their class means $H_l = \bar{H}_l \otimes 1_n^T$.
(ii) **DNC2:** The class mean matrix forms an orthogonal frame $\bar{H}_l^T \bar{H}_l \propto I_K$.
(iii) **DNC3:** The rows of the weight matrix $W_l$ are linear combinations of the columns of $\bar{H}_l$.

**Deep Learning Spectra:** The Hessian is defined as $\mathrm{Hess}(\theta) = \nabla_\theta \nabla_\theta^T \mathcal{L}$. Papyan (2020) showed that the Hessian can be decomposed into the sum of the Fisher information matrix $G$ and a residual matrix $E$. He further showed that $E$ does not contribute to the outliers, and that $G$ exhibits cross-class structure, meaning it can be expressed as

$$G = \sum_{i=1}^{n} \sum_{c,c'=1}^{K} w_{icc'} g_{icc'} g_{icc'}^T, \qquad (2)$$

where $w_{icc'} \geq 0$ are scalars and $g_{icc'} \in \mathbb{R}^p$ are extended gradients. Papyan then further decomposed this object

$$G = G_{\text{class}} + G_{\text{cross}} + G_{\text{within}}, \qquad (3)$$

where $G_{\text{class}}$, $G_{\text{cross}}$ and $G_{\text{within}}$ represent the covariances with respect to the $c, c'$, and $i$ indices, respectively. By subtracting these components from the Hessian, Papyan observed that the eigenvalues separate into $K$ large outliers due to $G_{\text{class}}$, $K(K-1)$ smaller ones that form a so-called "mini-bulk" due to $G_{\text{cross}}$, and a bulk at zero due to $G_{\text{within}}$. Theoretical explanations for, and recovery of, these highly structured spectral components remain open.

**Gradient Descent Alignment:** Define the gradient of the loss as $g(\theta) = \nabla_\theta \mathcal{L}$. We can project $g(\theta)$ onto the top $K$

eigenspace of $\mathrm{Hess}(\theta)$, producing the vector $g_{\text{top}}$. To quantify alignment with this subspace, we define the projection proportion $f_{\text{top}} = \|g_{\text{top}}\|_2^2/\|g\|_2^2$. Gur-Ari et al. (2018) observed that this proportion rapidly approaches one during training, indicating that the gradient becomes increasingly aligned with the top eigenspace.

**The Deep Unconstrained Feature Model:** To define the deep UFM, we approximate the expressiveness of the feature map $h(x; \bar{\theta})$ by treating the feature vectors $h_{ic}$ as freely optimized variables. Using MSE loss, the objective becomes

$$\mathcal{L} = \frac{1}{2Kn}\|Z - Y\|_F^2 + \frac{1}{2}\lambda \sum_{l=1}^{L} \|W_l\|_F^2 + \frac{1}{2}\lambda\|H_1\|_F^2, \quad (4)$$

where $Z = W_L \sigma(W_{L-1}\sigma(...W_2\sigma(W_1 H_1)...))$ is the network output, and $\mathcal{L}$ is optimized over $W_L, ..., W_1$ and $H_1$. Here $Y = I_K \otimes 1_n^T$ is the label matrix and $\lambda > 0$ is a regularization coefficient.

Dang et al. (2023) showed that the global optimizers of the deep UFM with linear activations all must satisfy DNC (see Lem. C.1) when $d \geq K$ and $\lambda$ lies below a threshold $\lambda_0$ (given in Eq. (17)). Súkeník et al. (2024) found that DNC is not globally optimal in the ReLU case. Nevertheless, they observed that DNC often emerges under typical hyperparameter regimes. This is consistent with empirical evidence that real neural networks frequently exhibit both NC and DNC (Papyan et al., 2020; Rangamani et al., 2023). A full justification for the use of the UFM as a model of overparameterized networks appears in App. A.

## 3. Low Dimensional Spectra in the Deep Linear UFM

Here we characterize how the low-dimensional structures described in Section 2 emerge in the Hessian, gradients, and weights of the deep UFM with linear activations. Our analysis proceeds in two steps. First, we prove the emergence of these structures at the layer-wise level, highlighting how DNC serves as a unifying source: the feature means determine all eigenvalues and eigenvectors. We then show that,

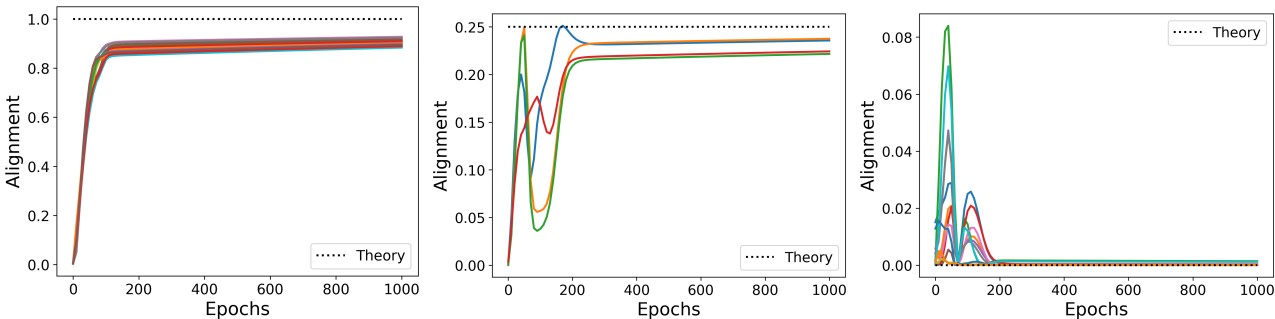

*Figure 2.* Training of a deep linear UFM with $L = 4$, $K = 4$ at an intermediate layer $l = 3$. In all cases we show early dynamics from low alignment to high alignment, and report final convergence values after 50,000 epochs in this caption. **Left**: Predicted eigenvector alignment, measured by squared cosine similarity between $\mu_c^{(l+1)} \otimes \mu_{c'}^{(l)}$ and $\text{Hess}_l(\mu_c^{(l+1)} \otimes \mu_{c'}^{(l)})$. All alignment values matched theory up to eight significant figures at convergence. **Middle & Right**: Gradient alignment, measured by the decomposition coefficients of $\tilde{g}^{(l)}$ in the basis of predicted eigenvectors $\mu_c^{(l+1)} \otimes \mu_{c'}^{(l)}$. Middle: $c = c'$, right: $c \neq c'$. All gradient alignment values matched theory up to five significant figures at convergence.

in the overparameterized limit, the full Hessian inherits this same structure, demonstrating that DNC ultimately characterizes deep-learning matrices at a global level. In Section 4, we extend these results to ReLU activations. Results for the gradient covariance and the backpropagated error matrix are deferred to App. E.

### 3.1. Layer-wise Hessian Spectra

The layer-wise Hessian is $\text{Hess}_l = \nabla_{w_l} \nabla_{w_l}^T \mathcal{L}$, where $w_l \in \mathbb{R}^{d^2}$ has entries $(w_l)_{d(x-1)+y} = (W_l)_{xy}$. In keeping with Papyan (2020), we omit the regularization terms from this definition. In App. C.1, we show that the layer-wise Hessian has the following Kronecker structure

$$\text{Hess}_l = A_{l+1}^T A_{l+1} \otimes \left[ \text{Av}_{ic} \left\{ h_{ic}^{(l)} h_{ic}^{(l)T} \right\} \right],$$

where $A_{l+1} = W_L \ldots W_{l+1}$. Using this structure, along with the properties of DNC, we obtain the following result.

**Theorem 3.1.** *Consider the deep linear UFM described in Eq. (4). Let the network width satisfy $d \geq K$, and consider a layer $l$ with $1 \leq l < L$. Assume that the regularization parameter $\lambda$ satisfies the condition in Eq. (17). Then, at any global optimum of the loss, the layer-wise Hessian at layer $l$ has the following eigen-decomposition*

$$\text{Hess}_l = \frac{1}{K} n^{\frac{-1}{L+1}} C^{\frac{2L}{L+1}} \sum_{c,c'=1}^K \nu_{cc'}^{(l)} \nu_{cc'}^{(l)T},$$

*where $\nu_{cc'}^{(l)} = \hat{\mu}_c^{(l+1)} \otimes \hat{\mu}_{c'}^{(l)}$. Hence, $\text{Hess}_l$ has rank $K^2$, with nonzero eigenvectors $\hat{\mu}_c^{(l+1)} \otimes \hat{\mu}_{c'}^{(l)}$, for $c, c' \in \{1, \ldots, K\}$. Moreover, all nonzero eigenvalues are equal to $\frac{1}{K} n^{\frac{-1}{L+1}} C^{\frac{2L}{L+1}}$. The scalar $C$ is the larger solution to the following equation, and determines the feature mean norms*

$$1 = C + K\lambda n^{\frac{1}{L+1}} C^{-\frac{L-1}{L+1}}, \quad \|\mu^{(l)}\|_2 = \frac{1}{\sqrt{n}} (C\sqrt{n})^{\frac{l}{L+1}}.$$

The proof appears in App. C.1. This result recovers the Hessian eigenvalue structure reported by Papyan (2020), except that here all nonzero eigenvalues are equal. In practice, DNNs do not reach the overparameterized and over-training limits, and the resulting noise perturbs the Hessian spectrum. The absence of a mini-bulk here arises from the use of MSE loss rather than cross-entropy (CE); we include the CE case in App. B to demonstrate this. This shows that many works that recover general numbers of outliers (Granziol, 2020; Liao & Mahoney, 2021), or that do not recover the separate components (Pennington & Worah, 2018; Baskerville et al., 2022), provide an incomplete description. Figure 1 provides empirical confirmation: the $K^2$ outliers emerge from the bulk, and converge to a common value in agreement with Theorem 3.1.

Crucially, the eigenvectors are built from the feature means, and thus DNC fully characterizes the Hessian. This explains why the class number $K$ consistently appears in empirical observations of Hessian spectra. Mapping data to maximally separated class means necessarily induces a low-dimensional eigenspectrum in the Hessian. The left panel of Figure 2 illustrates this phenomenon: while the feature-mean constructions are not eigenvectors at initialization, they rapidly become so in the early stages of training.

### 3.2. Layer-wise Hessian Cross-Class Structure

We next examine Papyan's decomposition, beginning with the Gauss-Newton decomposition, which splits $\text{Hess}_l$ into the sum of the layer-wise Fisher information matrix $G_l$ and a residual matrix $E_l$. In fact, for piecewise-linear activation functions, we show in App. C.2 that $E_l = 0$, so it does not contribute to the spectral outliers and $\text{Hess}_l = G_l$. Because Papyan considered the CE loss while we work with MSE loss, the expression for $G_l$ differs slightly from Eq. (2).

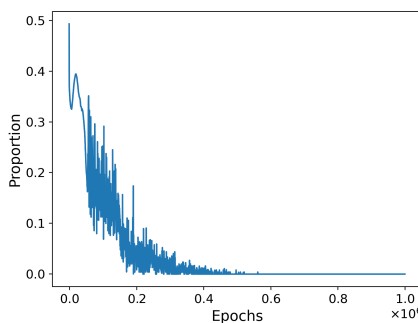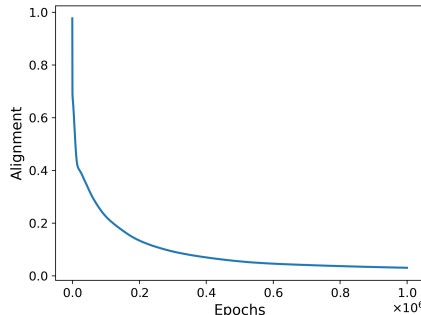

*Figure 3.* Training at layer $l = 3$ of a deep ReLU UFM with parameters $K = 4, L = 4$. **Left**: Proportion of feature vector entries below $-10^{-6}$. **Right**: Frobenius distance between $\bar{H}_l^T \bar{H}_l$ and $I$ after normalization, measuring the deviation of $\bar{H}_l$ from an orthogonal frame.

Nevertheless, it retains cross-class structure

$$G_l = \sum_{i=1}^{n} \sum_{c,c'=1}^{K} \frac{1}{Kn} v_{icc'} v_{icc'}^T,$$

where $v_{icc'} = a_{c'}^{(l+1)} \otimes h_{ic}^{(l)}$, and $(a_{c'}^{(l+1)})_x = (A_{l+1})_{c'x}$. The full derivation is given in App. C.2. As in Papyan (2020), we then decompose $G_l$ as

$$G_l = G_{l,\text{class}} + G_{l,\text{cross}} + G_{l,\text{within}}. \tag{5}$$

Exact expressions for these terms are given in App. C.2, along with a proof of the following theorem.

**Theorem 3.2.** *Consider the deep linear UFM, under the same assumptions as Theorem 3.1. Then, at any global optimum of the loss, the components of the decomposition in Eq. (5) satisfy*

*(i) $G_{l,within} = 0$.*
*(ii) $G_{l,cross}$ has rank $K(K-1)$. One choice of spanning eigenvectors for its nonzero eigenspace is $\left( \mu_1^{(l+1)} - \mu_{c'}^{(l+1)} \right) \otimes \mu_c^{(l)}$, for $c' \in \{2,\ldots,K\}$, $c \in \{1,\ldots,K\}$.*
*(iii) $G_{l,class}$ has rank $K$, with eigenvectors for its nonzero eigenspace given by $\mu_G^{(l+1)} \otimes \mu_c^{(l)}$, for $c \in \{1,\ldots,K\}$.*

*All nonzero eigenvalues are equal to the value reported in Theorem 3.1. Furthermore, the nonzero eigenspaces of $G_{l,class}$ and $G_{l,cross}$ are orthogonal.*

Thus, the decomposition experiments of Papyan (2020) are recoverable in this model: the two spectral components have the correct eigenvalue counts. Moreover, since their eigenspaces are orthogonal, each component of the decomposition cleanly explains a separate feature in the bulk/minibulk/spikes separation. As before, the structure is driven by the class feature means, highlighting that DNC is the source of this fine-grained structure. We provide numerical evidence of Theorem 3.2 in App. G.

### 3.3. Gradient Alignment with Outlier Eigenspace

We now compare the layer-wise gradients to the eigenvectors of $\text{Hess}_l$. In App. C.3, we show that

$$g^{(l)} = \frac{\partial \mathcal{L}}{\partial w^{(l)}} = \lambda w^{(l)} + \underbrace{\text{Av}_{ic}\{(A_{l+1}^T u_{ic}) \otimes h_{ic}^{(l)}\}}_{\tilde{g}^{(l)}}, \tag{6}$$

where $u_{ic} = W_L...W_1 h_{ic} - y_c$. To determine whether the gradient lies in the top eigenspace, we examine the term $\tilde{g}^{(l)}$. The following theorem describes $\tilde{g}^{(l)}$ at a global optimum.

**Theorem 3.3.** *Consider the deep linear UFM described in Eq. (4) under the same assumptions as Theorem 3.1. At any global optimum, the quantity $\tilde{g}^{(l)}$, defined in Eq. (6), is*

$$\tilde{g}^{(l)} = \frac{1}{K}(\sqrt{n})^{\frac{-1}{L+1}} C^{\frac{L}{L+1}} (C-1) \sum_{c=1}^{K} \left( \hat{\mu}_c^{(l+1)} \otimes \hat{\mu}_c^{(l)} \right),$$

*where $C$ is the constant defined in Theorem 3.1. Hence, $\tilde{g}^{(l)}$ has exactly $K$ equal nonzero coefficients when expanded in the natural basis given in Theorem 3.1.*

In the natural feature-mean basis, the gradient occupies only a $K$-dimensional subspace of the Hessian's nonzero eigenspace, despite the Hessian having $K^2$ nonzero eigenvectors. This matches the empirical observations of Gur-Ari et al. (2018) while remaining consistent with the results of Papyan (2020) and clarifying the role of DNC. Previous theoretical works have failed to capture such fine-grained details of these canonical objects. The middle and right panels of Figure 2 confirm the result: when decomposing $\tilde{g}^{(l)}$ in the basis $\mu_c^{(l+1)} \otimes \mu_{c'}^{(l)}$, the $c = c'$ coefficients approach $1/K$, and all others vanish.

### 3.4. Weight Spectra

The next theorem recovers the low-dimensional spectra results for the weight matrices reported by Papyan et al. (2020), and shows that the weights are entirely determined by the feature means. A proof is provided in App. C.4.

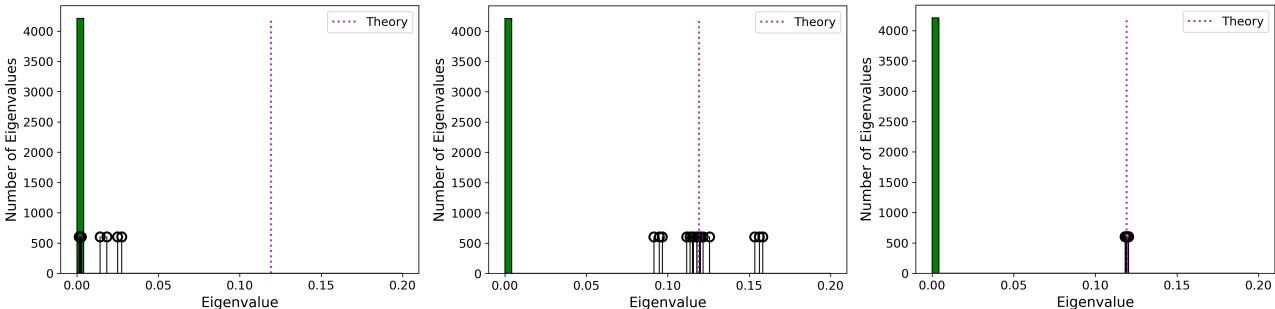

*Figure 4.* Histograms of the spectrum of $\text{Hess}_l$ for a deep ReLU UFM at an intermediate layer $l = 3$, using the same hyperparameters and setup as Figure 5, shown at epochs $0$, $4 \times 10^5$ and $3 \times 10^6$. The top $K^2 = 16$ outlier eigenvalues are plotted as spikes.

**Theorem 3.4.** *Consider the deep linear UFM described in Eq. (4), under the same assumptions as Theorem 3.1. Then, at a global optimum, the weight matrices $W_l$ are*

$$W_l = \frac{1}{K\lambda}(\sqrt{n})^{\frac{-1}{L+1}} C^{\frac{L}{L+1}}(1-C)\sum_{c=1}^{K}\hat{\mu}_c^{(l+1)}\hat{\mu}_c^{(l)\top}.$$

*Hence, the rank of $W_l$ is $K$, with all nonzero singular values equal to $\frac{1}{K\lambda}(\sqrt{n})^{\frac{-1}{L+1}} C^{\frac{L}{L+1}}(1-C)$. The corresponding left and right singular vectors are given by $\{\hat{\mu}_c^{(l+1)}\}_{c=1}^{K}$ and $\{\hat{\mu}_c^{(l)}\}_{c=1}^{K}$, respectively.*

### 3.5. Deep Neural Collapse Induces The Full Hessian Spectrum

We now show that the full Hessian exhibits structure similar to that of the layer-wise Hessians. Returning to the full DNN in Eq. (1), suppose that the feature map $h(x; \bar{\theta})$ consists of $\bar{L}$ layers, and that the deep UFM provides a suitable approximation. The entire network then has $\bar{L} + L$ layers, and its Hessian can be expressed in block form

$$\text{Hess}(\theta) = \begin{bmatrix} \text{Hess}_{\bar{L},\bar{L}} & \text{Hess}_{\bar{L},L} \\ \text{Hess}_{\bar{L},L}^{T} & \text{Hess}_{L,L} \end{bmatrix},$$

where the blocks are separated by the parameters in the first $\bar{L}$ layers and last $L$ layers. We begin with the bottom-right block, which can be analyzed directly in the UFM setting.

**Theorem 3.5.** *Consider the deep linear UFM described in Eq. (4), under the same assumptions as Theorem 3.1. At any global optimum, the Hessian with respect to the weight layers $W_1, ..., W_L$, denoted $\text{Hess}_{L,L}$, can be written as the sum of two terms, one of which vanishes in the limit $\lambda \to 0$. The leading-order term has rank $K^2$, with all nonzero eigenvalues equal to $\frac{1}{K}n^{\frac{-1}{L+1}} C^{\frac{2L}{L+1}} L$.*

The proof appears in App. C.5. This result shows that, for small regularization, the bottom-right block of the Hessian exhibits the same spectral structure as the layer-wise Hessians. The next theorem extends this to the full Hessian.

**Theorem 3.6.** *Consider the network described in Eq. (1), where the feature map $h(x; \bar{\theta})$ has $\bar{L}$ layers, and assume this feature map is expressive enough for the UFM modeling assumption to hold. Let $\hat{\lambda}_i(M)$ denote the $i^{th}$ largest eigenvalue of the matrix $M/\|M\|_F$. Further assume that all block Hessians are of the same asymptotic scale. Then, in the limit $L \to \infty$, with regularization strength decaying linearly in $L$, we have at global optimum*

$$\hat{\lambda}_i(\text{Hess}(\theta)) - \hat{\lambda}_i(\text{Hess}_{L,L}) \to 0, \quad \text{for } i = 1, ..., K^2,$$

$$\hat{\lambda}_i(\text{Hess}(\theta)) \to 0, \quad \text{for } i > K^2.$$

The proof is in App. C.6. Under our modeling assumptions, the theorem shows that when the network sufficiently over-parameterizes the data, the full Hessian at a DNC solution inherits the outlier structure of the layer-wise Hessians. In practice, additional noise enters through the theorem's approximations, so our results both match and explain the fine-grained Hessian structure observed empirically (Sagun et al., 2016; 2017; Papyan, 2018; 2019; 2020). Notably, the theorem imposes minimal constraints on the feature-map Hessian blocks; since these layers also exhibit approximate DNC in practice, this will further reinforce the spectral structure. Overall, DNC strongly shapes the Hessian—and thus the local flatness—of converged solutions in overparameterized networks, suggesting that controlling the emergence of DNC can be used as a practical tool, given the importance of flatness for generalization (Wu et al., 2017), optimization (Li et al., 2022), and network design (Foret et al., 2020).

Our modeling assumptions posit that, in the depth limit, the network can attain arbitrarily high levels of collapse. This has been justified theoretically by Súkeník et al. (2026), though in practice it can be influenced by the optimization method and architecture (He & Su, 2022). Overall, the experiments of Papyan (2020) make it clear that levels of collapse in practice are sufficient for our results to hold. We leave a detailed exploration of architectural effects for future work.

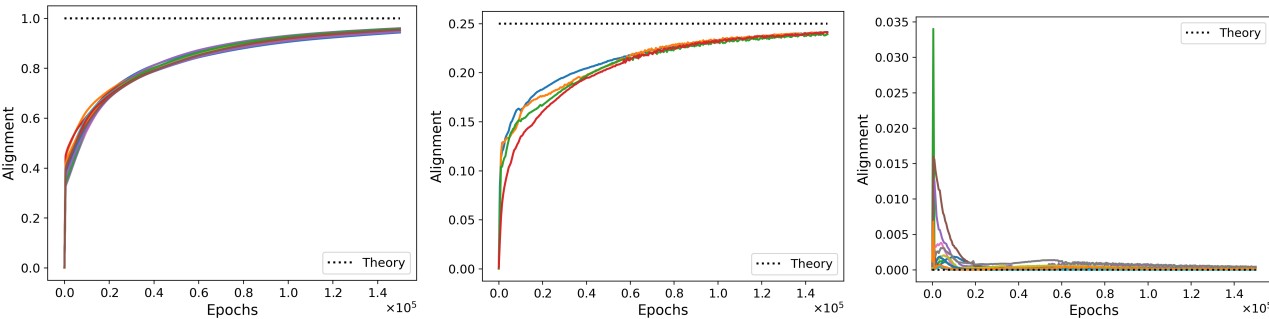

**Figure 5.** Early stages of training for a deep ReLU UFM at layer $l = 3$ with $K = 4, L = 4$. We zero out parameters with magnitude below $10^{-6}$ and set $\sigma'(0) = 1$. **Left**: Predicted eigenvector alignment, measured by the squared cosine similarity between $\mu_c^{(l+1)} \otimes \mu_{c'}^{(l)}$ and $\text{Hess}_l(\mu_c^{(l+1)} \otimes \mu_{c'}^{(l)})$. **Middle & Right**: Gradient alignment, measured by the decomposition coefficients of $\tilde{g}^{(l)}$ in the basis of predicted eigenvectors $\mu_c^{(l+1)} \otimes \mu_{c'}^{(l)}$. Middle: $c = c'$, right: $c \neq c'$.

# 4. Low Dimensional Structure in the Deep ReLU UFM

We now turn to ReLU activations. Súkeník et al. (2024) showed that DNC is not a global optimum in this case. Nevertheless, we analyze its implications due to the prevalence of DNC in real networks (Papyan et al., 2020). We show that the previous theorems hold for the best-performing DNC solutions, which emerge over long timescales.

The loss function is stated in Eq. (4), with $\sigma(x) = x1(x \geq 0)$. It is helpful to define the matrices $\tilde{W}_{l,ic}$ and $\tilde{A}_{l,ic}$ as

$$(\tilde{W}_{l,ic})_{xy} = (W_l)_{xy}\sigma'(h_{ic}^{(l)})_y,$$

$$\tilde{A}_{l,ic} = (\tilde{W}_{L,ic})(\tilde{W}_{L-1,ic})...(\tilde{W}_{l,ic}),$$

where $\sigma'(x) = 1(x \geq 0)$. In App. D.2 we show that for $1 \leq l < L$, the layer-wise Hessians take the form

$$\text{Hess}_l = \text{Av}_{ic}\left\{(\tilde{A}_{l+1,ic}^T\tilde{A}_{l+1,ic}) \otimes \sigma(h_{ic}^{(l)})\sigma(h_{ic}^{(l)})^T\right\}.$$

We now examine this object under DNC. Numerically, when DNC solutions arise in the ReLU case, the pre-ReLU feature matrices evolve to have nonnegative entries. This is intuitive: ReLU reduces the Frobenius norm of matrices with negative entries, and a lower norm requires the weight matrices to increase in scale to maintain the same fit loss, which in turn increases the overall loss through regularization. This is shown in the left panel of Figure 3, where the number of feature-vector entries below a small negative threshold diminishes to zero during training.

In the early stages, these DNC solutions resemble global minima of the linear case, modified by a rank-one update to enforce non-negativity. As training continues, this update decays to zero, and the final solution coincides with a global minimum of the linear model with all feature matrices nonnegative. This is supported by the right panel of

Figure 3, where the feature matrix is shown to converge to an orthogonal frame.

To demonstrate that this behavior will occur consistently, and is in fact optimal, we show that among the set of solutions satisfying the DNC properties identified by Papyan—and adopted as the definition by Súkeník et al. (2024)—those that minimize the loss are the ones that recover feature matrices that form non-negative orthogonal frames. The proof is in App. D.1.

**Theorem 4.1.** *Consider the deep ReLU UFM described in Eq. (4). Suppose the network width satisfies $d \geq K$, and the regularization parameter $\lambda$ satisfies Eq. (17). Define the matrices $\Lambda_l = \sigma(H_l)$, for $l = 2, \ldots, L$. Consider parameter matrices $(W_L, \ldots, W_1, H_1)$ satisfying the following properties for $l = 1, ...L$*

- *DNC1: $H_l = \bar{H}_l \otimes 1_n^\top, \quad \Lambda_l = \bar{\Lambda}_l \otimes 1_n^\top.$*
- *DNC2: $\bar{H}_l^\top \bar{H}_l \propto I_K, \quad \bar{\Lambda}_l^\top \bar{\Lambda}_l \propto I_K.$*
- *DNC3: The rows of the weight matrix $W_l$ are linear combinations of the columns of $\bar{H}_l$.*

*In addition, assume the network output $Z = W_L\sigma(\ldots W_2\sigma(W_1H_1)\ldots)$ aligns with an orthogonal frame, meaning $Z \propto I_K \otimes 1_n^\top$. Then the loss-minimizing solutions among this class are precisely the global minima of the linear model that also satisfy $H_l = \sigma(H_l)$ for all $l = 2, \ldots, L$.*

Motivated by the fact that these represent optimal DNC configurations, and numerically emerge when DNC occurs, we define DNC in the deep ReLU UFM as follows.

**Definition 4.2** (DNC Structure in the Deep ReLU UFM). DNC in this setting is any solution $(W_L^*, \ldots, W_1^*, H_1^*)$ that is a global minimum of the corresponding linear model, and additionally satisfies the following non-negativity condition for all features at each layer $l = 2, ..., L$

$$\sigma\left(h_{ic}^{(l)}\right) = h_{ic}^{(l)}.$$

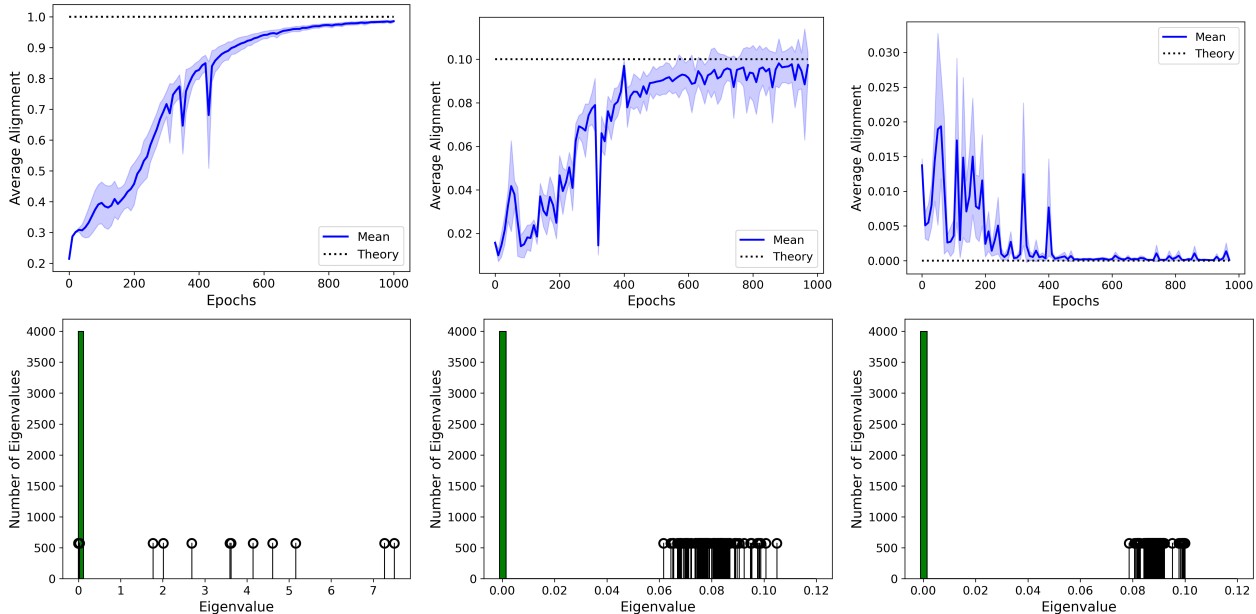

*Figure 6.* Training dynamics of a DNN at an intermediate separated layer $l$ on the MNIST dataset, when using linear layers in the FC head and UFM-style regularization. In the top row each plot shows quantities averaged over $c, c'$ with one–standard-deviation error bars. **Top Left**: Predicted eigenvector alignment, measured by the squared cosine similarity between $\mu_c^{(l+1)} \otimes \mu_{c'}^{(l)}$ and $\mathrm{Hess}_l(\mu_c^{(l+1)} \otimes \mu_{c'}^{(l)})$. **Top Middle & Right**: Gradient alignment, measured by the decomposition coefficients of $\tilde{g}^{(l)}$ in the basis of predicted eigenvectors $\mu_c^{(l+1)} \otimes \mu_{c'}^{(l)}$. Middle: $c = c'$, right: $c \neq c'$. **Bottom**: Histogram of the spectrum of $\mathrm{Hess}_l$ shown at epochs $0, 10^3$ and $4 \times 10^3$. The top $K^2 = 100$ outlier eigenvalues are plotted as spikes.

Using this definition, we have $(\tilde{W}_{l,ic}) = W_l$, and consequently $\tilde{A}_{l+1,ic} = A_{l+1}$. Moreover, since the activation functions have no effect at a DNC solution, the feature vectors $h_{ic}^{(l)}$ coincide with those of the linear model. This allows Theorems 3.2–3.4 to extend to the ReLU setting.

**Theorem 4.3.** *Consider the deep ReLU UFM described in Eq. (4). Suppose the network width satisfies $d \geq K$, and the regularization parameter $\lambda$ satisfies Eq. (17). Let the parameter values $(W_L^*, \ldots, W_1^*, H_1^*)$ have DNC structure as in Definition 4.2. Then the conclusions of Theorems 3.1–3.4 regarding the layer-wise Hessian, gradients, and weight matrices also hold in this nonlinear setting.*

The proof appears in App. D.2. Thus, our theoretical results extend beyond linear layers to ReLU layers, the activation function of choice in practice. This result adopts the convention $\sigma'(0) = 1$. Even without this convention, the minimal DNC solutions can be represented by linear networks; in that case, however, the layer-wise Hessians are not well defined due to non-differentiability at zero, which merely obscures the underlying structure. Setting $\sigma'(0) = 1$ allows us to recover the same Hessians under a minimal convention. We provide empirical confirmation of our theory in Figures 4 and 5, which exhibit the same phenomenology as in the linear case.

## 5. Numerical Experiments

We validate our theory on the MNIST (Lecun et al., 1998) dataset. In all experiments, the feature map is taken to be a ResNet-20, followed by a fully connected (FC) head. Additional training details, together with experiments on CIFAR-10 (Krizhevsky, 2009) and further evaluations of Hessian eigenvalues, the Hessian decomposition, and weight eigenvalues, are provided in App. G.

In Figure 6, we apply UFM-style regularization both to the output of the feature map and to each layer of the FC head, focusing on layer $l = 3$. We use linear layers in the FC head to match the setting covered by most of our theory. The top-left panel shows that the predicted eigenvectors rapidly align with the true layer-wise Hessian eigenvectors, with the alignment approaching one. The top-middle and top-right panels report the corresponding gradient decomposition: coefficients with $c = c'$ concentrate near $1/K = 0.1$, while those with $c \neq c'$ vanish. The bottom panels show the evolution of the Hessian spectra over training. Although the predicted spike structure and number of outliers are not present at initialization, the correct number of outliers emerges early in training. The spectral spread then decreases over time, with some residual variation remaining due to noise relative to the idealized approximations.

In Figure 7 we conduct the same experiment but with ReLU

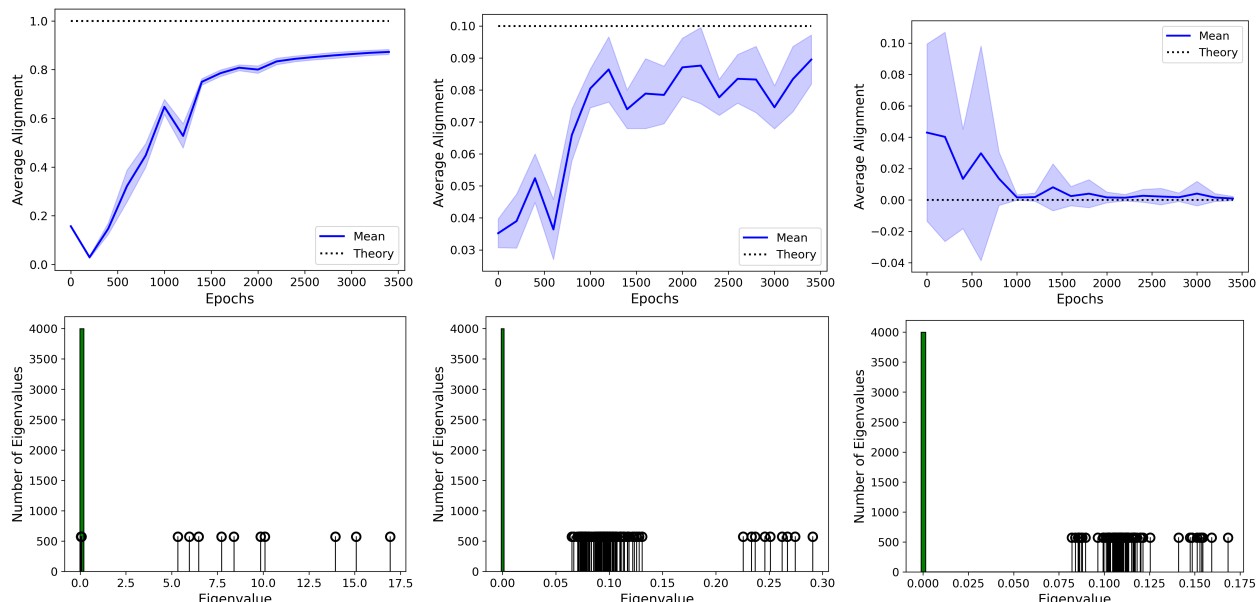

*Figure 7.* Training dynamics of a DNN at an intermediate separated layer $l$ on the MNIST dataset, when using ReLU layers in the FC head and standard-style regularization. In the top row each plot shows quantities averaged over $c, c'$ with one–standard-deviation error bars. **Top Left**: Predicted eigenvector alignment, measured by the squared cosine similarity between $\mu_c^{(l+1)} \otimes \mu_{c'}^{(l)}$ and $\mathrm{Hess}_l(\mu_c^{(l+1)} \otimes \mu_{c'}^{(l)})$. **Top Middle & Right**: Gradient alignment, measured by the decomposition coefficients of $\tilde{g}^{(l)}$ in the basis of predicted eigenvectors $\mu_c^{(l+1)} \otimes \mu_{c'}^{(l)}$. Middle: $c = c'$, right: $c \neq c'$. **Bottom**: Histogram of the spectrum of $\mathrm{Hess}_l$ shown at epochs $0, 10^3$ and $4 \times 10^3$. The top $K^2 = 100$ outlier eigenvalues are plotted as spikes.

layers in the FC head and standard regularization. One qualitatively observes the same phenomenology as the theory of the deep UFM, again with some levels of noise due to the approximations associated with the modeling assumptions of the theory. Overall, these results demonstrate that our theory extends beyond the deep UFM setting.

## 6. Concluding Remarks

In this work, we show that the feature means of DNC completely determine several canonical deep-learning objects—including the Hessian, weights, and gradients—in the overparameterized and training limits captured by the deep UFM. Our theory recovers fine-grained spectral structure: the Hessian exhibits $K^2$ outlier eigenvalues that decompose into $K$ dominant outliers and a $K(K-1)$ "mini-bulk," while the gradient is confined to the span of only $K$ outlier eigenvectors. Moving beyond prior empirical observations, we derive closed-form expressions for the relevant eigenvalues and eigenvectors. These formulas make explicit how DNC drives the emergence of this structure, thereby providing a unified explanation for the low-dimensional spectra commonly observed in deep learning. We establish these results for both linear and ReLU activations, and validate the predictions with experiments on standard classification architectures and canonical datasets.

By identifying DNC as a unifying mechanism governing curvature, gradient alignment, and weight structure, our results open avenues for characterizing optimization landscapes, developing principled diagnostics of training dynamics, and designing architectures and regularization schemes that improve performance by targeting DNC rather than less interpretable objects such as the Hessian directly.

**Limitations:** A comprehensive discussion of the modeling assumptions used in the development of this work is given in App. A.

## Acknowledgments

We thank the anonymous reviewers for their thoughtful feedback and constructive suggestions, which helped improve the clarity and presentation of the paper. JPK is grateful to Gérard Ben Arous for an extremely helpful discussion which stimulated this work. CG is supported by the Charles Coulson Scholarship. The authors also acknowledge support from His Majesty's Government in the development of this research. For the purpose of Open Access, the authors have applied a CC BY public copyright license to any Author Accepted Manuscript (AAM) version arising from this submission.

## Impact Statement

This paper presents work whose goal is to advance the field of machine learning, and more specifically, the theoretical understanding of neural collapse and deep matrix spectral phenomena. There are many potential societal consequences of our work, none of which we feel must be specifically highlighted here.

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

# A. Motivating the deep Unconstrained Feature Model

In this section, we justify the use of UFMs as a modeling tool by drawing on prior empirical and theoretical evaluations, as well as examining the plausibility of the underlying modeling assumptions.

## A.1. Experimental Evaluations of UFMs

**The UFM reproduces previously documented phenomena:** A fundamental criterion for the usefulness of any abstract model is its ability to capture known empirical behavior. The UFM has been the focus of an extensive body of work and has repeatedly matched empirical observations across a variety of settings—including neural collapse (Mixon et al., 2020; Zhu et al., 2021; Ji et al., 2022; Lu & Steinerberger, 2022), its variants under different loss functions (Zhou et al., 2022a;b; Han et al., 2022; Behnia & Thrampoulidis, 2024), DNC (Súkeník et al., 2023; Dang et al., 2023; Tirer & Bruna, 2022; Tirer et al., 2023), and geometric structures arising in high–class-count regimes (Jiang et al., 2023).

Our paper extends this line of evidence by showing that UFM predictions align with spectral and gradient-related phenomena observed in modern deep networks, as summarized in the introduction.

**The UFM generates new predictions that are subsequently validated:** A more compelling justification for a model is its ability to produce novel predictions that are later confirmed empirically. The literature provides many such examples for UFMs, including predictions of NC behavior under class imbalance (Fang et al., 2021; Thrampoulidis et al., 2022; Hong & Ling, 2023; Dang et al., 2023), the emergence of new low-rank solutions (Súkeník et al., 2024; Garrod & Keating, 2026), extensions to regression (Andriopoulos et al., 2024) and graph neural networks (Kothapalli et al., 2023), and analogues in language models (Thrampoulidis, 2024; Zhao et al., 2024).

Our work provides another such instance: the eigenvector structure of deep-network Hessians and other deep learning matrices, as described in our paper, had not been predicted previously. The UFM yields explicit analytic formulas, and our experiments show that real networks closely follow these predictions.

These two considerations—agreement with established empirical phenomena and the ability to forecast new ones—carry more weight than direct arguments about the plausibility of modeling assumptions in isolation. Without empirical alignment, such assumptions would be of limited relevance; with alignment, they become scientifically meaningful.

## A.2. The Theoretical Basis of the UFM

**The unconstrained feature assumption:** This assumption corresponds to the hypothesis that, in the overparameterized limit, the learned feature representations at global minima coincide with choosing the optimal feature representations of the training data directly. Under mild assumptions on the data, universal approximation results ensure that in the overparameterization limit the feature map can express the function that places the feature vectors at an optimal location. Thus, at global minima and in the high-parameter regime, the optimal features and the features of the actual network can coincide. This is the mathematical foundation underlying the assumption. For certain architectures, a direct correspondence between UFMs and real networks has been rigorously established by Súkeník et al. (2026).

This foundation, of course, idealizes several aspects of real training. Deep networks do not typically reach exact global minima, and the overparameterized limit is never literally achieved in practice. Our use of the model takes these limitations into account: we apply the UFM only to describe the behavior of highly overparameterized networks at convergence, not to model the trajectory of training. Ultimately, the validity of such assumptions is justified not in isolation but through the empirical validations discussed above.

Crucially, the UFM is not unusual in relying on idealizations. Many influential theoretical frameworks for deep learning employ assumptions known to hold only approximately: NTK theory (Jacot et al., 2018) invokes the infinite-width limit; spin-glass analogies (Choromanska et al., 2015) predict Hessian spectra that differ markedly from those observed in practice; and linearized models (Saxe et al., 2013) sacrifice expressiveness by design. Nonetheless, these models have been valuable precisely because, despite their idealizations, they capture essential empirical behaviors and have yielded significant insights into learning theory. We view the UFM similarly: its scientific utility derives not from the literal correctness of its assumptions but from its consistent empirical alignment with phenomena observed in practical overparameterized networks.

### A.3. Why the UFM Can Exactly Fit the Data

A defining feature of the UFM is that its feature map is assumed to be arbitrarily expressive. Consequently, the model can, in principle, map each input directly to its corresponding label and achieve zero training error. At first glance this might seem problematic. Below, we explain why this property is intentional and foundational to the design of the UFM.

**Modeling the Overparameterized Regime:** The central purpose of the UFM is to isolate and study the geometric structures that emerge in the limit of extreme overparameterization. In such regimes, real neural networks possess enough capacity to fit the training data exactly, and additional layers or parameters contribute progressively less to reducing the loss once interpolation is achievable.

The UFM captures this limiting behavior by assuming a feature map flexible enough to represent the data perfectly. This is not a flaw but an explicit modeling choice: it reflects the empirical fact that modern networks often operate far beyond the interpolation threshold. By doing so, the UFM allows us to analyze the organization of learned representations when capacity is not a limiting factor, which is precisely the setting emphasized in the original Neural Collapse (NC) work of Papyan et al. (2020) and numerous other empirical studies (Papyan, 2018; 2019; 2020). This controlled abstraction enables a clean examination of the geometric consequences of overparameterization without confounding details.

**Exact Fitting as a Standard Setting for Studying Implicit Bias:** The assumption that the model can fit the data exactly is also firmly rooted in a long tradition of theoretical work on implicit bias in optimization and deep learning. Many influential papers deliberately study training dynamics in interpolation settings—settings where the loss can be driven to zero—to reveal how gradient-based methods or depth implicitly favor certain solutions.

To highlight just a few canonical examples:

- Saxe et al. (2013) analyze how depth and gradient flow induce the sequential emergence of features in linear networks trained with MSE.

- Soudry et al. (2018) and Lyu & Li (2019) show that gradient descent on losses with exponential tails implicitly maximizes the margin in linearly separable classification.

- Arora et al. (2019) study deep matrix factorization and demonstrate that depth induces low-rank biases that improve generalization.

- Gunasekar et al. (2018) characterize how different optimization methods lead to different implicit biases in separable linear classification problems.

For a survey of this line of work, see the review by Vardi (2023).

The UFM should be understood as operating squarely within this tradition. NC and DNC manifest across many architectures and datasets, yet common loss functions contain no explicit term encouraging simplex equiangular tight frames or orthogonal frames. Their emergence instead arises as a consequence of implicit bias induced by overparameterization. The UFM provides an analytically tractable model for studying this bias in the limit of high overparameterization, and its correspondence is supported by extensive empirical evidence, as summarized previously in this appendix.

### A.4. Other Modelling Assumptions

**Regularization term:** For analytical simplicity, the UFM applies regularization directly to the features produced by the feature map, rather than to the network parameters themselves. Nevertheless, under hyperparameter settings typical of modern deep learning, UFMs and trained neural networks continue to exhibit closely aligned behavior (Súkeník et al., 2024; Garrod & Keating, 2026; Papyan et al., 2020). In addition, in Section 5 we provide empirical evidence that standard network architectures recover the same phenomenology under conventional parameter-space regularization schemes.

**MSE loss:** In the main text, we focus on the MSE loss rather than the more practically common CE loss. This is a standard tractability-driven choice in theoretical work (Tirer & Bruna, 2022; Tirer et al., 2023; Dang et al., 2023; Súkeník et al., 2023; 2024). It is further motivated by prior evidence that networks trained with MSE perform comparably to those trained with CE on real data (Hui & Belkin, 2020), while displaying very similar qualitative behavior. To clarify the effect of this modeling choice, Appendix B presents the CE analogue of our main Hessian result. The difference is confined to the

behavior of the outlier Hessian eigenvalues: under MSE, the minibulk and outlier spikes group together, whereas under CE they separate.

$L \to \infty$ **in Theorem 3.6:** Our full-Hessian result takes the limit $L \to \infty$, ensuring that most network layers satisfy DNC. By Theorem 3.5, this implies that the principal Hessian block has the stated eigenspectrum structure, which then extends to the full Hessian. Since the feature map is defined as the first layer at which the network becomes expressive enough for the UFM assumption to hold, this occurs at some fixed depth $\bar{L}$. Appending additional layers does not alter this definition or the precise value of $\bar{L}$. Accordingly, in practical network training, the limit $L \to \infty$ should be interpreted as a large-depth regime in which the full network is sufficiently deep. This of course still relies on the UFM assumption, which we discussed earlier in this appendix.

**Same Asymptotic Scale Hessian Blocks Assumption:** In Theorem 3.6, we assume all Hessian blocks have the same scale asymptotically. Intuitively this should hold for regularized homogeneous networks by the AM-GM inequality due the sum of scales in the regularization term. Additionally the theorem agrees and captures phenomenology observed in standard experiments in Hessian empirical works (Papyan, 2018; 2019; 2020). We leave further consideration of this assumption to future work.

## B. The Cross-Entropy Case

The cross-entropy version of the deep UFM was analyzed in detail by Garrod & Keating (2026). They showed that although DNC is not a global optimum of this model, it remains a critical point with a positive semi-definite Hessian, and under certain hyperparameter choices it is preferentially selected during training. Here, we examine the implications of DNC in such a model for the layer-wise Hessians of the network. The corresponding unconstrained feature model is defined as

$$\mathcal{L}(W_L, ..., W_1, H_1) = g(W_L...W_1 H_1) + \sum_{l=1}^{L} \frac{1}{2}\lambda \|W_l\|_F^2 + \frac{1}{2}\lambda \|H_1\|_F^2, \tag{7}$$

where $g$ implements the CE loss

$$g(Z) = -\frac{1}{Kn} \sum_{c=1}^{K} \sum_{i=1}^{n} \log \left( \frac{\exp((z_{ic})_c)}{\sum_{c'=1}^{K} \exp((z_{ic})_{c'})} \right), \tag{8}$$

and $z_{ic} = W_L...W_1 h_{ic}$ are the logit vectors forming the columns of the matrix $Z \in \mathbb{R}^{K \times Kn}$, using the same ordering as for the feature matrices. As before, $\lambda > 0$ is a regularization coefficient applied equally to all parameters. We also define the class feature means to be the same as in Section 2.

Within this model, Zhu et al. (2021) and Garrod & Keating (2026) clarify that the appropriate definition of DNC is the following:

**Definition B.1** (DNC in the Deep CE UFM). A parameter set $(W_L, ..., W_1, H_1)$ of the CE deep UFM exhibits DNC structure if the following hold

$$Z = \alpha S \otimes 1_n^T, \quad \text{where } S = I_K - \frac{1}{K} 1_K 1_K^T,$$

and the constant $\alpha$ is given by the larger solution to

$$\lambda n^{\frac{1}{L+1}} = \frac{1}{(K-1) + e^\alpha} \alpha^{\frac{L-1}{L+1}}. \tag{9}$$

Furthermore, the parameter matrices admit singular value decompositions of the following form

$$W_l = U_l \Sigma U_{l-1}^T, \quad \text{for } l = 1, ..., L-1; \quad W_L = U_L \tilde{\Sigma} U_{L-1}^T; \quad H_1 = U_0 \bar{\Sigma} V_0^T, \tag{10}$$

where $U_L \in \mathbb{R}^{K \times K}$, $V_0 \in \mathbb{R}^{Kn \times Kn}$, $U_{L-1}, ..., U_0 \in \mathbb{R}^{d \times d}$ are all orthogonal matrices. The matrices $\Sigma \in \mathbb{R}^{d \times d}$, $\tilde{\Sigma} \in \mathbb{R}^{K \times d}$, and $\bar{\Sigma} \in \mathbb{R}^{d \times Kn}$ have their top $K \times K$ block equal to $(\alpha\sqrt{n})^{\frac{1}{L+1}} \text{diag}(1, ..., 1, 0)$, and all other entries zero.

Garrod & Keating (2026) showed that such a solution exists and is a critical point with a positive semi-definite Hessian whenever $\lambda$ is sufficiently small and $d \geq K$. While this definition may appear detached from the original notion of DNC, they demonstrate that it naturally implies all the DNC properties. Moreover, Zhu et al. (2021) show that in the $L = 1$ case it coincides with the global minimum. From Eq. (9), it is also clear that as $\lambda \to 0$, the logit scale $\alpha$ diverges.

Defining the layer-wise Hessian as in the main text, we obtain the following theorem concerning its spectrum at convergence to a DNC structure.

**Theorem B.2.** *Consider the deep UFM with CE loss given in Eq. (7), with network width $d \geq K$ and regularization parameter $\lambda$ small enough that the DNC structure of Definition B.1 is a critical point. For a layer $l$ with $1 \leq l < L$, the layer-wise Hessian decomposes as the sum of two terms, one of which vanishes exponentially faster in the logit scale $\alpha \to \infty$ limit associated with small regularization. Writing $\hat{\nu}_{cc'}^{(l)} = \hat{\mu}_c^{(l+1)} \otimes \hat{\mu}_{c'}^{(l)}$, the leading-order term at DNC can be written as*

$$Hess_l = \beta \left( \frac{K-1}{K} \right)^2 \left[ \sum_c \hat{\nu}_{cc}^{(l)} \hat{\nu}_{cc}^{(l)T} + \frac{1}{K} \sum_{c,c'} \hat{\nu}_{cc'}^{(l)} \hat{\nu}_{cc'}^{(l)T} \right],$$

*where*

$$\beta = \frac{e^\alpha (\alpha \sqrt{n})^{\frac{2L}{L+1}}}{n((K-1) + e^\alpha)^2}.$$

*The nonzero spectrum consists of the following $(K-1)^2$ eigenvalues:*

- *Eigenvalue $\beta$ with multiplicity 1, eigenvector $\sum_c \hat{\nu}_{cc}^{(l)}$.*

- *Eigenvalue $\frac{K-1}{K}\beta$, multiplicity $K-1$, eigenvectors of the form $\hat{\nu}_{cc}^{(l)} - \hat{\nu}_{c'c'}^{(l)}$, $c \neq c'$, $c, c' = 1, ..., K$.*

- *Eigenvalue $\frac{1}{K}\beta$, multiplicity $K^2 - 3K + 1$, eigenvectors of the form $\hat{\nu}_{cc'}^{(l)} - \hat{\nu}_{c'c}^{(l)}$, $c \neq c'$, $c, c' = 1, ..., K$.*

This result reveals a clear separation into bulk, mini-bulk, and outliers, as observed in Papyan (2020). Specifically, the bulk corresponds to the $d^2 - (K-1)^2$ zero eigenvalues, the mini-bulk to the $K^2 - 3K + 1$ smaller but nonzero eigenvalues, and the $K$ spikes to the large eigenvalues, including a single dominant one. Thus, the separation reported by Papyan (2020) arises from the choice of loss function, and its absence in the main text is not a consequence of the UFM assumption. Apart from this distinction, the interpretation parallels that of the MSE case.

The only difference to what Papyan reported empirically is that here we recover $(K-1)^2$ total spikes, whereas Papyan recovers $K^2$. This arises from the nonlinearity in Papyan's work: with ReLU activations, it is shown by Dang et al. (2024) that networks adopt orthogonal frame structure, with rank $K$, rather than simplex ETF structure, with rank $K-1$. This change due to activation raises the Hessian rank to that observed empirically.

We include this section to emphasize that the absence of a mini-bulk in the main text stems from the loss function rather than from the unconstrained feature assumption. Similar explanation can likely be used to recover the other results of Section 3, indeed the Hessian result is by far the most intricate. We leave confirmation of this to future work.

## C. Deep Linear UFM Proofs

In this appendix, we provide proofs of the theoretical results for the deep linear UFM. Supporting lemmas appear in App. C.7, while the model definition is given in Eq. (4) and additional notation in Section 2. Most results are stated and proved for layers $1 \leq l < L$. The case $l = L$ also holds, but requires additional care with matrix dimensions since $W_L \in \mathbb{R}^{K \times d}$ rather than $\mathbb{R}^{d \times d}$. For clarity of exposition, we omit this case.

### C.1. Proof of Theorem 3.1

We consider the Hessian with respect to the parameters of a given layer $W_l$ for $1 \leq l < L$, ignoring the regularization terms. It is convenient to express the objective function in terms of the individual training points. Defining $u_{ic} = W_L...W_1 h_{ic} - y_c$,

we can write

$$\mathcal{L}(H_1, W_1, ..., W_L) = \mathrm{Av}_{ic}\left\{\frac{1}{2}\|W_L...W_1 h_{ic} - y_c\|_2^2\right\} = \mathrm{Av}_{ic}\left\{\frac{1}{2}u_{ic}^T u_{ic}\right\}.$$

Taking second derivatives gives

$$\frac{\partial^2 \mathcal{L}}{\partial(W_l)_{ab}\partial(W_l)_{ef}} = \mathrm{Av}_{ic}\left\{\frac{\partial u_{ic}}{\partial(W_l)_{ab}} \cdot \frac{\partial u_{ic}}{\partial(W_l)_{ef}}\right\} + \mathrm{Av}_{ic}\left\{\frac{\partial^2 u_{ic}}{\partial(W_l)_{ab}(W_l)_{ef}} \cdot u_{ic}\right\}.$$

The second term clearly vanishes in the linear case. We can then calculate the relevant first derivative

$$\frac{\partial(u_{ic})_x}{\partial(W_l)_{ab}} = (A_{l+1})_{xa}(h_{ic}^{(l)})_b, \quad \text{where } A_{l+1} = W_L...W_{l+1}. \tag{11}$$

Substituting this into the expression above gives

$$\frac{\partial^2 \mathcal{L}}{\partial(W_l)_{ab}\partial(W_l)_{ef}} = (A_{l+1}^T A_{l+1})_{ae}\mathrm{Av}_{ic}\left\{(h_{ic}^{(l)})_b(h_{ic}^{(l)})_f\right\}.$$

To write this as a $d^2 \times d^2$ matrix, we flatten $W_l$ using the standard convention $(w_l)_{d(x-1)+y} = (W_l)_{xy}$. This induces the following Kronecker structure in the layer-wise Hessian

$$\mathrm{Hess}_l = \frac{\partial^2 \mathcal{L}}{\partial^2 w_l} = A_{l+1}^T A_{l+1} \otimes \left[\mathrm{Av}_{ic}\left\{h_{ic}^{(l)} h_{ic}^{(l)T}\right\}\right]. \tag{12}$$

We now assume we are at a global optimum, and so can use the properties of deep neural collapse as detailed in Lemma C.1. Specifically, assume the network width obeys $d \geq K$, and assume the level of regularization obeys the condition stated in Eq. (17). Then we have by the DNC1 property of Lemma C.1 that $h_{ic} = \mu_c$, where $\mu_c = \mathrm{Av}_i\{h_{ic}\}$. In addition, recalling that $\mu_c^{(l)} = W_{l-1}...W_1\mu_c$, our Hessian is now

$$\mathrm{Hess}_l = A_{l+1}^T A_{l+1} \otimes \left[\mathrm{Av}_c\left\{\mu_c^{(l)} \mu_c^{(l)T}\right\}\right].$$

Also recall the matrix of class means at the $l$th layer, denoted $\bar{H}_l = [\mu_1^{(l)}, ..., \mu_K^{(l)}] \in \mathbb{R}^{d \times K}$. By Lemma C.2 this matrix also forms an orthogonal frame in the sense that $\bar{H}_l^T \bar{H}_l \propto I_K$.

We begin by considering the right side of our Kronecker product. Since the class means are orthogonal, the matrix $\frac{1}{K}\sum_{c=1}^K \mu_c^{(l)}\mu_c^{(l)T}$ is already in the form of an eigen-decomposition. This matrix has rank $K$ and is simply a scaled projection onto $\mathrm{Span}\{\mu_c^{(l)}\}_{c=1}^K$ with eigenvectors $\hat{\mu}_c^{(l)}$. In addition, since the class means have the same norm, the nonzero eigenvalues are all equal, with value $\|\mu^{(l)}\|_2^2/K$.

For the left side of our Kronecker product, note that from the DNC3 property of Lemma C.1, we have $A_{l+1}^T \propto \bar{H}_{l+1}$, and so the rows $a_c^{(l+1)}$ of $A_{l+1}$ are such that $a_c^{(l+1)} = \alpha^{(l+1)}\mu_c^{(l+1)}$, for some constant $\alpha^{(l+1)}$. Hence

$$A_{l+1}^T A_{l+1} = (\alpha^{(l+1)})^2 \bar{H}_{l+1}\bar{H}_{l+1}^T = (\alpha^{(l+1)})^2 \sum_{c=1}^K \mu_c^{(l+1)}\mu_c^{(l+1)T},$$

and again this is the eigen-decomposition of this matrix, showing this matrix is also rank $K$, and has eigenvectors $\hat{\mu}_c^{(l+1)}$ with all eigenvalues equal to $(\alpha^{(l+1)})^2\|\mu^{(l+1)}\|_2^2$.

As a consequence, our layer-wise Hessian has the following eigen-decomposition

$$\mathrm{Hess}_l = \frac{(\alpha^{(l+1)})^2}{K}\|\mu^{(l+1)}\|_2^2\|\mu^{(l)}\|_2^2 \sum_{c,c'=1}^K \left[\hat{\mu}_c^{(l+1)} \otimes \hat{\mu}_{c'}^{(l)}\right]\left[\hat{\mu}_c^{(l+1)} \otimes \hat{\mu}_{c'}^{(l)}\right]^T.$$

It remains to re-express the coefficient in terms of a constant $C$ that solves an algebraic equation in the hyper-parameters. First note, as a consequence of Lemma C.1, we have that $W_L...W_1\bar{H}_1 \propto I_K$, call the constant of proportionality $C$. We also have that

$$W_L...W_1\bar{H}_1 = A_{l+1}\bar{H}_{l+1} = \alpha^{(l+1)}\bar{H}_{l+1}^T\bar{H}_{l+1} = \alpha^{(l+1)}\|\mu^{(l+1)}\|_2^2 I_K,$$

and so

$$C = \alpha^{(l+1)}\|\mu^{(l+1)}\|_2^2. \tag{13}$$

Using this to replace $\alpha^{(l+1)}$ in the eigen-decomposition of $\text{Hess}_l$, along with the result of Lemma C.5 that gives the feature mean norms in terms of the constant $C$, gives the form stated in the theorem. From this the eigenvalues and eigenvectors stated in the theorem can be read off.

We only need now show that $C$ is the solution of an algebraic equation in the hyper-parameters. By Lemma C.4, we have that at any optimal point the loss can be written as

$$\mathcal{L} = \frac{1}{2Kn}\|Z - Y\|_F^2 + \frac{1}{2}\lambda(L+1)\|Z\|_{S_{\frac{2}{L+1}}}^{\frac{2}{L+1}},$$

where $Z = W_L...W_1H_1$ and $\|\cdot\|_{S_p}$ is a Schatten quasi-norm with parameter $p$. Using that $Z = CI_K \otimes 1_n^T$, and hence has singular values $C\sqrt{n}$, with multiplicity $K$, this gives that the loss at a DNC solution is given by

$$\mathcal{L} = \frac{1}{2}(1-C)^2 + \frac{1}{2}K\lambda(L+1)n^{\frac{1}{L+1}}C^{\frac{2}{L+1}}.$$

Since DNC represents a global minimum, the value of $C$ must minimize this loss, and so $\partial_C\mathcal{L} = 0$. This gives

$$1 = \underbrace{C + K\lambda n^{\frac{1}{L+1}}C^{-\frac{L-1}{L+1}}}_{f(C)}.$$

Wherever DNC is globally optimal this equation has a solution, and so this algebraic equation specifies the value of $C$ subject to our regularization condition. Note this equation has two solutions when $\lambda$ is small enough, this can be seen by considering the derivative of $f(C)$, which shows $f(C)$ only has one turning point and diverges as $C \to 0, \infty$. The larger solution clearly performs better on the loss, since when $C$ small $\mathcal{L} \approx 1$ whereas when $C \approx 1$, $\mathcal{L} \approx 0$, and hence the larger solution is the global minimum.

### C.2. Proof of Theorem 3.2

Using our unconstrained feature approximation, the mapping of a data point is $f(x_{ic}; \theta) = W_L...W_1h_{ic}$. Ignoring regularization terms, the Hessian at layer $l$ can be decomposed using the Gauss-Newton decomposition, which follows from the chain and product rules

$$= \underbrace{\text{Av}_{ic}\left\{\frac{\partial f(x_{ic};\theta)^T}{\partial w_l}\frac{\partial^2 l(z, y_c)}{\partial z^2}\Big|_{z=z_{ic}}\frac{\partial f(x_{ic};\theta)}{\partial w_l}\right\}}_{G_l} + \underbrace{\text{Av}_{ic}\left\{\sum_{c'}\frac{\partial l(z, y_c)}{\partial z_{c'}}\Big|_{z_{ic}}\frac{\partial^2 f_{c'}(x_{ic};\theta)}{\partial w_l^2}\right\}}_{E_l},$$

where $z_{ic} = f(x_{ic};\theta)$. Denote the first of these by $G_l$ and the second by $E_l$. In our case, since $\partial_{w_l}^2 f_{c'}(x_{ic};\theta) = 0$ we have that $E_l = 0$, and so $\text{Hess}_l = G_l$.

Noting that $\frac{\partial^2 l(z, y_c)}{\partial z^2} = I$, we can write $\text{Hess}_l$ in the following form

$$\text{Hess}_l = G_l = \sum_{i,c,c'}\frac{1}{Kn}v_{icc'}v_{icc'}^T, \quad \text{where } v_{icc'} = \frac{\partial f_{c'}(x_{ic};\theta)}{\partial w_l}.$$

Calculating the derivative gives $v_{icc'} = a_{c'}^{(l+1)} \otimes h_{ic}^{(l)}$, with $(a_{c'}^{(l+1)})_x = (A_{l+1})_{c'x}$ being the rows of $A_{l+1}$. Note this is a weighted second moment matrix over the objects $v_{icc'}$.

The analogy to Papyan's decomposition for our MSE case is the following: first defining the quantities

$$v_{cc'} = \sum_i \frac{1}{n} v_{icc'}, \qquad v_c = \sum_{c'} \frac{1}{K} v_{cc'}.$$

We decompose as

$$G_l = G_{l,\text{class}} + G_{l,\text{cross}} + G_{l,\text{within}},$$

where the components of the decomposition are given by

$$G_{l,\text{class}} = \sum_c v_c v_c^T,$$

$$G_{l,\text{cross}} = \sum_c G_{l,\text{cross},c}, \quad \text{where} \quad G_{l,\text{cross},c} = \sum_{c'} \frac{1}{K} (v_{cc'} - v_c)(v_{cc'} - v_c)^T,$$

$$G_{l,\text{within}} = \sum_{c,c'} \frac{1}{K} G_{l,\text{within},c,c'}, \quad \text{where} \quad G_{l,\text{within},c,c'} = \sum_i \frac{1}{n} (v_{icc'} - v_{cc'})(v_{icc'} - v_{cc'})^T.$$

We can now again assume we are at a global optimum, that $d \geq K$, and $\lambda$ obeys Eq. (17). We can then use the properties of DNC outlined in Lemma C.1. The DNC1 property of Lemma C.1 gives us that $h_{ic} = \mu_c$, and hence $v_{icc'} = v_{cc'}$. As a consequence $G_{l,\text{within}} = 0$ and does not contribute to the outlier eigenvalues of the spectrum.

In addition using Lemma C.2, and the DNC3 property of Lemma C.1, gives us that $a_{c'}^{(l+1)} = \alpha^{(l+1)} \mu_{c'}^{(l+1)}$, for some constant $\alpha^{(l+1)}$. So at the optima we have

$$v_{cc'} = \alpha^{(l+1)} \mu_{c'}^{(l+1)} \otimes \mu_c^{(l)},$$

as well as

$$v_c = \alpha^{(l+1)} \mu_G^{(l+1)} \otimes \mu_c^{(l)}, \qquad v_{cc'} - v_c = \alpha^{(l+1)} (\mu_{c'}^{(l+1)} - \mu_G^{(l+1)}) \otimes \mu_c^{(l)}.$$

So, the forms of our decomposition components are given by

$$G_{l,\text{class}} = \alpha^{(l+1)2} \sum_{c=1}^{K} (\mu_G^{(l+1)} \otimes \mu_c^{(l)})(\mu_G^{(l+1)} \otimes \mu_c^{(l)})^T,$$

$$G_{l,\text{cross}} = \frac{\alpha^{(l+1)2}}{K} \sum_{c,c'=1}^{K} ([\mu_{c'}^{(l+1)} - \mu_G^{(l+1)}] \otimes \mu_c^{(l)})([\mu_{c'}^{(l+1)} - \mu_G^{(l+1)}] \otimes \mu_c^{(l)})^T.$$

This provides an eigen-decomposition of $G_{l,\text{class}}$, it is simply a scaled projection operator onto $\text{Span}\{\mu_G^{(l+1)} \otimes \mu_c^{(l)}\}_{c=1}^{K}$, and so has rank $K$. The eigenvectors are clearly given by $\hat{\mu}_G^{(l+1)} \otimes \hat{\mu}_c^{(l)}$ for $c \in \{1, ..., K\}$. Also note the eigenvalues are equal to what was reported in Theorem 3.1, since $\|\mu_G^{(l+1)}\|_2^2 = \frac{1}{K}\|\mu^{(l+1)}\|_2^2$, and using this with the expressions for $C$ in Eq. (13) and Eq. (18) recovers the previously stated value.

We also see $G_{l,\text{cross}}$ has image $\text{Span}\{(\mu_{c'}^{(l+1)} - \mu_G^{(l+1)}) \otimes \mu_c^{(l)}\}_{c,c'=1}^{K}$. The globally centered means $\mu_{c'}^{(l+1)} - \mu_G^{(l+1)}$ are clearly not linearly independent, since they sum to zero, and so span a $K-1$ dimensional space. They in fact form a simplex ETF, as a consequence the rank of $G_{l,\text{cross}}$ is $K(K-1)$. It is easy to verify that one choice of spanning eigenvectors for the nonzero eigenvalues is given by $(\mu_1^{(l+1)} - \mu_{c'}^{(l+1)}) \otimes \mu_c^{(l)}$, for $c' \in \{2, ..., K\}$, $c \in \{1, ..., K\}$, with eigenvalues equal to what was reported in Theorem 3.1.

It is also easy to see that the vectors in the two projections are orthogonal, and so $G_{l,\text{class}}$ and $G_{l,\text{cross}}$ knockout different parts of the projection in $G_l$, or equivalently $\text{Hess}_l$.

## C.3. Proof of Theorem 3.3

We now consider the layer-wise gradient, denoted $g_l$, for $1 \leq l < L$, at a global optimum, and show how it relates to the top eigenspace of the corresponding layer-wise Hessian. As before we have

$$\mathcal{L} = \mathrm{Av}_{ic} \left\{ \frac{1}{2} \|W_L...W_1 h_{ic} - y_c\|_2^2 \right\} + \frac{1}{2} \lambda \|W_l\|_F^2 + ....$$

Writing $u_{ic} = W_L...W_1 h_{ic} - y_c$, and using the expression for $\partial_{W_l} u_{ic}$ from Eq. (11), we obtain the derivative

$$\frac{\partial \mathcal{L}}{\partial (W_l)_{ab}} = \lambda (W_l)_{ab} + \mathrm{Av}_{ic} \left\{ (A_{l+1}^T u_{ic})_a (h_{ic}^{(l)})_b \right\}.$$

After flattening $W_l$, as described in Section 3.1, we obtain

$$g^{(l)} = \frac{\partial \mathcal{L}}{\partial w^{(l)}} = \lambda w^{(l)} + \underbrace{\mathrm{Av}_{ic} \left\{ (A_{l+1}^T u_{ic}) \otimes h_{ic}^{(l)} \right\}}_{\tilde{g}^{(l)}}. \tag{14}$$

We now focus on the term $\tilde{g}^{(l)}$. We again assume we are at a global optimum, that the network width satisfies $d \geq K$, and the level of regularization $\lambda$ obeys the condition detailed in Eq. (17). The DNC1 property of Lemma C.1 gives $h_{ic} = \mu_c$, which simplifies our quantity at this global optimum to

$$\tilde{g}_l = \mathrm{Av}_{ic} \left\{ (A_{l+1}^T u_{ic}) \otimes h_{ic}^{(l)} \right\} = \mathrm{Av}_c \left\{ (A_{l+1}^T u_c) \otimes \mu_c^{(l)} \right\},$$

where we have defined the quantity $u_c = W_L...W_1 \mu_c - y_c$.

We consider the left term in our Kronecker product separately. Recall from Lemma C.1 and Corollary C.2 that the rows $a_c^{(l+1)}$ of $A_{l+1}$ have the form $a_c^{(l+1)} = \alpha^{(l+1)} \mu_c^{(l+1)}$. Hence

$$A_{l+1}^T = \alpha^{(l+1)} [\mu_1^{(l+1)}...\mu_K^{(l+1)}],$$

and so

$$A_{l+1}^T y_c = \alpha^{(l+1)} \mu_c^{(l+1)}.$$

Also,

$$A_{l+1}^T W_L...W_1 \mu_c = A_{l+1}^T A_{l+1} \mu_c^{(l+1)}.$$

Using the form of $A_{l+1}$ We find that

$$A_{l+1}^T A_{l+1} \mu_c^{(l+1)} = \alpha^{(l+1)2} \|\mu^{(l+1)}\|_2^2 \mu_c^{(l+1)},$$

and thus,

$$A_{l+1}^T u_c = \alpha^{(l+1)} \left[ \alpha^{(l+1)} \|\mu^{(l+1)}\|_2^2 - 1 \right] \mu_c^{(l+1)}.$$

Using the expression for the constant $\alpha^{(l+1)}$ in terms of the constant $C$ stated in Eq. (13), this simplifies to

$$A_{l+1}^T u_c = \frac{C}{\|\mu^{(l+1)}\|_2^2} \left[ C - 1 \right] \mu_c^{(l+1)}. \tag{15}$$

Substituting this into the expression of $\tilde{g}^{(l)}$ then gives, at optima,

$$\tilde{g}^{(l)} = \frac{C(C-1)}{K} \frac{\|\mu^{(l)}\|_2}{\|\mu^{(l+1)}\|_2} \sum_{c=1}^{K} \hat{\mu}_c^{(l+1)} \otimes \hat{\mu}_c^{(l)}.$$

Then using Eq. (18) to express the feature mean norms in terms of $C$ gives the final result.

Recall that the set of eigenvectors of the corresponding layer-wise Hessian at a global optimum are given by $\{\hat{\mu}_{c'}^{(l+1)} \otimes \hat{\mu}_c^{(l)}\}_{c,c'=1}^{K}$. We see that with this natural basis the gradient lies in a $K$ dimensional subspace with equal coefficients for each of these $K$ eigenvectors.

### C.4. Proof of Theorem 3.4

We now consider each weight matrix $W_l$ at a global optimum. Again, we assume the network width satisfies $d \geq K$ and the level of regularization $\lambda$ obeys the condition detailed in Eq. (17).

First note that since, at any optimum, the first order derivatives are zero, we have

$$\frac{\partial \mathcal{L}}{\partial W_l} = \lambda W_l + \frac{1}{Kn} A_{l+1}^T [W_L...W_1 H_1 - Y] H_l^T = 0 \tag{16}$$

$$\implies W_l = -\frac{1}{Kn\lambda} A_{l+1}^T [W_L...W_1 H_1 - Y] H_l^T.$$

Using the DNC2 and DNC1 properties from Lemma C.1, we have that, at a global optimum, $W_L...W_1 H_1 = C(I_K \otimes 1_n^T)$, and $H_l = \bar{H}_l \otimes 1_n^T$. Substituting these into the expression for $W_l$, we obtain

$$W_l = \frac{1-C}{Kn\lambda} A_{l+1}^T [I_K \otimes 1_n^T](\bar{H}_l \otimes 1_n^T)^T$$

$$= \frac{1-C}{K\lambda} A_{l+1}^T \bar{H}_l^T.$$

Using the DNC3 property from Lemma C.1, which states that $A_{l+1}^T = \alpha^{(l+1)} \bar{H}^{(l+1)}$, we further simplify this expression to

$$W_l = \frac{1-C}{K\lambda} \alpha^{(l+1)} \bar{H}^{(l+1)} \bar{H}^{(l)T}.$$

Next, using the fact that $\bar{H}_{ij}^{(l)} = (\mu_j^{(l)})_i$ we can rewrite the product of the $H$ matrices as

$$W_l = \frac{1-C}{K\lambda} \alpha^{(l+1)} \sum_c \mu_c^{(l+1)} \mu_c^{(l)T}.$$

Finally, substituting the expression for $\alpha^{(l+1)}$ in terms of $C$ in Eq. (13) we obtain

$$W_l = \frac{C(1-C)}{K\lambda} \frac{\|\mu^{(l)}\|_2}{\|\mu^{(l+1)}\|_2} \sum_c \hat{\mu}_c^{(l+1)} \hat{\mu}_c^{(l)T}.$$

Then using Eq. (18) to express the feature mean norms in terms of $C$ gives the final result.

This form is precisely a singular value decomposition of $W_l$. As a consequence, $W_l$ clearly has rank $K$, and the nonzero singular values, as well as the left and right singular vectors, can be immediately identified.

### C.5. Proof of Theorem 3.5

We now wish to consider the Hessian with respects to the parameters $(W_L, ..., W_1)$. First note, using the notation $u_{ic} = W_L...W_1 h_{ic} - y_c$ as before, we have:

$$\frac{\partial^2 \mathcal{L}}{\partial (W_l)_{xy} \partial (W_r)_{ab}} = \text{Av}_{ic} \left\{ \frac{\partial u_{ic}}{\partial (W_l)_{xy}} \cdot \frac{\partial u_{ic}}{\partial (W_r)_{ab}} + \frac{\partial^2 u_{ic}}{\partial (W_l)_{xy} \partial (W_r)_{ab}} \cdot u_{ic} \right\}.$$

The second term did not contribute to the diagonal terms of the Hessian, which we computed earlier. However for off-diagonal terms it is now nonzero. The first of these terms has a similar form to the diagonal case:

$$\text{Av}_{ic} \left\{ \frac{\partial u_{ic}}{\partial (W_l)_{xy}} \cdot \frac{\partial u_{ic}}{\partial (W_r)_{ab}} \right\} = (A_{l+1}^T A_{r+1})_{xa} \, \text{Av}_{ic} \left\{ (h_{ic}^{(l)})_y (h_{ic}^{(r)})_b \right\}.$$

Whilst for off-diagonal, taking wlog $r > l$, we have the second term given by

$$\text{Av}_{ic} \left\{ \frac{\partial^2 u_{ic}}{\partial (W_l)_{xy} \partial (W_r)_{ab}} \cdot u_{ic} \right\} = \text{Av}_{ic} \left\{ (A_{r+1}^T u_{ic})_a (W_{r-1}...W_{l+1})_{bx} (h_{ic}^{(l)})_y \right\}.$$

Intuitively, one expects this term to be small. This is for the same reason as argued by (Papyan, 2020): for well-performing networks $u_{ic} \approx 0$, and since $u_{ic}$ appears explicitly in this term, it should also be approximately zero. To verify this one can consider the second terms scale and compare it to the other term. First we assume we are at a global minimum and that the relevant conditions hold. The exact same steps used in the proof of Theorem 3.1 give that the first term, post flattening, is given by

$$\frac{1}{K} n^{\frac{-1}{L+1}} C^{\frac{2L}{L+1}} \sum_{c,c'} \left[ \hat{\mu}_c^{(l+1)} \otimes \hat{\mu}_{c'}^{(l)} \right] \left[ \hat{\mu}_c^{(r+1)} \otimes \hat{\mu}_{c'}^{(r)} \right]^T,$$

and note since $C \to 1$ as $\lambda \to 0$, this term does not decay for small regularization. The second term can be written, using Eq. (15), as (here we do not flatten it, since it does not have a nice Kronecker expression):

$$C(C-1) \frac{\|\mu^{(l)}\|_2}{\|\mu^{(r+1)}\|_2} (W_{r-1}...W_{l+1})_{bx} \text{Av}_{ic} \left\{ (\hat{\mu}_c^{(r+1)})_a (\hat{\mu}_c^{(l)})_y \right\}.$$

Next using Lemma C.3 to separate the scales of the weight matrices out, and Eq. (18) to write the feature mean norms in terms of $C$, this becomes

$$= n^{\frac{-1}{L+1}} C^{\frac{L-1}{L+1}} (C-1)(U_{r-1} D U_l^T) \text{Av}_{ic} \left\{ (\hat{\mu}_c^{(r+1)})_a (\hat{\mu}_c^{(l)})_y \right\},$$

where $D \in \mathbb{R}^{d \times d}$ has its top $K \times K$ block be $\text{diag}(1, ..., 1, 0)$, and all other entries zero. It is clear to see that as $C \to 1$, which occurs as $\lambda \to 0$, this term has norm tending to zero, and so can be seen as a perturbative term. We now drop it and only work with the terms coming from the leading order component of the Hessian.

Hence, to leading order, our top $L$ layer Hessian is given by

$$\text{Hess}_{1:L} = \begin{bmatrix} \text{Hess}^{(1,1)} & ... & \text{Hess}^{(1,L)} \\ ... & ... & ... \\ \text{Hess}^{(L,1)} & ... & \text{Hess}^{(L,L)} \end{bmatrix},$$

where

$$\text{Hess}^{(l,r)} = \frac{1}{K} n^{\frac{-1}{L+1}} C^{\frac{2L}{L+1}} \sum_{c,c'} \left[ \hat{\mu}_c^{(l+1)} \otimes \hat{\mu}_{c'}^{(l)} \right] \left[ \hat{\mu}_c^{(r+1)} \otimes \hat{\mu}_{c'}^{(r)} \right]^T.$$

Now using that $\hat{\mu}_c^{(l+1)} = U_l D U_L^T e_c$ (this comes from Lemma C.3 and $A_{l+1}^T \propto \bar{H}_{l+1}$), we have:

$$\sum_c \hat{\mu}_c^{(l+1)} \hat{\mu}_c^{(r+1)T} = U_l D U_r^T,$$

and hence

$$\text{Hess}^{(l,r)} = \frac{1}{K} n^{\frac{-1}{L+1}} C^{\frac{2L}{L+1}} (U_l \otimes U_{l-1})(D \otimes D)(U_r \otimes U_{r-1})^T.$$

Consequently, defining $Q_l = U_l \otimes U_{l-1}$, we can write:

$$\text{Hess}_{1:L} = \frac{1}{K} n^{\frac{-1}{L+1}} C^{\frac{2L}{L+1}} \begin{bmatrix} Q_1(D \otimes D)Q_1^T & ... & Q_1(D \otimes D)Q_L^T \\ ... & ... & ... \\ Q_L(D \otimes D)Q_1^T & ... & Q_L(D \otimes D)Q_L^T \end{bmatrix}$$

$$= \frac{1}{K} n^{\frac{-1}{L+1}} C^{\frac{2L}{L+1}} \text{diag}(Q_1, ..., Q_L)(D \otimes D \otimes 1_L 1_L^T)\text{diag}(Q_1, ..., Q_L)^T,$$

where the matrix $\text{diag}(Q_1, ..., Q_L)$ is understood to be block diagonal, and inherits orthogonality from the orthogonal matrices $Q_l$.

Hence the matrix $\text{Hess}_{1:L}$ has the same eigenvalues as the matrix $\frac{1}{K} n^{\frac{-1}{L+1}} C^{\frac{2L}{L+1}} (D \otimes D \otimes 1_L 1_L^T)$, consequently it has $K^2$ nonzero eigenvalues, each taking the value $\frac{1}{K} n^{\frac{-1}{L+1}} C^{\frac{2L}{L+1}} L$.

### C.6. Proof of Theorem 3.6

Recall that we assume the feature map $h(x; \bar{\theta})$ has $\bar{L}$ many layers, and then denote the Hessian of the full network by

$$\text{Hess}(\theta) = \begin{bmatrix} \text{Hess}_{\bar{L},\bar{L}} & \text{Hess}_{\bar{L},L} \\ \text{Hess}_{L,\bar{L}} & \text{Hess}_{L,L} \end{bmatrix}.$$

We can write this as

$$\text{Hess}(\theta) = \underbrace{\begin{bmatrix} 0 & 0 \\ 0 & \text{Hess}_{L,L} \end{bmatrix}}_{\widetilde{\text{Hess}}} + \underbrace{\begin{bmatrix} \text{Hess}_{\bar{L},\bar{L}} & \text{Hess}_{\bar{L},L} \\ \text{Hess}_{L,\bar{L}} & 0 \end{bmatrix}}_{\tilde{P}}.$$

We shall show, subject to the assumptions, that the second term can be viewed as a perturbation to the first matrix.

We prove the result assuming that all layers have equal scale Hessian blocks, it should be clear that the same result goes through by assuming the same asymptotic scale with the same steps. Labeling this scale $\nu$, we have:

$$\|\text{Hess}_{\bar{L},\bar{L}}\|_F^2 = \bar{L}^2 \nu^2,$$

$$\|\text{Hess}_{\bar{L},L}\|_F^2 = \bar{L}L\nu^2,$$

$$\|\text{Hess}_{L,L}\|_F^2 = L^2 \nu^2,$$

and hence

$$\left\|\tilde{P}\right\|_F^2 = \left(\bar{L}^2 + 2\bar{L}L\right)\nu^2,$$

$$\left\|\widetilde{\mathrm{Hess}}\right\|_F^2 = L^2\nu^2.$$

Recalling that $\bar{L} << L$, we see that the ratio of their Frobenius norms is

$$\frac{\left\|\tilde{P}\right\|_F^2}{\left\|\widetilde{\mathrm{Hess}}\right\|_F^2} \sim \frac{1}{L}.$$

We now apply Weyl's inequality: this states that for $A, B$ two symmetric matrices of the same dimension, and $\lambda_i(M)$ denoting the $i^{\mathrm{th}}$ largest singular value of the matrix $M$, we have

$$|\lambda_i(A + B) - \lambda_i(B)| \le \|A\|_F.$$

Applying this with $B = \widetilde{\mathrm{Hess}}$, $A = \tilde{P}$ gives

$$\left|\lambda_i\left(\mathrm{Hess}(\theta)\right) - \lambda_i\left(\widetilde{\mathrm{Hess}}\right)\right| \le \|\tilde{P}\|_F.$$

Now use the normalized matrices, using $\hat{\lambda}_i(M)$ to denote the eigenvalues of $M/\|M\|_F$, we have

$$\left|\left(\bar{L} + L\right)\nu\hat{\lambda}_i\left(\mathrm{Hess}(\theta)\right) - L\nu\hat{\lambda}_i\left(\widetilde{\mathrm{Hess}}\right)\right| \le \sqrt{\bar{L}^2 + 2\bar{L}L}\nu,$$

which reduces to

$$\left|\left(1 + \frac{\bar{L}}{L}\right)\hat{\lambda}_i\left(\mathrm{Hess}(\theta)\right) - \hat{\lambda}_i\left(\widetilde{\mathrm{Hess}}\right)\right| \le \sqrt{\frac{\bar{L}^2}{L^2} + 2\frac{\bar{L}}{L}}.$$

It is then simple to see that as $L \to \infty$

$$\hat{\lambda}_i\left(\mathrm{Hess}(\theta)\right) - \hat{\lambda}_i\left(\widetilde{\mathrm{Hess}}\right) \to 0.$$

Since $\widetilde{\mathrm{Hess}}$ clearly has the same top $K^2$ eigenvalues as $\mathrm{Hess}_{L,L}$, with the remaining being zero, this gives:

$$\hat{\lambda}_i\left(\mathrm{Hess}(\theta)\right) - \hat{\lambda}_i\left(\mathrm{Hess}_{L,L}\right) \to 0, \quad \text{for } i = 1, ..., K^2,$$
$$\hat{\lambda}_i\left(\mathrm{Hess}(\theta)\right) \to 0, \quad \text{for } i > K^2.$$

## C.7. Supporting Lemmas

The supporting lemmas necessary for the proofs in App. C are provided here.

**Lemma C.1** (from Theorem 3.1 in Dang et al. (2023)). *Consider the deep linear UFM described in Eq. (4). Let the network width satisfy $d \ge K$, and assume the level of regularization $\lambda$ is such that*

$$0 < \sqrt[L]{Kn\lambda^{L+1}} < \frac{1}{KL^2}(L-1)^{\frac{L-1}{L}}. \tag{17}$$

*Let the set of parameter values $(W_L^*, W_{L-1}^*, \dots, W_1^*, H_1^*)$ be a global optimizer. Then, the following properties hold for all $l = 1, ..., L$*

- **DNC1:** $H_1^* = \bar{H}^* \otimes 1_n^\top$, where $\bar{H}^* = [\mu_1, \ldots, \mu_K] \in \mathbb{R}^{d \times K}$.

- **DNC2:** $\bar{H}^{*\top}\bar{H}^* \propto W_L^* W_{L-1}^* \cdots \bar{H}^* \propto (W_L^* W_{L-1}^* \cdots W_l^*)(W_L^* W_{L-1}^* \cdots W_l^*)^\top \propto I_K$.

- **DNC3:** $W_L^* W_{L-1}^* \cdots W_1^* \propto \bar{H}^{*\top}, \quad W_L^* W_{L-1}^* \cdots W_l^* \propto (W_{l-1}^* \cdots W_1^* \bar{H}^*)^\top$.

**Lemma C.2.** *Under the context of Lemma C.1, the columns of $\bar{H}_l = W_{l-1} \cdots W_1 \bar{H}_1$ form an orthogonal frame in the sense that $\bar{H}_l^\top \bar{H}_l \propto I_K$.*

**Proof**: Although this is not explicitly stated in Lemma C.1, it follows immediately from the DNC2 and DNC3 properties in the lemma statement.

**Lemma C.3** (from Lemma 1 in Garrod & Keating (2026)). *Let $d \geq K$. At any optimal point of the deep linear UFM described in Eq. (4), there exists a singular value decomposition (SVD) of the parameter matrices of the following form*

$$W_l = U_l \Sigma U_{l-1}^\top, \quad for \ l = 1, \ldots, L-1,$$

$$W_L = U_L \tilde{\Sigma} U_{L-1}^\top, \quad H_1 = U_0 \bar{\Sigma} V_0^\top,$$

*where $U_L \in \mathbb{R}^{K \times K}$, $V_0 \in \mathbb{R}^{Kn \times Kn}$, $U_{L-1}, \ldots, U_0 \in \mathbb{R}^{d \times d}$ are all orthogonal matrices, $\Sigma \in \mathbb{R}^{d \times d}$, $\tilde{\Sigma} \in \mathbb{R}^{K \times d}$, and $\bar{\Sigma} \in \mathbb{R}^{d \times Kn}$ are diagonal or block-diagonal matrices whose top $K \times K$ blocks are given by $\mathrm{diag}(\sigma_1, \ldots, \sigma_K)$, with all other entries being zero.*

Note whilst (Garrod & Keating, 2026) consider a slightly different model to ours, the proof hinges only on the use of $L_2$ regularization, and the exact same steps in their proof can be used for our model.

**Lemma C.4.** *The loss $\mathcal{L}$ at an optimal point can be expressed entirely in terms of the matrix $Z = W_L \cdots W_1 H_1$ as*

$$\mathcal{L} = \frac{1}{2Kn}\|Z - Y\|_F^2 + \frac{1}{2}\lambda(L+1)\|Z\|_{S_{\frac{2}{L+1}}}^{\frac{2}{L+1}},$$

*where $\|\cdot\|_{S_p}$ denotes the Schatten quasi-norm with parameter $p$.*

**Proof:** consider an optimal point of the loss. Using Lemma C.3, We have that if $Z = W_L...W_1 H_1$ has SVD given by $Z = U_L \mathrm{diag}[(\sigma_1, ..., \sigma_K), 0_{K \times K(n-1)}]V_0^T$, then each parameter matrix has at most $K$ nonzero singular values given by $\sigma_1^{\frac{1}{L+1}}, ..., \sigma_K^{\frac{1}{L+1}}$. Using that the Frobenius norm is the sum of the squares of the singular values, we have

$$\mathcal{L} = \frac{1}{2Kn}\|Z - Y\|_F^2 + \frac{1}{2}\lambda(L+1)\sum_{c=1}^{K} \sigma_c^{\frac{2}{L+1}}.$$

Using the definition of the Schatten quasi norm then immediately gives the result.

**Lemma C.5.** *Suppose we are at a global minimum of the deep UFM, $d \geq K$, and the regularization condition of Eq. (17) holds. Then the norms of the feature means can be written as*

$$\|\mu^{(l)}\|_2 = \frac{1}{\sqrt{n}}(C\sqrt{n})^{\frac{l}{L+1}}, \tag{18}$$

*where $C$ is given by the network output as $W_L...W_1 H_1 = CI \otimes 1_n^T$.*

**Proof:** Note $H_l = W_{l-1}...W_1 H_1$. By Lemma C.3, we have

$$H_l = U_l \bar{\Sigma}^l V_0^T$$

Where for a DNC solution the nonzero singular values in $\bar{\Sigma}^l$ are $(C\sqrt{n})^{\frac{l}{L+1}}$ with multiplicity $K$. Taking Frobenius norm squared of both sides then gives

$$Kn\|\mu^{(l)}\|_2^2 = K(C\sqrt{n})^{\frac{2l}{L+1}}$$

Which gives

$$\|\mu^{(l)}\|_2 = \frac{1}{\sqrt{n}}(C\sqrt{n})^{\frac{l}{L+1}}$$

## D. Deep ReLU UFM Proofs

Here we detail the forms of the layer-wise Hessians, gradients, and weights for the deep ReLU UFM. The deep ReLU UFM loss function is provided in Eq. (4).

### D.1. Proof of Theorem 4.1

Here, we demonstrate that among solutions satisfying the DNC properties described by Súkeník et al. (2024), the ones that minimize the loss are precisely those from Lemma C.1 that additionally have all their intermediate representations with non-negative entries. We establish this by showing that such solutions attain a lower bound on the loss within the class of considered solutions. We shall assume the same regularization condition, stated in Eq. (17), as in the linear case. This ensures that the best-performing DNC solution outperforms the zero solution, allowing us to restrict our analysis to cases where no parameter matrix is zero.

We express the parameter matrices with their scales separated

$$W_l = \alpha_l \hat{W}_l, \quad H_1 = \alpha_0 \hat{H}_1, \quad \text{where } \alpha_i > 0, \ i = 0, ..., L.$$

The matrices $\hat{W}_l$ for $l = 1, ..., L$ and $\hat{H}_1$ have unit Frobenius norm. Using the homogeneity of ReLU, we can write the output matrix $Z$ as

$$Z = W_L \sigma(...W_2 \sigma(W_1 H_1)...) = \left(\prod_{l=0}^{L} \alpha_l\right) \hat{W}_L \sigma(...\hat{W}_2 \sigma(\hat{W}_1 \hat{H}_1)...).$$

Since we assume that $Z$ aligns with the matrix $I_K \otimes 1_n^T$, we write

$$\hat{W}_L \sigma(...\hat{W}_2 \sigma(\hat{W}_1 \hat{H}_1)...) = \beta I_K \otimes 1_n^T.$$

Substituting this into the loss, and using the parameter matrices decompositions, the loss for a DNC solution is

$$\mathcal{L}_{\text{DNC}} = \frac{1}{2Kn}\|Z - Y\|_F^2 + \frac{1}{2}\lambda\|H_1\|_F^2 + \frac{1}{2}\lambda\sum_{l=1}^{L}\|W_l\|_F^2$$

$$= \frac{1}{2}\left(\beta\prod_{l=1}^{L}\alpha_l - 1\right)^2 + \frac{1}{2}\lambda\sum_{l=0}^{L}\alpha_l^2 := f(\beta, \alpha_0, ..., \alpha_L).$$

We now argue that for a fixed $\beta$, the function $f(\beta, \alpha_0, ..., \alpha_L)$ is minimized when all $\alpha_l$ are equal. First, since $f \to \infty$ if any $\alpha_l \to \infty$, the minimum must occur at a finite turning point. Taking a derivative,

$$\frac{\partial f}{\partial \alpha_l} = \lambda\alpha_l + \beta\left(\prod_{l' \neq l}^{L}\alpha_{l'}\right)\left(\beta\prod_{l'=0}^{L}\alpha_{l'} - 1\right).$$

Setting this to zero, and using that $\alpha_l \neq 0$, gives

$$\lambda \alpha_l^2 = \beta \left( \prod_{l'=0}^{L} \alpha_{l'} \right) \left( \beta \prod_{l'=0}^{L} \alpha_{l'} - 1 \right).$$

Since the right-hand side is the same for all $l$, it follows that each $\alpha_l$ must be equal at the minimum. Thus, defining $\alpha$ as the common value,

$$\mathcal{L}_{\text{DNC}} \geq \min_\alpha \underbrace{\left\{ \frac{1}{2}(\beta \alpha^{L+1} - 1)^2 + \frac{1}{2}\lambda(L+1)\alpha^2 \right\}}_{f(\beta,\alpha)} := g(\beta),$$

where we have defined the function $f(\beta, \alpha)$ to acknowledge we are focused on the $\alpha_l = \alpha$ case, as well as the function $g(\beta)$ where the minimum over $\alpha$ of $f(\beta, \alpha)$ is attained.

We now claim $g(\beta)$ is monotonically decreasing for $\beta > 0$. Let $0 < \beta_1 < \beta_2$, and let $\alpha' = \operatorname{argmin}_\alpha\{f(\beta_1, \alpha)\}$. Defining

$$\alpha^* = \alpha' \left( \frac{\beta_1}{\beta_2} \right)^{\frac{1}{L+1}},$$

we obtain

$$g(\beta_1) = \frac{1}{2}(\beta_1(\alpha')^{L+1} - 1)^2 + \frac{1}{2}(L+1)(\alpha')^2$$

$$> \frac{1}{2}(\beta_1(\alpha')^{L+1} - 1)^2 + \frac{1}{2}(L+1)(\alpha')^2 \left( \frac{\beta_1}{\beta_2} \right)^{\frac{2}{L+1}} = f(\beta_2, \alpha^*) \geq g(\beta_2),$$

which gives $g(\beta_1) > g(\beta_2)$, confirming $g(\beta)$ is monotonically decreasing.

To summarize what we have shown so far, the loss of any DNC solution is lower bounded by a solution where the frames are chosen so as to maximize $\beta$, and the scales $\alpha_l$ are equal and set to the value that minimizes $f(\beta_{\max}, \alpha)$. We now show that the solutions that maximize $\beta$ are precisely the ones with positive intermediate representations.

Recall we define $\Lambda_l = \sigma(H_l)$ for $l > 1$. By assumption we have that $H_l$ and $\Lambda_l$ have the following forms

$$H_l = \bar{H}_l \otimes 1_n^T, \quad \Lambda_l = \bar{\Lambda}_l \otimes 1_n^T,$$

where the matrices $\bar{H}_l, \bar{\Lambda}_l$ align with an orthogonal frame. This implies that their SVDs can be written as

$$\bar{H}_l = \gamma_l U_l^H \tilde{D}^T Q^T, \quad \bar{\Lambda}_l = \rho_l U_l^\Lambda \tilde{D}^T Q^T,$$

where $Q \in \mathbb{R}^{K \times K}$ and $U_l^H, U_l^\Lambda \in \mathbb{R}^{d \times d}$ are orthogonal matrices, $\tilde{D} = [I_K, 0_{K \times (d-K)}] \in \mathbb{R}^{K \times d}$, and $\gamma_l, \rho_l$ are scales that give the nonzero singular values of $\bar{H}_l$ and $\bar{\Lambda}_l$ respectively.

We now aim to derive an upper bound for $\beta$ in terms of the singular values of the normalized features and weight matrices. For each $l$, we have the equation $H_l = W_{l-1}\Lambda_{l-1}$. Since these matrices have repeated columns, it follows that $\bar{H}_l = W_{l-1}\bar{\Lambda}_{l-1}$. Denote the first $K$ singular values of $W_{l-1}$ by $\omega_i^{(l-1)}$ for $i = 1, ..., K$, arranged in decreasing order. All other singular values are zero by the assumed third DNC property. Given that the singular values of $\bar{H}_l$ are $\gamma_l$ with multiplicity $K$, and a similar statement holds for $\bar{\Lambda}_{l-1}$ with $\rho_{l-1}$ being the nonzero singular values, Lemma D.1 gives us the following inequality for $2 \leq l \leq L+1$

$$\gamma_l \leq \omega_K^{(l-1)} \rho_{l-1}. \tag{19}$$

This gives us an inequality relating $\gamma_l$ to $\rho_{l-1}$. We will now get an inequality that relates $\rho_{l-1}$ to $\gamma_{l-1}$. We have $\Lambda_l = \sigma(H_l)$, and using that there are repeated columns this gives $\bar{\Lambda}_l = \sigma(\bar{H}_l)$. Taking Frobenius norm squared and using the property $\|\sigma(M)\|_F^2 \leq \|M\|_F^2$, with equality only if $\sigma(M) = M$, we obtain

$$\|\bar{\Lambda}_l\|_F^2 \leq \|\bar{H}_l\|_F^2,$$

which, using the forms of these matrices, gives

$$\rho_l \le \gamma_l. \tag{20}$$

This inequality also extends to $l = 1$, where it is trivially satisfied since we do not apply a nonlinearity at this layer, and so $H_1 = \Lambda_1$. Using the two inequalities in Eq. (19) and Eq. (20) as a recurrence relation in $\gamma_l$ and $\rho_l$, we derive

$$\gamma_{L+1} \le \gamma_1 \omega_K^{(L)} ... \omega_K^{(1)}.$$

Since $H_{L+1} = Z$, and hence $\gamma_{L+1} = \beta \prod_{l=0}^{L} \alpha_l$, we have

$$\beta \prod_{l=0}^{L} \alpha_l \le \gamma_1 \omega_K^{(L)} ... \omega_K^{(1)}.$$

Using $\omega_K^{(l)} / \alpha_l = \hat\omega_K^{(l)}$, where $\hat\omega_K^{(l)}$ are the smallest potentially nonzero singular values of the normalized weight matrices, and noting that $\gamma_1 / \alpha_0 = 1/\sqrt{nK}$, we arrive at

$$\beta \le \frac{1}{\sqrt{nK}} \hat\omega_K^{(L)} ... \hat\omega_K^{(1)}.$$

Since $\hat{W}_l$ are rank $K$ matrices with unit Frobenius norm, the maximum value of their smallest nonzero singular value occurs when all singular values are equal, giving $\omega_K^{(l)} = 1/\sqrt{K}$. Thus,

$$\beta \le \frac{1}{\sqrt{n}} K^{-\frac{L+1}{2}}.$$

Going back through the conditions for this inequality to be attained, we have the following

1. Each $W_l$ has $K$ equal nonzero singular values.

2. Intermediate representations are positive for $l = 2...., L$, meaning $H_l = \sigma(H_l)$.

3. The singular values of $\Lambda_l$ and $H_l$ obey $\gamma_l = \omega_K^{(l-1)} \rho_{l-1}$.

These conditions are only on the frames of the parameter matrices, we also require the following of their scales

4. For $l_1, l_2 \in \{0, ..., L\}$ we have $\alpha_{l_1} = \alpha_{l_2} = \alpha$.

5. The value $\alpha$ attains the minimum of the function $f(\beta_{\max}, \alpha)$, where $\beta_{\max} = n^{-\frac{1}{2}} K^{-\frac{L+1}{2}}$.

Condition (2) implies that the nonlinearity has no effect for the best-performing DNC solutions. As a result, the network function can be expressed equivalently by a linear network, and has a loss that is attainable in the linear case. We know the global minimum of our linear loss is given by solutions with the structure described in Lemma C.1, and a subset of these global minima obey condition (2). Since these are the best performing linear solutions and can be expressed by a nonlinear network, they are the only candidates for the minimal DNC solutions in the nonlinear case. It follows immediately from Lemma C.1, as well as properties detailed in Lemmas D.1 and C.3, that they obey all of the other conditions, and so the best performing DNC solutions in the nonlinear model are precisely the set of global minimum solutions of the linear model, with the extra condition that the intermediate representations are non-negative.

## D.2. Proof of Theorem 4.3

Here we show the implications of the first four theorems carry over to the ReLU case when DNC solutions arise. We will assume $2 \leq l < L$ for the duration of this proof. The result also holds for $l = 1$, but requires a slightly different description due to the convention of not using a nonlinearity after the matrix $H_1$. As before, when examining the Hessian, we drop the regularization terms. For this subsection we define the vectors $u_{ic}$ as

$$u_{ic} = W_L \sigma(...W_2 \sigma(W_1 h_{ic})...) - y_c,$$

so that we can express our MSE loss as

$$\mathcal{L} = \text{Av}_{ic}\left\{\frac{1}{2} u_{ic}^T u_{ic}\right\}.$$

Consequently, our layer-wise Hessian, before flattening the weights, is given by

$$\frac{\partial^2 \mathcal{L}}{\partial (W_l)_{ab} \partial (W_l)_{ef}} = \text{Av}_{ic}\left\{\frac{\partial u_{ic}}{\partial (W_l)_{ab}} \cdot \frac{\partial u_{ic}}{\partial (W_l)_{ef}} + u_{ic} \cdot \frac{\partial^2 u_{ic}}{\partial (W_l)_{ab} \partial (W_l)_{ef}}\right\}.$$

The second term is zero here since we take $\sigma$ to be ReLU, which is a piece-wise linear function. We note that

$$\frac{\partial (u_{ic})_d}{\partial (W_l)_{ab}} = \sum_{ef} (W_L)_{de} 1((h_{ic}^{(L)})_e \geq 0)(W_{L-1})_{ef} \frac{\partial}{\partial (W_l)_{ab}} \sigma(h_{ic}^{(L-1)})_f.$$

Now, define the following matrices for $2 \leq l \leq L$

$$(\tilde{W}_{l,ic})_{xy} = (W_l)_{xy} 1((h_{ic}^{(l)})_y \geq 0).$$

Continuing,

$$= \sum_{ef} (\tilde{W}_{L,ic})_{de} (W_{L-1})_{ef} \frac{\partial}{\partial (W_l)_{ab}} \sigma(h_{ic}^{(L-1)})_f.$$

Repeating this process leads to

$$= \sum_{ef} (\tilde{W}_{L,ic}...\tilde{W}_{l+1,ic})_{de} \frac{\partial}{\partial (W_l)_{ab}} [(W_l)_{ef} \sigma(h_{ic}^{(l)})_f]$$

$$= \sum_{ef} (\tilde{W}_{L,ic}...\tilde{W}_{l+1,ic})_{de} \delta_{ae} \delta_{bf} \sigma(h_{ic}^{(l)})_f.$$

Define the matrix

$$\tilde{A}_{l+1,ic} = \tilde{W}_{L,ic}...\tilde{W}_{l+1,ic}.$$

Thus, we obtain

$$\frac{\partial (u_{ic})_d}{\partial (W_l)_{ab}} = (\tilde{A}_{l+1,ic})_{da} \sigma(h_{ic}^{(l)})_b,$$

and

$$\frac{\partial \mathcal{L}}{\partial (W_l)_{ab} \partial (W_l)_{ef}} = \text{Av}_{ic}\left\{(\tilde{A}_{l+1,ic}^T \tilde{A}_{l+1,ic})_{ae} \sigma(h_{ic}^{(l)})_b \sigma(h_{ic}^{(l)})_f\right\}.$$

Finally, after flattening, we have

$$\text{Hess}_l = \frac{\partial^2 \mathcal{L}}{\partial w_l^2} = \text{Av}_{ic}\Big\{(\tilde{A}_{l+1,ic}^T \tilde{A}_{l+1,ic}) \otimes \sigma(h_{ic}^{(l)})\sigma(h_{ic}^{(l)})^T\Big\}.$$

Using the definition of DNC in the ReLU case from Definition 4.2, We have that $\sigma(h_{ic}^{(l)}) = h_{ic}^{(l)}$. Thus, $1((h_{ic}^{(l)})_y \geq 0) = 1$ for all $y \in \{1, .., d\}$ and for all $l \in \{2, ..., L\}$. This implies $\tilde{W}_{l,ic} = W_l$, and so $\tilde{A}_{l+1,ic} = A_{l+1}$. Consequently, the Hessian simplifies to

$$\text{Hess}_l = A_{l+1}^T A_{l+1} \otimes \Big[\text{Av}_{ic}\Big\{h_{ic}^{(l)} h_{ic}^{(l)T}\Big\}\Big].$$

Since each feature matrix passes through the nonlinearity with no changes, the features $h_{ic}^{(l)}$ are the same as those which occur in the linear network with the same parameter matrices.

The layer-wise Hessian of a solution with DNC structure has the same form as the linear case, and the DNC solutions by definition are global minimum of the linear model and thus obeys the properties detailed in Lemmas C.1, C.2, C.3 and C.4. Hence, the proofs of Theorem 3.1 and Theorem 3.2 in the linear case, detailed in App. C, follow through identically.

We can now consider the gradients. Using the working from above it is simple to show the gradients in the deep UFM have the following form

$$\frac{\partial \mathcal{L}}{\partial (W_l)_{ab}} = \lambda(W_l)_{ab} + \text{Av}_{ic}\Big\{(\tilde{A}_{l+1,ic}^T u_{ic})_a \sigma(h_{ic}^{(l)})_b\Big\}.$$

After flattening, and using that $\sigma(h_{ic}) = h_{ic}$ we then have

$$g^{(l)} = \lambda w^{(l)} + \text{Av}_{ic}\Big\{(A_{l+1}^T u_{ic}) \otimes h_{ic}^{(l)}\Big\}.$$

We also have for our DNC structure that $u_{ic}$ match the values of the corresponding linear model, and hence this again reduces to the linear gradient, and the proof of Theorem 3.3, detailed in App. C.3, follows through exactly the same.

In the case of the weights, since the DNC structure obeys the same properties as in the linear case, they will necessarily have the same singular value spectrum and singular vectors, and so Theorem 3.4 also carries over trivially.

### D.3. Supporting Lemmas

The supporting lemma necessary for the proofs in App. D is provided here.

**Lemma D.1** (from Horn & Johnson (1985)). *Let $A \in \mathbb{R}^{m \times k}$ and $B \in \mathbb{R}^{k \times n}$. Denoting the $i^{th}$ singular value of a matrix by $\sigma_i(\cdot)$, ordered in descending magnitude, we have*

$$\sigma_i(AB) \leq \sigma_i(A)\,\sigma_1(B).$$

## E. Other Deep Learning Matrices

Here, we examine the covariance of gradients and backpropagation errors, describing the structure at global minima for the deep linear UFM.

### E.1. Covariance of Gradients

The covariance of gradients was considered empirically by Jastrzebski et al. (2020). Specifically, we examine the matrix

$$C^{(l)} = \text{Av}_{ic}\Big\{(g_{ic}^{(l)} - g^{(l)})(g_{ic}^{(l)} - g^{(l)})^T\Big\}, \tag{21}$$

where

$$g_{ic}^{(l)} = \nabla_{w^{(l)}} l(f(x_{ic}; \theta), y_c), \quad g^{(l)} = \text{Av}_{ic}\{g_{ic}^{(l)}\}.$$

Note that we only consider the gradient with respect to the MSE loss, not the full loss including the regularizes, and work at a layer-wise level. From Eq. (14) in App. C.3, we recall that

$$\nabla_{w^{(l)}} l(f(x_{ic}; \theta), y_c) = A_{l+1}^T u_{ic} \otimes h_{ic}^{(l)}.$$

Under the regularization condition in Eq. (17) and assuming $d \geq K$, Eq. (15) in App. C.3 gives $A_{l+1}^T u_{ic}$, in terms of the feature means and the constant $C$, leading to

$$g_{ic}^{(l)} = C(C-1)\frac{\|\mu^{(l)}\|_2}{\|\mu^{(l+1)}\|_2}\hat{\mu}_c^{(l+1)} \otimes \hat{\mu}_c^{(l)}, \quad g^{(l)} = \frac{C(C-1)}{K}\frac{\|\mu^{(l)}\|_2}{\|\mu^{(l+1)}\|_2}\sum_{c=1}^{K}\hat{\mu}_c^{(l+1)} \otimes \hat{\mu}_c^{(l)}.$$

Writing $\hat{\nu}_c^{(l)} = \hat{\mu}_c^{(l+1)} \otimes \hat{\mu}_c^{(l)}$ to compress the notation, we find that at the optimum, $C^{(l)}$ can be expressed as

$$C^{(l)} = \frac{C^2(C-1)^2}{K}\frac{\|\mu^{(l)}\|_2^2}{\|\mu^{(l+1)}\|_2^2}\sum_{c=1}^{K}\left[\hat{\nu}_c^{(l)} - \text{Av}_{c'}\left\{\hat{\nu}_{c'}^{(l)}\right\}\right]\left[\hat{\nu}_c^{(l)} - \text{Av}_{c''}\left\{\hat{\nu}_{c''}^{(l)}\right\}\right]^T.$$

The vectors $\hat{\nu}_c^{(l)}$, for $c \in \{1, ..., K\}$, are orthogonal, and so the centered vectors

$$\hat{\nu}_c^{(l)} - \text{Av}_{c'}\{\hat{\nu}_{c'}^{(l)}\}, \quad \text{for } c \in \{1, ..., K\},$$

form a simplex ETF and span a $K-1$ dimensional subspace. Additionally, it follows that $C^{(l)}$ has a spanning set of eigenvectors with nonzero eigenvalue given by

$$\hat{\nu}_1^{(l)} - \hat{\nu}_c^{(l)}, \quad \text{for } c \in \{2, ..., K\},$$

each with eigenvalue

$$\frac{1}{K}n^{\frac{-1}{L+1}}C^{\frac{2L}{L+1}}(C-1)^2,$$

where we used Eq. (18) to express the feature mean norms in terms of $C$.

Thus, we state the following theorem for the second moment of gradients.

**Theorem E.1.** *Consider the deep linear UFM described in Eq. (4). Let the network width satisfy $d \geq K$, and consider a layer $l$ satisfying $1 \leq l < L$. Additionally, assume that the regularization parameter $\lambda$ satisfies the condition in Eq. (17). Then, at any global optimum of the loss, the layer-wise covariance of gradients matrix $C^{(l)}$, defined in Eq. (21), takes the form*

$$C^{(l)} = \frac{1}{K}n^{\frac{-1}{L+1}}C^{\frac{2L}{L+1}}(C-1)^2\sum_{c=1}^{K}\left[\hat{\nu}_c^{(l)} - Av_{c'}\left\{\hat{\nu}_{c'}^{(l)}\right\}\right]\left[\hat{\nu}_c^{(l)} - Av_{c''}\left\{\hat{\nu}_{c''}^{(l)}\right\}\right]^T,$$

*where $\hat{\nu}_c^{(l)} = \hat{\mu}_c^{(l+1)} \otimes \hat{\mu}_c^{(l)}$. As a consequence, $C^{(l)}$ has rank $K-1$, with a spanning set of nonzero eigenvectors given by*

$$\hat{\nu}_1^{(l)} - \hat{\nu}_c^{(l)}, \quad \text{for } c \in \{2, \ldots, K\}.$$

*Furthermore, all the nonzero eigenvalues are equal and given by*

$$\frac{1}{K}n^{\frac{-1}{L+1}}C^{\frac{2L}{L+1}}(C-1)^2,$$

*where $C$ is the constant referred to in Theorem 3.1.*

### E.2. Backpropagation Errors

The backpropagation errors have been studied by Oymak et al. (2019) and Papyan (2020). Here we consider the extended backpropagation errors, similar to the analysis by Papyan (2020), but adapted to the MSE loss. For a given sample $x_{ic}$, the extended backpropagation error at layer $l$ in our model is defined as

$$\delta_{icc'}^{(l)} = \frac{\partial l(f(x_{ic}; \theta), y_{c'})}{\partial h_{ic}^{(l)}}.$$

We then consider the weighted second moment matrix of the extended backpropagation errors, using the same weights we had in our Hessian matrix, similar to the approach by Papyan

$$\Delta^{(l)} = \sum_{i,c,c'} \frac{1}{Kn} \delta_{icc'}^{(l)} \delta_{icc'}^{(l)T}. \tag{22}$$

Using the fact that

$$\frac{\partial l(f(x_{ic}; \theta), y_{c'})}{\partial h_{ic}^{(l)}} = A_l^T u_{icc'},$$

where $u_{icc'} = W_L...W_1 h_{ic} - y_{c'}$, we obtain

$$\Delta^{(l)} = \frac{1}{Kn} \sum_{i,c,c'} (A_l^T u_{icc'})(A_l^T u_{icc'})^T.$$

Applying the regularization condition in Eq. (17), assuming $d \geq K$, and considering a global optimum, the DNC1 property from Lemma C.1 gives

$$\Delta^{(l)} = \frac{1}{K} \sum_{c,c'} (A_l^T u_{cc'})(A_l^T u_{cc'})^T,$$

where $u_{cc'} = W_L...W_1 \mu_c - y_{c'}$. The DNC2 and DNC3 properties of Lemma C.1 then yield

$$A_l^T u_{cc'} = A_l^T A_l \mu_c^{(l)} - A_l^T y_{c'}$$

$$= \alpha^{(l)2} \|\mu^{(l)}\|_2^2 \mu_c^{(l)} - \alpha^{(l)} \mu_{c'}^{(l)}.$$

Using the expression for $C$ in terms of $\alpha^{(l)}$ and $\|\mu^{(l)}\|_2$ from Eq. (13), we obtain

$$A_l^T u_{cc'} = \alpha^{(l)} \left[ C\mu_c^{(l)} - \mu_{c'}^{(l)} \right].$$

Thus, the extended backpropagated error matrix takes the form

$$\Delta^{(l)} = \frac{\alpha^{(l)2}}{K} \sum_{c,c'} \left[ C\mu_c^{(l)} - \mu_{c'}^{(l)} \right] \left[ C\mu_c^{(l)} - \mu_{c'}^{(l)} \right]^T.$$

Expanding the summation and simplifying gives

$$= \alpha^{(l)2} \left[ (C^2 + 1) \sum_{c=1}^{K} \mu_c^{(l)} \mu_c^{(l)T} - 2CK \mu_G^{(l)} \mu_G^{(l)T} \right].$$

Clearly, the image of $\Delta^{(l)}$ lies in $\text{Span}\{\mu_c^{(l)}\}_{c=1}^{K}$, implying that $\Delta^{(l)}$ has at most rank $K$. Considering the action of $\Delta^{(l)}$ on specific vectors, we find

$$\Delta^{(l)}\big[\mu_c^{(l)} - \mu_G^{(l)}\big] = \alpha^{(l)}C(C^2+1)\big[\mu_c^{(l)} - \mu_G^{(l)}\big], \qquad \Delta^{(l)}\mu_G^{(l)} = \alpha^{(l)}C(C-1)^2\mu_G^{(l)}.$$

Thus, the eigenspace corresponding to the eigenvalue $\alpha^{(l)}C(C^2+1)$ has dimension $K-1$, while the eigenspace corresponding to $\alpha^{(l)}C(C-1)^2$ has dimension 1. Notably, all but one nonzero eigenvalue are equal, with the distinct eigenvalue being smaller, since $C \approx 1$ for well-performing models. Replacing $\alpha^{(l)}$ using Eq. (13), and writing the norm of the feature means in terms of $C$ using Eq. (18), we arrive at the following theorem.

**Theorem E.2.** *Consider the deep linear UFM described in Eq. (4). Let the network width satisfy $d \geq K$, and consider a layer $l$ satisfying $1 \leq l < L$. Additionally, assume that the regularization parameter $\lambda$ satisfies the condition in Eq. (17). Then, at any global optimum of the loss, the extended backpropagation error matrix $\Delta^{(l)}$, defined in Eq. (22), has rank $K$. Moreover, the top eigenspace has dimension $K-1$, with eigenvalue*

$$(C\sqrt{n})^{\frac{2L-2l+2}{L+1}}(C^2+1),$$

*and a set of corresponding spanning eigenvectors given by*

$$\mu_1^{(l)} - \mu_c^{(l)}, \quad \text{for } c \in \{2, \ldots, K\}.$$

*The lower one-dimensional eigenspace has eigenvalue*

$$(C\sqrt{n})^{\frac{2L-2l+2}{L+1}}(C-1)^2,$$

*with corresponding eigenvector $\mu_G^{(l)}$. Here, $C$ is the same constant as referred to in Theorem 3.1.*

Notably, our result appears to differ from Papyan et al. (2020), which reported $K^2$ outliers in the weighted second moment of backpropagation errors. This discrepancy may arise from differences in the choice of loss function.

# F. CE Case Proofs

recall in this case our loss function is

$$\mathcal{L}(H_1, W_1, ..., W_L) = \frac{1}{Kn}g(W_L...W_1H_1, Y) + \frac{1}{2}\lambda\|H_1\|_F^2 + \frac{1}{2}\lambda\sum_{l=1}^{L}\|W_l\|_F^2,$$

where, if we define the matrix $Z = W_L...W_1H_1$, which has columns $z_{ic} = W_L...W_1h_{ic}$, we have

$$g(Z, Y) = -\sum_{i=1}^{n}\sum_{c=1}^{K}\log\left(\frac{\exp((z_{ic})_c)}{\sum_{c'=1}^{K}\exp((z_{ic})_{c'})}\right).$$

Additionally, define $A_{l+1} = W_L...W_{l+1}$ and $H_l = W_{l-1}...W_1H_1$ as before.

## F.1. Proof of Theorem B.2

First note that

$$\frac{\partial Z_{ab}}{\partial(W_l)_{uv}} = (A_{l+1})_{au}(H_l)_{vb},$$

$$\frac{\partial g(Z, Y)}{\partial Z_{ab}} = (P - Y)_{ab}, \quad \text{where} \quad P_{ab} = \frac{\exp(Z_{ab})}{\sum_{a'=1}^{K}\exp(Z_{a'b})}.$$

Taking derivatives of $\mathcal{L}$ with respects to $W_l$ then gives

$$\frac{\partial \mathcal{L}}{\partial (W_l)_{uv}} = \frac{1}{Kn} \frac{\partial Z_{ab}}{\partial (W_l)_{uv}} \frac{\partial g(Z, Y)}{\partial Z_{ab}} + \lambda (W_l)_{uv} \implies$$

$$\frac{\partial \mathcal{L}}{\partial W_l} = \frac{1}{Kn} A_{l+1}^T (P - Y) H_l^T + \lambda W_l.$$

We now drop the regularization term, since again it just translates the eigenvalues and leaves eigenvectors unchanged.

Now also note that

$$\frac{\partial P_{ab}}{\partial Z_{uv}} = \delta_{vb}[\text{diag}(p_b) - p_b p_b^T]_{au},$$

where $p_b$ is the $b^{\text{th}}$ column of $P$, meaning $(p_b)_i = P_{ib}$. We also define the matrix $\rho_b = \text{diag}(p_b) - p_b p_b^T$.
Using this, the second derivative is

$$\frac{\partial^2 \mathcal{L}}{\partial (W_l)_{uv} (W_l)_{xy}} = \frac{1}{Kn} (A_{l+1}^T)_{ua} \frac{\partial P_{ab}}{\partial (W_l)_{xy}} (H_l^T)_{bv}$$

$$= \frac{1}{Kn} (A_{l+1}^T)_{ua} \frac{\partial Z_{rs}}{\partial (W_l)_{xy}} \frac{\partial P_{ab}}{\partial Z_{rs}} (H_l^T)_{bv}.$$

Now being explicit over which variables are summed over:

$$= \frac{1}{Kn} \sum_{abrs} (A_{l+1}^T)_{ua} (A_{l+1})_{rx} (H_l)_{ys} (H_l^T)_{bv} \delta_{sb} (\rho_b)_{ar}.$$

Using our notation of the columns of $H_l$ being $h_b^{(l)}$ in the sense that $(h_b^{(l)})_v = (H_l)_{vb}$, this reduces to

$$\frac{\partial^2 \mathcal{L}}{\partial (W_l)_{uv} (W_l)_{xy}} = \frac{1}{Kn} \sum_b (A_{l+1}^T \rho_b A_{l+1})_{ux} (h_b^{(l)} h_b^{(l)T})_{yv}.$$

Now note that $b$ sums over the number of data points, we can hence choose to replace it with a sum over $i$ and $c$, meaning a sum over the samples, giving

$$\frac{\partial^2 \mathcal{L}}{\partial (W_l)_{uv} (W_l)_{xy}} = \text{Av}_{ic} \left\{ (A_{l+1}^T \rho_{ic} A_{l+1})_{ux} (h_{ic}^{(l)} h_{ic}^{(l)T})_{yv} \right\}.$$

Now flattening the weights as we have done previously gives us that

$$\text{Hess}_l = \text{Av}_{ic} \left\{ \left( A_{l+1}^T \rho_{ic} A_{l+1} \right) \otimes \left( h_{ic}^{(l)} h_{ic}^{(l)T} \right) \right\}.$$

Note this is different from the MSE case, since the left side of the Kronecker product does depend on the considered data-point, and so we cannot just consider the spectrum of each side individually.

We now specialize to our DNC solution. First note using the properties of the definition we have that for $1 \leq l < L$: $A_{l+1} = U_L \tilde{\Sigma}^{L-l} U_l^T$, $H_l = U_{l-1} \bar{\Sigma}^l V_0^T$, and hence $h_{ic}^{(l)} = U_{l-1} \bar{\Sigma}^l V_0^T e_{ic}$, where $e_{ic}$ are the standard basis vectors ordered to match the columns of $H_1$.

Plugging this into our layer-wise Hessian and using properties of the Kronecker product this gives

$$\text{Hess}_l = (U_l \otimes U_{l-1})[\text{Av}_{ic}\{((\tilde{\Sigma}^{L-l})^T U_L^T \rho_{ic} U_L \tilde{\Sigma}^{L-l}) \otimes (\bar{\Sigma}^l V_0^T e_{ic} e_{ic}^T V_0 (\bar{\Sigma}^l)^T)\}](U_l \otimes U_{l-1})^T.$$

We now drop the orthogonal transformation for ease of exposition, and will return to it later. Define the resulting matrix as $\text{Hess}_l'$. We can also seperate the scales from the matrices $\tilde{\Sigma}, \bar{\Sigma}$ since all nonzero singular values are equal, hence write $\bar{\Sigma} = (\alpha\sqrt{n})^{\frac{1}{L+1}}\bar{D}$ and $\tilde{\Sigma} = (\alpha\sqrt{n})^{\frac{1}{L+1}}\tilde{D}$, where $\bar{D} \in \mathbb{R}^{d \times Kn}$, $\tilde{D} \in \mathbb{R}^{K \times d}$ both have their top $K \times K$ block being $\text{diag}(1, 1, ..., 1, 0)$, with all other entries being zero.

This gives

$$\text{Hess}_l' = (\alpha\sqrt{n})^{\frac{2L}{L+1}} \text{Av}_{ic}\{(\tilde{D}^T U_L^T \rho_{ic} U_L \tilde{D}) \otimes (\bar{D} V_0^T e_{ic} e_{ic}^T V_0 \bar{D}^T)\}.$$

We then drop this constant out front for now and will return to it later. Also perform the orthogonal transformation given by the following matrix

$$Q = \begin{bmatrix} U_L & 0_{K \times (d-K)} \\ 0_{(d-K) \times K} & I_{d-K} \end{bmatrix} \otimes \begin{bmatrix} U_L & 0_{K \times (d-K)} \\ 0_{(d-K) \times K} & I_{d-K} \end{bmatrix} \in \mathbf{R}^{d^2 \times d^2}.$$

Call the resulting matrix $\text{Hess}_l''$. This matrix is an average of cross products. From here we shall suppress the matrix dimensions on the identity matrix and zero matrix, which should be clear from context. The left side of the Kronecker product within the average is given by

$$\begin{bmatrix} U_L & 0 \\ 0 & I \end{bmatrix} \tilde{D}^T U_L^T \rho_{ic} U_L \tilde{D} \begin{bmatrix} U_L^T & 0 \\ 0 & I \end{bmatrix}$$

$$= \begin{bmatrix} U_L D U_L^T \rho_{ic} U_L D U_L^T & 0 \\ 0 & 0 \end{bmatrix},$$

where $D = \text{diag}(1, 1, ..., 1, 0) \in \mathbf{R}^{K \times K}$. Note that the standard simplex $S = I_K - \frac{1}{K} 1_K 1_K^T$ can be diagonalized as $S = U_L D U_L^T$, hence the left side of the Kronecker product finally reduces to

$$= \begin{bmatrix} S\rho_{ic}S & 0 \\ 0 & 0 \end{bmatrix}.$$

The right side of the Kronecker product within the average is given by

$$\begin{bmatrix} U_L & 0 \\ 0 & I \end{bmatrix} \bar{D} V_0^T e_{ic} e_{ic}^T V_0 \bar{D}^T \begin{bmatrix} U_L^T & 0 \\ 0 & I \end{bmatrix}$$

$$= \begin{bmatrix} U_L D' V_0^T e_{ic} e_{ic}^T V_0 D' U_L^T & 0 \\ 0 & 0 \end{bmatrix},$$

where $D' \in \mathbf{R}^{K \times Kn}$ is a singular value matrix with the nonzero values being 1 with multiplicity $K - 1$. Now use the fact that $S \otimes 1_n^T = \sqrt{n} \, U_L D' V_0^T$, this reduces to

$$\frac{1}{n} \begin{bmatrix} (S \otimes 1_n^T) e_{ic} e_{ic}^T (S \otimes 1_n^T) & 0 \\ 0 & 0 \end{bmatrix}.$$

The layer-wise Hessian is then

$$\text{Hess}_l'' = \frac{1}{n} \text{Av}_{ic} \left\{ \begin{bmatrix} S\rho_{ic}S & 0 \\ 0 & 0 \end{bmatrix} \otimes \begin{bmatrix} (S \otimes 1_n^T) e_{ic} e_{ic}^T (S \otimes 1_n^T) & 0 \\ 0 & 0 \end{bmatrix} \right\}.$$

Now note that this matrix only has nonzero entries in its top $K^2 \times K^2$ block. If we are interested in the nonzero eigenvalues and eigenvectors we can focus on this top block matrix. We also drop the factor of $1/n$ for now. Call the resulting $K^2 \times K^2$ matrix $\text{Hess}_l'''$.

Now note that, by definition, for a DNC solution we have $\rho_{ic} = \rho_c$, i.e. they are independent of the $i$ index. In addition, if we define $s_{ic} = (S \otimes 1_n^T)e_{ic}$, this is also independent of the $i$ index, so $s_{ic} = s_c$, where $s_c$ are the columns of the standard simplex $S$. This further reduces our Hessian to

$$\text{Hess}_l''' = \text{Av}_c \left\{ S\rho_c S \otimes s_c s_c^T \right\}.$$

We now demonstrate that $\rho_c$ has two terms, one of which is exponentially smaller than the other. We show this for $c = 1$, for simplicity, though it should be clear that this calculation is the same for all $c = 1, ..., K$. First note using the form of $Z$ in the definition of DNC

$$p_1^T = \frac{1}{(K-1) + e^\alpha}(e^\alpha, 1..., 1).$$

Hence

$$\text{diag}(p_1) = \frac{1}{(K-1) + e^\alpha}\text{diag}(e^\alpha, 1, ..., 1),$$

$$p_1 p_1^T = \left(\frac{1}{(K-1) + e^\alpha}\right)^2 \begin{bmatrix} e^{2\alpha} & e^\alpha & e^\alpha & ... & e^\alpha \\ e^\alpha & 1 & 1 & ... & 1 \\ e^\alpha & 1 & 1 & ... & 1 \\ ... & ... & ... & ... & ... \\ e^\alpha & 1 & 1 & ... & 1 \end{bmatrix},$$

and so

$$\rho_1 = \text{diag}(p_1) - p_1 p_1^T$$

$$= \left(\frac{1}{(K-1) + e^\alpha}\right)^2 \begin{bmatrix} (K-1)e^\alpha & -e^\alpha & -e^\alpha & ... & -e^\alpha \\ -e^\alpha & (K-2) + e^\alpha & -1 & ... & -1 \\ -e^\alpha & -1 & (K-2) + e^\alpha & ... & -1 \\ ... & ... & ... & ... & ... \\ -e^\alpha & -1 & -1 & ... & (K-2) + e^\alpha \end{bmatrix}.$$

$$= \frac{e^\alpha}{((K-1) + e^\alpha)^2} \underbrace{\begin{bmatrix} (K-1) & -1 & -1 & ... & -1 \\ -1 & 1 & 0 & ... & 0 \\ -1 & 0 & 1 & ... & 0 \\ ... & ... & ... & ... & ... \\ -1 & 0 & 0 & ... & 1 \end{bmatrix}}_{\rho_1'} + O(e^{-2\alpha}). \tag{23}$$

We refer to this leading order term as $\rho_c'$, and from here drop the higher order term. This is reasonable since $\alpha$ is large when the level of regularization is small. We also drop the constant out front for now and will bring it back with the previously dropped constant later.

We now use Lemma F.1, which states that

$$\rho_c' = K s_c s_c^T + S.$$

Also noting that $Ss_c = s_c$, we get that $S\rho'_c S = \rho'_c$. This further reduces us to the leading order term being

$$\text{Hess}'''_l = \text{Av}_c \left\{ \rho'_c \otimes s_c s_c^T \right\},$$

and hence that

$$\text{Hess}'''_l = \frac{1}{K} \sum_{c=1}^{K} \left( \left[ K s_c s_c^T + \sum_{b=1}^{K} s_b s_b^T \right] \otimes s_c s_c^T \right)$$

$$= \left[ \sum_{c=1}^{K} s_c s_c^T \otimes s_c s_c^T \right] + \frac{1}{K} \left( \sum_{c=1}^{K} s_c s_c^T \right) \otimes \left( \sum_{c=1}^{K} s_c s_c^T \right)$$

$$= \left[ \sum_{c=1}^{K} s_c s_c^T \otimes s_c s_c^T \right] + \frac{1}{K} S \otimes S. \tag{24}$$

Note this matrix has the following eigenvectors and eigenvalues:

- eigenvalue 1 with multiplicity 1 and eigenvector $\sum_c s_c \otimes s_c$.

- eigenvalue $\frac{K-1}{K}$, multiplicity $K-1$, eigenvectors of the form $s_a \otimes s_a - s_b \otimes s_b$, $a \neq b$, $a, b = 1, ..., K$.

- eigenvalue $\frac{1}{K}$, multiplicity $K^2 - 3K + 1$, eigenvectors of the form $s_a \otimes s_b - s_b \otimes s_a$, $a \neq b$, $a, b = 1, ..., K$.

This is the separation of the spectrum into $K$ spikes, with one being larger than the others, and a mini-bulk of size $O(K(K-1))$. The bulk itself represents the remaining zero singular values of the original larger matrix.

We can now reverse the orthogonal transformations and scaling to get back to $\text{Hess}_l$. The combination of the constants that we dropped is equal to

$$\beta = \frac{e^{\alpha}(\alpha\sqrt{n})^{\frac{2L}{L+1}}}{n((K-1) + e^{\alpha})^2}.$$

Reversing the orthogonal transformations, the component from the second of the two matrices in Eq. (24) gives

$$= \frac{1}{K} \beta \left( U_l D U_l^T \otimes U_{l-1} D U_{l-1}^T \right),$$

whilst the first gives

$$= \beta \sum_c U_l \tilde{D}^T U_L^T e_c e_c^T U_L \tilde{D} U_l^T \otimes U_{l-1} \tilde{D}^T U_L^T e_c e_c^T U_L \tilde{D} U_{l-1}^T.$$

Using that $\bar{H}_{l+1} = (\alpha\sqrt{n})^{\frac{l}{L+1}} U_l \tilde{D}^T U_L^T$ and $\bar{H}_{l+1} \bar{H}_{l+1}^T = (\alpha\sqrt{n})^{\frac{2l}{L+1}} U_l D U_l^T$, as well as $\|\bar{H}_{l+1}\|_F^2 = K\|\mu^{(l+1)}\|_2^2$ gives

$$\sum_c \hat{\mu}_c^{(l+1)} \hat{\mu}_c^{(l+1)T} = \frac{K}{K-1} U_l D U_l^T, \quad \hat{\mu}_c^{(l+1)} = \sqrt{\frac{K}{K-1}} U_l \tilde{D}^T U_L^T e_c,$$

Writing $\hat{\nu}_{cc'}^{(l)} = \hat{\mu}_c^{(l+1)} \otimes \hat{\mu}_{c'}^{(l)}$, the leading order term of the layer-wise Hessian is then given by:

$$\text{Hess}_l = \beta \left(\frac{K-1}{K}\right)^2 \left[ \sum_c \hat{\nu}_{cc}^{(l)} \hat{\nu}_{cc}^{(l)T} + \frac{1}{K} \sum_{c,c'} \hat{\nu}_{cc'}^{(l)} \hat{\nu}_{cc'}^{(l)T} \right],$$

Since the class means $\hat{\mu}_c^{(l)}$ satisfy the same dot product relationships as the columns of the matrix $S$, and the eigenvalues are simply scaled by $\beta$, we arrive at the following eigenvalues and eigenvectors:

- Eigenvalue $\beta$ with multiplicity 1 and eigenvector $\sum_c \hat{\nu}_{cc}^{(l)}$.

- Eigenvalue $\frac{K-1}{K}\beta$, multiplicity $K-1$, eigenvectors of the form $\hat{\nu}_{cc}^{(l)} - \hat{\nu}_{c'c'}^{(l)}$, $c \neq c'$, $c, c' = 1, ..., K$.

- Eigenvalue $\frac{1}{K}\beta$, multiplicity $K^2 - 3K + 1$, eigenvectors of the form $\hat{\nu}_{cc'}^{(l)} - \hat{\nu}_{c'c}^{(l)}$, $c \neq c'$, $c, c' = 1, ..., K$.

- the remaining $d^2 - (K-1)^2$ eigenvalues are zero.

### F.2. Supporting Lemmas

The supporting lemma necessary for the proof in App. F is provided here.

**Lemma F.1.** *The quantity $\rho_c'$, defined in Eq. (23), can be written as:*

$$\rho_c' = K s_c s_c^T + S.$$

**Proof:** Note that the entries of $\rho_c'$ are given by

$$(\rho_c')_{ij} = \begin{cases} K-1, & \text{if } i = j = c \\ 1, & \text{if } i = j \neq c \\ -1, & \text{if } i = c, \text{ or } j = c, \text{ but } i \neq j \\ 0, & \text{otherwise} \end{cases}$$

Similarly the entries of $K s_c s_c^T$ and $S$ are given by

$$(K s_c s_c^T)_{ij} = \begin{cases} \frac{(K-1)^2}{K}, & \text{if } i = j = c \\ -\frac{K-1}{K}, & \text{if } i = c, \text{ or } j = c, \text{ but } i \neq j \\ \frac{1}{K}, & \text{otherwise} \end{cases}$$

$$S_{ij} = \begin{cases} \frac{K-1}{K}, & \text{if } i = j \\ -\frac{1}{K}, & \text{otherwise} \end{cases}$$

Looking at this case by case, it is clear that $\rho_c' = K s_c s_c^T + S$.

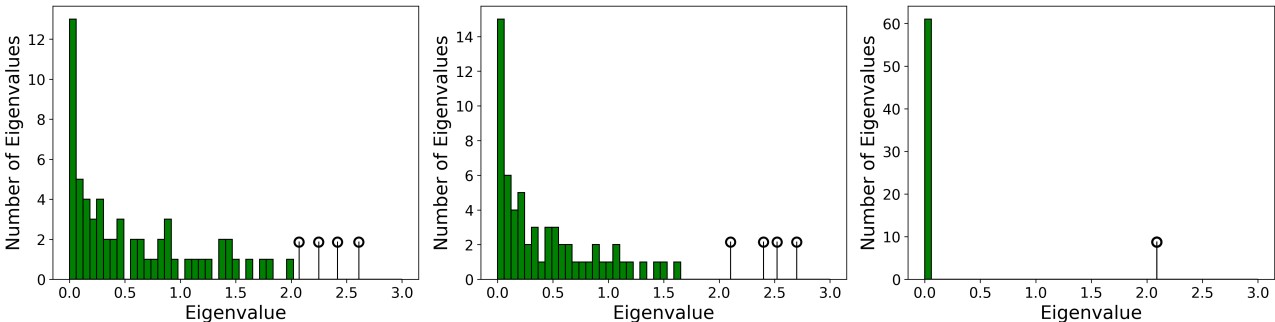

*Figure 8.* Histogram of the eigenvalues of $W_l^T W_l$ for the deep linear UFM at an intermediate layer for epochs 0,2500 and 50,000. The top $K = 4$ outliers are plotted separately as spikes.

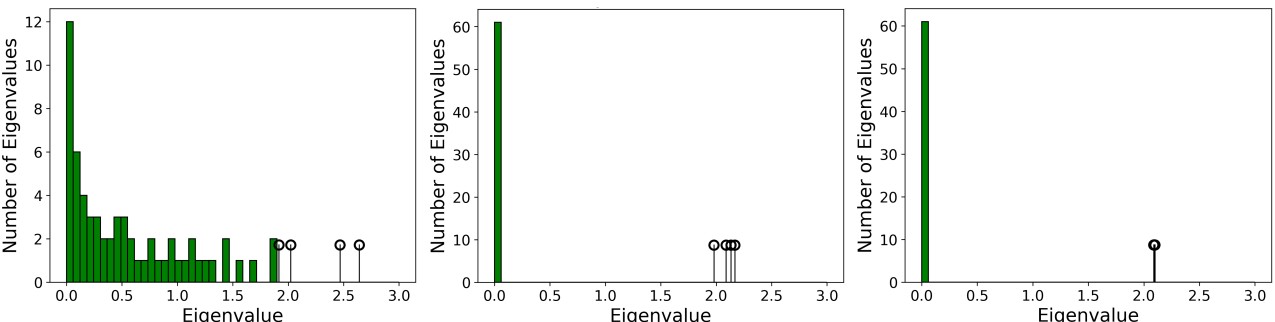

*Figure 9.* Histogram of the eigenvalues of $W_l^T W_l$ for the deep ReLU UFM at an intermediate layer, for epochs $0$, $4 \times 10^5$ and $3 \times 10^6$. The top $K = 4$ outliers are plotted separately as spikes.

## G. Further Numerical Experiments

In this section, we provide additional details about the experiments in the main text, as well as further experiments involving the weight matrices and Papyan's decomposition. We also present further experiments on standard DNNs applied to canonical datasets.

### G.1. Further Experiments in the Deep UFM

**More Details about the Main Text Experiments:** For the main text experiments we trained a deep linear UFM using gradient descent with hyperparameters $d = 65$, $L = 4$, $K = 4$, $n = 40$, and $\lambda = 5 \times 10^{-5}$, focusing on layer $l = 3$. We use the same hyperparameters for both the linear and ReLU cases. We stress that the choice $l = 3$ is arbitrary, all layers show the exact same phenomenology.

We next elaborate on the metrics used for the figures in Section 5 of the main text.

To measure the extent to which $\{\mu_c^{(l+1)} \otimes \mu_{c'}^{(l)}\}_{c,c'=1}^{K}$ form eigenvectors of $\text{Hess}_l$, as shown in Figures 2 and 5, we used the cosine similarity squared between $\mu_c^{(l+1)} \otimes \mu_{c'}^{(l)}$ and $\text{Hess}_l(\mu_c^{(l+1)} \otimes \mu_{c'}^{(l)})$, given by

$$f_{cc'} = \frac{|(\mu_c^{(l+1)} \otimes \mu_{c'}^{(l)})^T \text{Hess}_l(\mu_c^{(l+1)} \otimes \mu_{c'}^{(l)})|^2}{\|(\mu_c^{(l+1)} \otimes \mu_{c'}^{(l)})\|_2^2 \|\text{Hess}_l(\mu_c^{(l+1)} \otimes \mu_{c'}^{(l)})\|_2^2}.$$

This metric equals one precisely when $\mu_c^{(l+1)} \otimes \mu_{c'}^{(l)}$ is an eigenvector of $\text{Hess}_l$.

To analyze the decomposition of the gradient $\tilde{g}^{(l)}$ in the natural basis $\{\mu_c^{(l+1)} \otimes \mu_{c'}^{(l)}\}_{c,c'=1}^{K}$, also shown in Figures 2 and 5, we used the cosine similarity squared between $\tilde{g}^{(l)}$ and $\mu_c^{(l+1)} \otimes \mu_{c'}^{(l)}$, given by

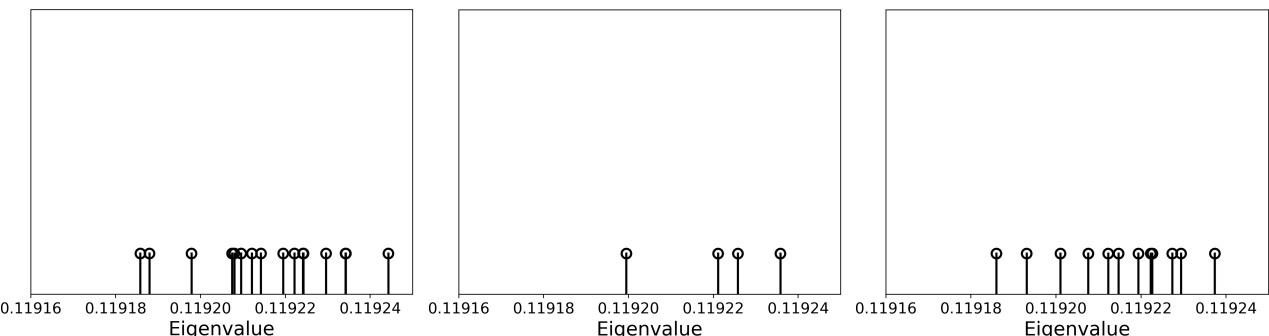

*Figure 10.* Plot of the top $K^2 = 16$ eigenvalues of $\text{Hess}_l$ for the deep linear UFM after 50,000 epochs, together with knockouts of each component from the decomposition in Eq. (5). **Left:** Outliers of $\text{Hess}_l$. **Middle:** Outliers of $\text{Hess}_l - G_{\text{cross}}$. **Right:** Outliers of $\text{Hess}_l - G_{\text{class}}$.

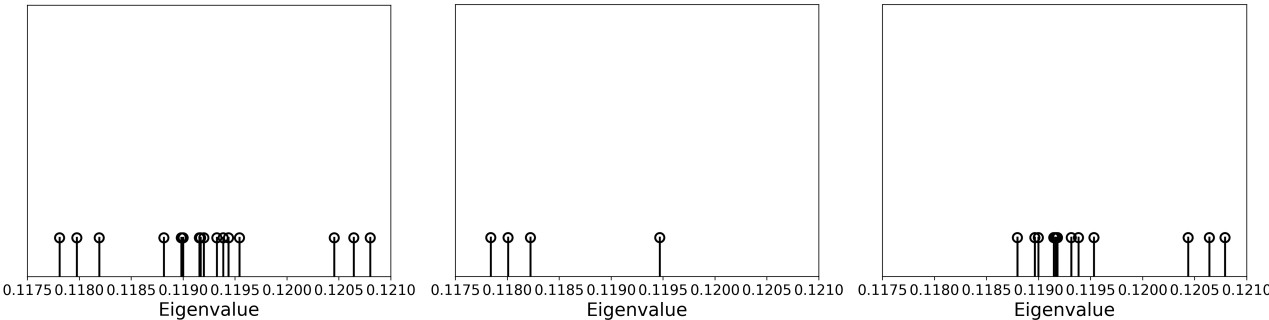

*Figure 11.* Plot of the top $K^2 = 16$ eigenvalues of $\text{Hess}_l$ for the deep ReLU UFM after $10^6$ epochs, together with knockouts of each component from the decomposition in Eq. (5). **left:** Outliers of $\text{Hess}_l$. **Middle:** Outliers of $\text{Hess}_l - G_{\text{cross}}$. **Right:** Outliers of $\text{Hess}_l - G_{\text{class}}$.

$$\tilde{f}_{cc'} = \frac{|(\mu_c^{(l+1)} \otimes \mu_{c'}^{(l)})^T \tilde{g}^{(l)}|^2}{\|\mu_c^{(l+1)} \otimes \mu_{c'}^{(l)}\|_2^2 \|\tilde{g}^{(l)}\|_2^2}.$$

When the vectors $\mu_c^{(l+1)} \otimes \mu_{c'}^{(l)}$ form an orthogonal basis, these coefficients sum to 1.

To assess the extent to which the matrix $\bar{H}_l^T \bar{H}_l$ forms an orthogonal frame, as shown in Figure 3, we considered the Frobenius distance after normalization between $\bar{H}_l^T \bar{H}_l$ and $I$, defined as

$$M_l = \left\| \frac{\bar{H}_l^T \bar{H}_l}{\|\bar{H}_l^T \bar{H}_l\|_F} - \frac{I_K}{\sqrt{K}} \right\|_F.$$

This metric is non-negative, taking the value zero precisely when $\bar{H}_l^T \bar{H}_l$ forms an orthogonal frame.

**More Experiments in deep UFMs:**

We now provide experimental evidence for the remaining theorems in the main text. We begin with plots of the spectrum of the Gram matrix $W_l^T W_l$ at various points in training, corresponding to the squared singular values. These spectra are shown in Figure 8 for the linear case and in Figure 9 for the ReLU case. As with the Hessian, the top eigenvalues initially blend into the bulk but progressively separate during training, eventually converging uniformly to an identical value. Simultaneously, the bulk eigenvalues collapse to a single atom at zero, in agreement with Theorem 3.4.

To investigate the decomposition results described in Theorem 3.2, we display the Hessian's outlier eigenvalues after training, along with their behavior when each component of the decomposition from Eq. (5) is individually removed. These results are shown in Figure 10 for the linear case and Figure 11 for the ReLU case. We omit the case where $G_{\text{within}}$ is subtracted,

since the spectrum is identical to that of the full Hessian, as predicted by our theory. We observe that removing $G_{l,\text{cross}}$ and $G_{l,\text{class}}$ eliminates exactly $K(K-1) = 12$ and $K = 4$ outlier eigenvalues, respectively, as predicted by Theorem 3.2.

Finally, we examine the eigenvalues of the feature Gram matrix $\bar{H}_l^T \bar{H}_l$ in the ReLU case, shown in Figure 12. In Section 4, we claimed that the network quickly develops non-negative entries and forms an orthogonal frame with an additional rank-one perturbation to enforce non-negativity. This spike then decays over the course of training. This can be seen in the figure: at intermediate times there is one dominant spike relative to the others, but over a longer time horizon it converges to the same value as the remaining spikes.

### G.2. Full Networks with UFM-Style Regularization

Here we provide further experimental details and evaluations of how the theorems in the main text manifest in DNNs trained on the MNIST (Lecun et al., 1998) and CIFAR-10 (Krizhevsky, 2009) datasets. For the MNIST experiments, we subsample 5,000 examples per class to match the class balance of CIFAR-10. Input data is preprocessed by subtracting the mean and dividing by the standard deviation. For the experiments in Figure 6, and the further experiments in this section, we use the ResNet-20 architecture as the feature map $h(x; \bar{\theta})$, followed by four linear layers of width $d = 64$. We additionally use UFM-style regularization, regularizing the outputs of the feature map and the layers in the fully connected head. Also, to stay coupled to theory, we use MSE loss. The regularization parameter is set to $\lambda = 5 \times 10^{-4}$, except for the feature layer, where it is set to $\lambda_H = 1 \times 10^{-7}$. This lower value accounts for the impact of the number of data points on the overall regularization strength. As in the UFM experiments, we focus on the layer $l = 3$ (though again all layers show the same phenomenology). For the experiments of Figure 7, we use standard regularization with parameter $\lambda = 5 \times 10^{-3}$ and ReLU activations in the fully connected head, but otherwise the training setup is the same.

For MNIST, we train for 4,000 epochs, starting with a learning rate of 0.04, which is halved after 2,000 epochs. For CIFAR-10, we train for 5,000 epochs, starting with a learning rate of 0.05, halved after 2,500 epochs. We use batch gradient descent with a batch size of 10,000 to approximate full gradient descent, consistent with the model. We present detailed results from individual runs, noting that the conclusions are robust provided the regularization is not so large as to enforce the trivial zero solution.

Figures 13 and 15 show the eigenvalues of $\text{Hess}_l$, together with the knockouts of Papyan's decomposition described in Eq. (5). The $G_{l,\text{within}}$ case is omitted, as its spectrum is identical to that of $\text{Hess}_l$. We observe that while the predicted number of eigenvalues associated with each component is preserved, exact equality does not hold. This discrepancy arises because the unconstrained feature assumption is only approximate in practice, introducing noise. We also show the weights in Figures 14 and 16, with the same conclusions as in the model results presented in App. G.1.

Across all plots, CIFAR-10 displays greater noise and weaker convergence compared to MNIST. This is intuitive: CIFAR-10 is a more complex dataset, requiring more sophisticated DNNs to achieve the same level of overparameterization. Consequently, the unconstrained feature assumption holds less closely, introducing additional noise into the results.

In Figure 17, we show how, for the MNIST network, the predicted left singular vectors $\hat{\mu}_c^{(l+1)}$ and predicted right singular vectors $\hat{\mu}_c^{(l)}$, for $c = 1, \ldots, K$, align with the true singular vectors. This alignment is quantified using the cosine similarity computed before and after applying the appropriate Gram matrix of the weights. We observe that our theoretical predictions for the singular vectors emerges rapidly during training.

The effective rank, denoted $f_{\text{ER}}$, of a matrix $M$ with singular values $\sigma_1, ..., \sigma_m$, is given by

$$f_{\text{ER}} = \exp\left( -\sum_{i=1}^{m} \frac{\sigma_i}{\sum_j \sigma_j} \log\left( \frac{\sigma_i}{\sum_j \sigma_j} \right) \right), \tag{25}$$

with the convention that $0 \log(0) = 0$. In Figure 18, we show the effective ranks of the layer-wise Hessian and weight matrix throughout training. In both cases, we see that the effective rank converges to the value predicted by our theory for the true rank. This indicates that both matrices exhibit the correct set of large outliers together with very small bulk values. The weight matrix approach its convergence value from above, whereas the Hessian approaches its convergence value from below. This behavior is due to the initialization and architecture, rather than any specific aspect of our theory, and understanding these effects represent an interesting direction for future work.

We also plot the mean and standard deviation of the outlier eigenvalues of the weights and the Hessian for the MNIST

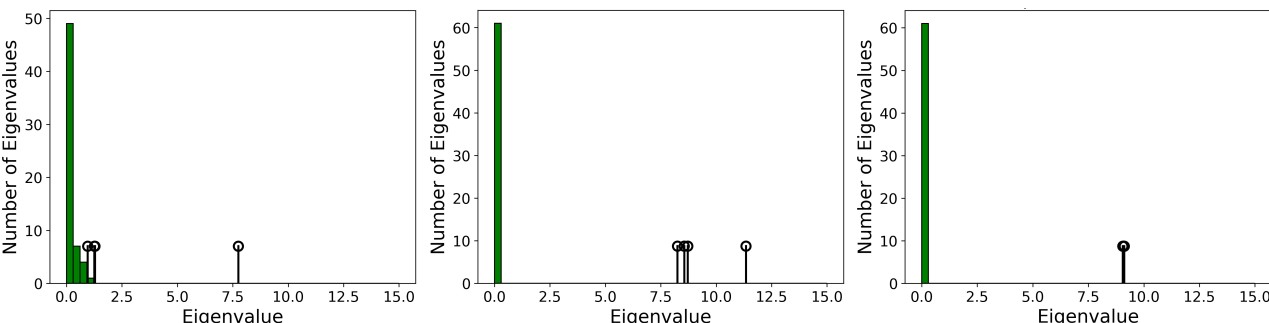

*Figure 12.* Histogram of the eigenvalues of $\bar{H}_l^T \bar{H}_l$ for the deep ReLU UFM at an intermediate layer, shown for epochs $0$, $4 \times 10^5$ and $3 \times 10^6$. The top $K$ outliers are plotted separately as spikes.

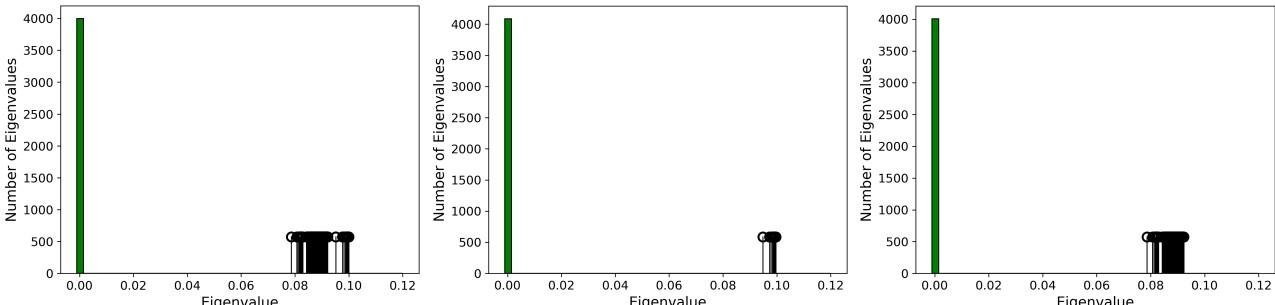

*Figure 13.* Training of a DNN applied to MNIST after 4000 epochs. **Left:** Histogram of the eigenvalues of $\mathrm{Hess}_l$, with the top $K^2 = 100$ plotted as spikes. **Middle:** eigenvalues of $\mathrm{Hess}_l - G_{l,\mathrm{cross}}$, with the top $K = 10$ plotted as spikes. **Right:** eigenvalues of $\mathrm{Hess}_l - G_{l,\mathrm{class}}$, with the top $K(K-1) = 90$ plotted as spikes.

network in Figure 19. In both cases, we again see that they converge to a single value with very little variability in their distribution.

Lastly, in Figure 20 we plot the equivalent of the top part of the main text Figure 6 but for CIFAR-10, with the same qualitative conclusion as observed in the MNIST case.

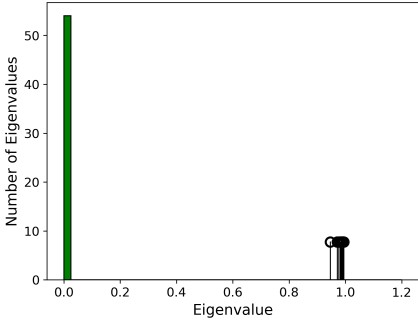

*Figure 14.* Histogram of the eigenvalues of $W_l^T W_l$ for a DNN applied to MNIST at an intermediate layer after 4000 epochs. The top $K$ outliers are plotted separately as spikes.

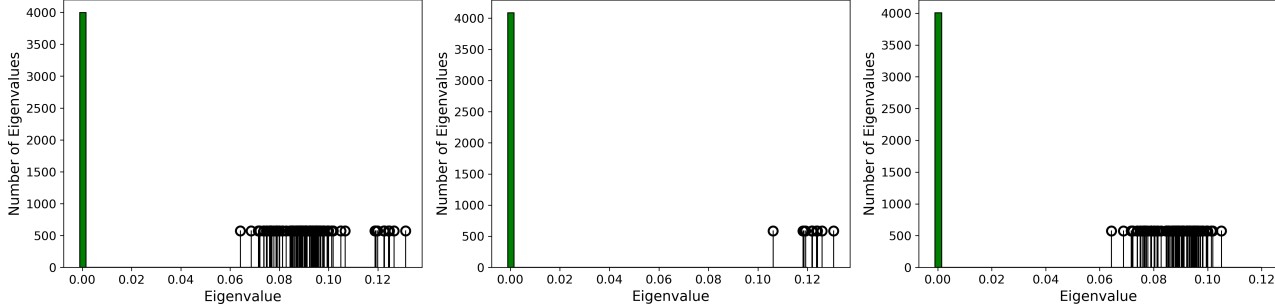

*Figure 15.* Training of a DNN applied to CIFAR-10 after 5000 epochs. **Left:** Histogram of the eigenvalues of $\text{Hess}_l$, with the top $K^2 = 100$ plotted as spikes. **Middle:** eigenvalues of $\text{Hess}_l - G_{l,\text{cross}}$, with the top $K = 10$ plotted as spikes. **Right:** eigenvalues of $\text{Hess}_l - G_{l,\text{class}}$, with the top $K(K-1) = 90$ plotted as spikes.

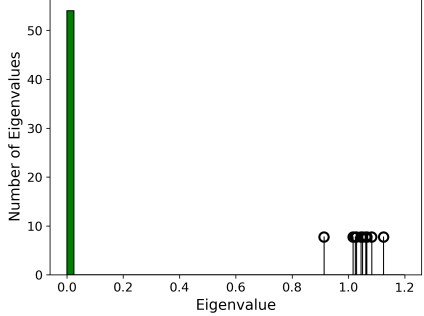

*Figure 16.* Histogram of the eigenvalues of $W_l^T W_l$ for a DNN applied to CIFAR-10 at an intermediate layer after 5000 epochs. The top $K$ outliers are plotted separately as spikes.

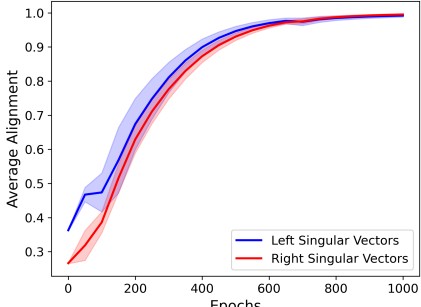

*Figure 17.* Extent to which the predicted left singular vectors $\hat{\mu}_c^{(l+1)}$ and predicted right singular vectors $\hat{\mu}_c^{(l)}$ align with the true singular vectors of the weight matrix $W_l$ for a DNN at an intermediate separated layer $l$ trained on MNIST. Alignment is measured using the cosine similarity before and after applying the appropriate Gram matrix. In both cases, the cosine similarity averaged over $c = 1, \ldots, K$ is shown, with one–standard-deviation error bars.

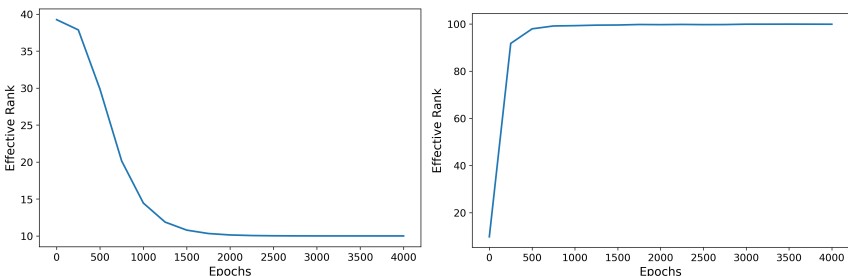

*Figure 18.* Plots of the effective ranks, given in Eq. (25), of the weight matrix (left) and the layer-wise Hessian (right) throughout training for an intermediate separated layer of a DNN trained on MNIST.

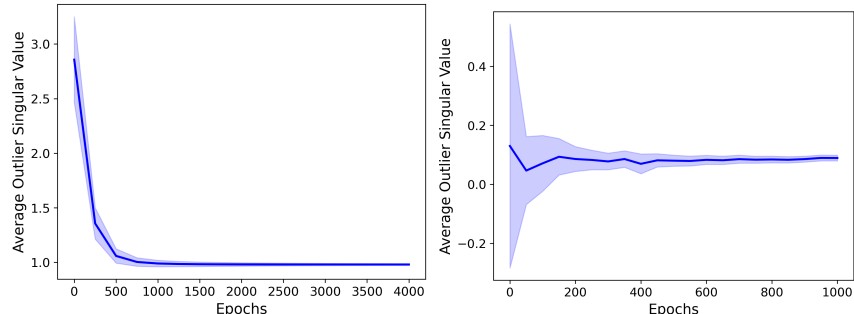

*Figure 19.* Outlier singular values of the weight matrix (left) and layer-wise Hessian (right) for an intermediate layer of a DNN trained on MNIST. For the weights, these correspond to the top $K$ singular values; for the Hessian, the top $K^2$ eigenvalues. In both cases, the average outlier value is shown, with one-standard-deviation error bars.

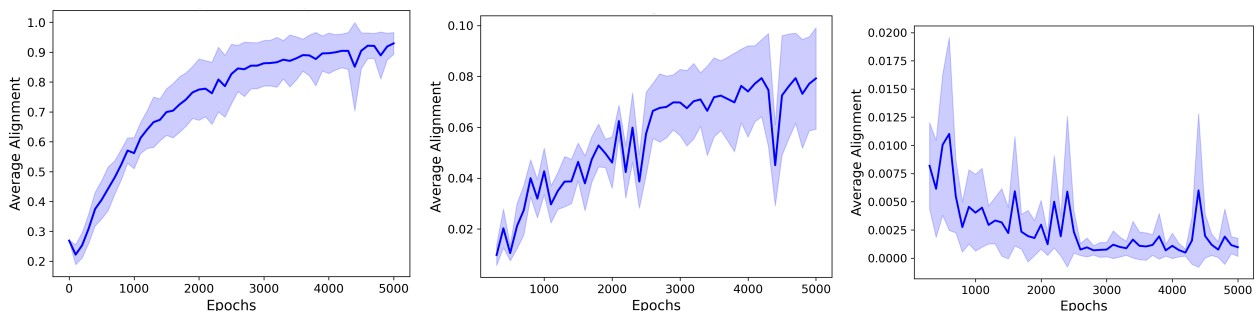

*Figure 20.* Training dynamics of a DNN at an intermediate separated layer $l$ on the CIFAR-10 dataset. Each plot shows quantities averaged over $c, c'$ with one–standard-deviation error bars. **Left**: Predicted eigenvector alignment, measured by the squared cosine similarity between $\mu_c^{(l+1)} \otimes \mu_{c'}^{(l)}$ and $\text{Hess}_l(\mu_c^{(l+1)} \otimes \mu_{c'}^{(l)})$. **Middle & Right**: Gradient alignment, measured by the decomposition coefficients of $\tilde{g}^{(l)}$ in the basis of predicted eigenvectors $\mu_c^{(l+1)} \otimes \mu_{c'}^{(l)}$. Middle: $c = c'$, right: $c \neq c'$.

