# OpenReview forum: "Unifying Low Dimensional Spectra in Deep Learning"
_ICML.cc/2026/Conference — ICML 2026 regular_

### Official Review · Reviewer_QDxo · 2026-03-12

**Soundness:** 3
**Presentation:** 3
**Significance:** 3
**Originality:** 3
**Overall Recommendation:** 5
**Confidence:** 4

**Summary:**

This work uses the Unconstrained Feature Model (UFM) to provide a theoretical explanation for various low-rank structures commonly found in Hessian, gradient, and weight matrices of deep neural networks via the (deep) neural collapse (DNC or NC). The authors show that the eigenvalues and eigenvectors can be fully characterized by the feature means. They begin with a linear model, extend to a ReLU case, and then empirically show that the theory also holds in a realistic feature map, i.e., ResNet.

**Compliance With Llm Reviewing Policy:**

Affirmed.

**Final Justification:**

The authors resolved most of my concerns, which helped with my understanding of their work better. Due to that, I am raising my confidence score, but keeping the recommendation score.

**Key Questions For Authors:**

Please see the questions **[Q1]**, **[Q2]**, and **[Q3]**, in the weaknesses  **W1**, **W2**, and **W3**, respectively.

**Limitations:**

Yes

**Strengths And Weaknesses:**

**Strengths**:

**S1:** As far as I can tell, the theoretical results are sound and rigorous.

**S2:** The paper is very well-written. The introduction clearly states the gap in the literature and the authors' contributions. As a reader without a background in UFM, I found Appendix A particularly helpful.

**S3:** I find this work theoretically significant. It is satisfying to know a clear analytical expression of the Hessian matrix (i.e., the eigen-decomposition of it) in terms of interpretable terms. I understand UFM is an idealized setup/approximation, but what is gained from this is worth the tradeoff.

**S4:** The theoretical contributions are original.

**Weaknesses**:

**W1:**  It would have been nice to see some theoretical predictions explicitly compared against experimental results. For example, (1) the theoretical value of non-zero eigenvalues in Theorem 3.1 could have been marked in Figure 1, and (2) the gradient alignment prediction, 1/K, could have been shown as a horizontal line in Figure 2 Middle. [**Q1**] Could the authors show the validations/comparisons?
 If there are agreements between the theory and experiment, I would bolster the validity of the overall theoretical framework. If there is a disagreement, it would be useful to know where the gap comes from. If the authors have justifiable reasons for not showing the comparison, it would be helpful for me, as a reviewer, to know what they are.

**W2:** Although the theoretical significance of Theorem 3.1 is high, I am not entirely sure what exactly the practical significance is. The last paragraph of Section 6, Concluding Remarks, attempts to address this, but it is not very convincing. [**Q2**] Wouldn't targeting DNC via regularization alter the relationship between DNC and Hessian, rendering Theorem 3.1., the very logical justification for targeting DNC instead of Hessian, useless, since it would alter the objective (Eqn 4)? Something here feels circular to me.

**W3:** Section 5, Numerical Experiments, could have better served its purpose by considering a more realistic setup. I find the ResNet + 4 linear-layer configuration to be a bit far from a realistic setup. [**Q3**] Have the authors tested the ResNet + ReLU layers setup?

---

> ### Author Rebuttal · Authors · 2026-03-30
>
> We thank the reviewer for the thoughtful feedback and constructive suggestions. Below, we address the main concerns and outline the specific changes we plan to make in the revised version of the paper.
>
> **It would have been nice to see some theoretical predictions explicitly compared against experimental results … the theoretical value of non-zero eigenvalues in Theorem 3.1 could have been marked in Figure 1, and the gradient alignment prediction, 1/K, could have been shown as a horizontal line in Figure 2 … Could the authors show the validations/comparisons?**
>
> Thank you for this helpful suggestion. We agree that explicitly overlaying the theoretical predictions on the plots would strengthen the presentation and make the empirical validation of the theory clearer. In the revision, we will add these theoretical predictions explicitly to all relevant plots.
>
> For clarity, in all of our model experiments we observe essentially exact agreement with the theory, up to minor numerical roundoff. Even in cases where this is not visually obvious from the current plots, such as Figure 2, the apparent discrepancy arises because we display only a limited number of epochs in order to highlight the early-stage evolution of the quantities from no alignment to strong alignment. To make this agreement more transparent, we will also report the final values at the end of training in the figure captions.
>
> **Although the theoretical significance of Theorem 3.1 is high, I am not entirely sure what exactly the practical significance is. The last paragraph of Section 6, Concluding Remarks, attempts to address this, but it is not very convincing. Wouldn't targeting DNC via regularization alter the relationship between DNC and Hessian, rendering Theorem 3.1., the very logical justification for targeting DNC instead of Hessian, useless, since it would alter the objective (Eqn 4)?**
>
> Thank you for raising this point. To answer your question, it is helpful to distinguish between two separate observations arising from the UFM:
>
> 1. DNC occurs as a consequence of optimization dynamics (as shown by Dang et al.).
> 2. DNC gives rise to the structures observed in deep learning spectra (our results).
>
> We view our work as showing that if DNC is present, then the other spectral structures follow as well. Consequently, we believe that altering the dynamics to promote DNC would not compromise the relationship between DNC and the other spectral phenomena. Rather, it would act on the first point above, potentially in a beneficial direction. That said, we have not yet investigated in detail how regularization designed to target DNC would behave in practice, and we would be happy either to remove this comment or to make its conjectural nature more explicit in the revision.
>
> Regarding the broader question of practical significance, we believe a comment from the introduction may be more compelling and better justified. Recent work has argued that the effect of flatness on generalization can be measured using only the final-layer Hessian (Han et al., 2025; Walter et al., 2025), and that layer-wise Hessians can serve as a useful diagnostic tool (Bolshim & Kugaevskikh, 2025). Our work provides motivation for further study using UFMs to investigate how architectural decisions affect layer-wise Hessians.
>
> Overall though, the goal of the present work is to provide theoretical significance through explanations of phenomenology, We leave it to future work to evaluate whether and how these insights can be leveraged for system design.
>
> **Section 5, Numerical Experiments, could have better served its purpose by considering a more realistic setup. I find the ResNet + 4 linear-layer configuration to be a bit far from a realistic setup. [Q3] Have the authors tested the ResNet + ReLU layers setup?**
>
> We agree that the experiments should move one step closer to a more realistic setting. In the revision, we will add a full set of ResNet + ReLU experiments. In preliminary versions of these experiments, we observe behaviour very similar to that of the ReLU UFM: one spike in the features becomes larger to ensure non-negativity, leading to a Hessian spectrum resembling the middle panel of Figure 4. We will aim to provide preliminary plots and metric values in a follow-up response as additional evidence that this is indeed the case.
>
> To provide further evidence that the UFM should capture such phenomena, we point to the experiments by Sukenik et al. (2024), where the ReLU UFM and the ResNet+ReLU experiments have exact agreement.
>
> We thank the reviewer again for their valuable feedback and for helping improve the clarity and impact of our paper.

---

> > ### Author Rebuttal · Reviewer_QDxo · 2026-04-03
> >
> > Thank you for the thoughtful answers. I have read through both your response and the paper again, and I think I understand the paper and its contribution much better. I will adjust my confidence score accordingly.

---

### Official Review · Reviewer_FTM4 · 2026-03-14

**Soundness:** 2
**Presentation:** 3
**Significance:** 3
**Originality:** 2
**Overall Recommendation:** 4
**Confidence:** 4

**Summary:**

This paper aims to provide a theoretical explanation for the low-dimensional Hessian eigenspectrum observed in neural networks. While this phenomenon has been widely reported empirically, there has been limited theoretical understanding of its origin. The paper argues that deep neural collapse (DNC) is a key mechanism behind this structure, and develops supporting analysis under the Deep Unconstrained Feature Model (UFM). In this setting, the authors derive structured results for the Hessian spectrum, gradient, and weights, and also discuss extensions beyond the basic MSE linear case.

**Compliance With Llm Reviewing Policy:**

Affirmed.

**Final Justification:**

The authors have addressed several of the concerns raised in my initial review, including providing additional experimental results using more realistic ResNet architectures. While the experimental setup remains somewhat removed from practical settings (e.g., training for 8,000 epochs), the theoretical claims are now supported by empirical observations in standard architectures, which strengthens their credibility. Accordingly, I increase my score to 4, but no further.

**Key Questions For Authors:**

1. In the left panel of Figure 2, why does the cosine similarity not reach 1? Is this due to optimization error, or a gap between the theoretical prediction and the experimental setting?
2. All of the experiments seem to be reported for $l=3$. Do similar spectral and alignment patterns also hold for other layers? If not, what makes layer $l=3$ special?
3. As discussed in Weakness 5, do similar alignment phenomena arise in more practical neural network settings as well? If not, could the authors provide some justification for why the theory is still informative for practical networks?

**Limitations:**

The paper does not explicitly discuss its limitations. In my view, several important limitations should be acknowledged, such as the restrictive UFM-based modeling assumptions, the focus on MSE loss in the main theory, and the dependence on explicit regularization.

**Strengths And Weaknesses:**

### **Strengths**

1. The paper is clearly written and well organized. The motivation, theorem statements, and empirical sections are arranged in a way that makes the main narrative relatively easy to follow, despite the technical nature of the subject.
2. The paper addresses an important and timely problem. The low-dimensional structure of the Hessian spectrum has been widely observed for many years, and a number of empirical studies have suggested a close connection between this structure and the number of classes. However, there has been very limited theoretical understanding of this phenomenon. This question is particularly important because prior work has also observed that the gradient is largely concentrated in the top eigenspace of the Hessian, suggesting that this low-dimensional structure may play a meaningful role in optimization. The paper makes a valuable contribution by arguing that this phenomenon can be explained through deep neural collapse (DNC), and by developing supporting theory under the unconstrained feature model (UFM).

### **Weaknesses**

1. The paper develops its analysis entirely within the Deep Unconstrained Feature Model (UFM). While I appreciate the technical difficulty of analyzing more realistic architectures, **this modeling choice seems quite restrictive.** In particular, the training data are effectively represented through a learnable feature map $h_{ic}$, which abstracts away much of the structure of the actual data distribution. As a result, it is unclear to what extent the derived spectral phenomena reflect properties of practical neural networks, as opposed to consequences of this highly simplified setting.

    The gap is further amplified by additional assumptions in the model. Although the problem under consideration is classification, the main objective is based on MSE loss rather than the more standard CE loss. In addition, regularization is imposed not only on the classifier weights $W$ but also on the feature map $h_{ic}$ itself, which departs from standard practice. However, except for the discussion of the ReLU case, the paper does not provide much explicit interpretation of what role $\lambda$ is playing in the theory or which aspects of the results critically depend on this regularization structure.

2. The main theorems analyze the Hessian under the assumption that DNC has already formed, which appears to correspond to a converged regime where the loss is close to its minimum. However, **in practice, the separation between the top-$K$ eigenvalues and the rest often emerges much earlier, well before the loss has fully converged** (e.g. Figure 2 in [1]). This suggests that DNC alone may not be sufficient to explain the observed low-dimensional Hessian structure throughout training. In particular, it raises the question of whether there are additional  underlying factors, beyond DNC itself, that help induce or stabilize this spectral structure during optimization. Clarifying this point would strengthen the paper, since the current theory seems more tailored to explaining the final converged regime than the earlier training dynamics where the phenomenon is already visible.
3. The paper would be easier to follow if it included a brief background paragraph on prior UFM results. In particular, the phrase “DNC is optimal” in line 137 is too vague as written. It would help to clarify the precise statement being used, for example whether the intended claim is that, “under a minimum-loss assumption, DNC in Definition 2.1 holds, as suggested by Lemma C.1 and C.2”. Making this dependency explicit would also clarify how the later theorems build on prior work and on the loss-related assumption.
4. There appears to be a notable gap between the theory developed in the paper and the spectral phenomena reported in prior empirical studies. In particular, Figure 2 of [1] suggests that the number of dominant Hessian outliers is approximately $K$, the number of classes. By contrast, the main theoretical result, Theorem 3.1, states that in the MSE setting the top $K^2$ eigenvalues are equal and all remaining eigenvalues are zero. This predicts a $K^2$-dimensional outlier subspace, which does not align with the commonly reported picture of $K$-dominant outliers. Appendix B partially addresses this issue by analyzing the cross-entropy case. There, the spectrum is more structured: the theory predicts $1+(K-1)$ leading outliers, which is consistent with having roughly $K$ dominant directions, together with an additional $K^2-3K+1$ mini-bulk component. This is a meaningful step toward reconciling the theory with practice. However, even this analysis still seems incomplete.

    First, as the authors acknowledge, the CE analysis has important limitations, most notably that DNC is no longer a global optimum under CE loss. Second, the predicted mini-bulk structure still does not fully match prior empirical observations. For example, [2] reports $K^2-K$ mini-bulk outliers, which differs quantitatively from the decomposition derived in Appendix B. Overall, while the paper offers several insightful idealized analyses, **it does not yet seem to reproduce the empirically observed Hessian spectrum in any single practically relevant setting**. As a result, the connection between the theory and the spectral structure observed in real neural networks remains somewhat unresolved.

5. While the additional experiments in Figure 6 and Appendix G.2 are interesting, they do not yet fully resolve the gap between the theoretical analysis and practical neural network behavior. If the paper aims to claim that the proposed theory explains phenomena observed in real neural networks, then **the empirical setup should more closely follow standard practical settings**. At present, this does not seem to be the case.
    1. First, the architecture described in Appendix G.2 is not representative of common practice. According to the experimental details, the model uses ResNet-20 as a feature map $h$, followed by four linear head layers. However, standard ResNet architectures typically use only a single linear classifier head. This raises the question of whether the observed behavior is specific to this modified architecture rather than reflective of practical networks more broadly.
    2. Second, the paper does not report results for the case where this multi-layer classifier head includes ReLU nonlinearities. This is particularly relevant because Section 4 discusses the ReLU UFM setting. If that part of the theory is meant to support the practical experiments, then the paper should also verify whether the same spectral and alignment phenomena hold in practical networks with ReLU-based classifier heads.
    3. Third, although the loss is not stated explicitly, it appears that the experiments likely use MSE loss, together with explicit layer-wise regularization as in the UFM analysis. Moreover, according to Appendix G.2, the regularization strengths applied to the feature map and the linear layers are different, which introduces an additional mismatch with the theoretical model. This makes it harder to assess whether the experimental observations genuinely validate the theory or instead arise from a specially designed setting.
    4. Finally, the results are shown only for layer $l=3$. Since the paper makes broader claims about the theoretical mechanism, it would be important to verify whether similar eigenvalue distributions and alignment patterns hold across all layers, rather than at a single selected layer.

    Overall, I think the current experiments are better interpreted as controlled supporting evidence than as convincing validation in realistic neural network settings. To more strongly support the claim that the theory explains practical phenomena, the paper should include experiments under a more standard setup, such as the original ResNet architecture with cross-entropy loss, and examine the spectral distribution and alignment behavior across multiple layers. As it stands, there still appears to be a noticeable gap between the theory and practical relevance.


**Minor Corrections**

1. In the rightmost plot of Figure 1, the 16 spikes appear to overlap, making the structure difficult to interpret visually. A zoomed-in view of the relevant region might improve clarity.
2. Line 136: “he” should be changed to “He”.
3. The indentation in Lemma C.1 may need to be cleaned up.
4. In the right column around lines 134 to 135, the definitions of $1_n$ and $\otimes$ would be better placed before Definition 2.1, since these notations are already used there.
5. When citing Figure 2, such as in the right column around lines 193 and 252, it would improve clarity to indicate the specific subplot being referenced. The former seems to refer to the left panel, while the latter appears to refer to the middle and right panels.
6. Figure 6 caption: “training” should be changed to “Training”.

---

**References**

1. Does SGD really Happen in Tiny Subspaces?, ICLR 2025.
2. Traces of Class/Cross-Class Structure Pervade Deep Learning Spectra, JMLR 2020.

---

> ### Author Rebuttal · Authors · 2026-03-31
>
> We thank the reviewers for their thoughtful feedback. Below we address each weakness.
>
> **1a UFM Assumptions**
>
> We agree that conclusions from the UFM must be interpreted carefully. App. A justifies the UFM as a modelling tool. We can expand this discussion if helpful.
>
> **1b MSE vs CE**
>
> We used MSE for tractability, as in prior UFM work. However, the Section 3 results extend to CE: most calculations already appear in App. F or follow by the same MSE argument. We will add these CE results to App. B.
>
> **1c UFM regularization**
>
> UFM regularization is introduced for tractability, but prior work has found identical structures under both standard and UFM regularization (Sukenik et al., 2024), suggesting this choice does not alter the conclusions.
>
> Regarding λ, Lemma 3.1 shows a threshold: above it the zero solution is optimal, below it DNC is optimal and our theory applies. Thus below threshold, the exact value does not affect the conclusions.
>
> **2 Analyzing Hessian at convergence**
>
> Both Hessian spikes and DNC emerge before convergence, both empirically and in the model (Rangamani et al., Dang et al.). We therefore view this as compatible with our conclusions. We focus on the converged regime as it is the setting treated in most prior works.
>
> **3 Prior UFM work**
>
> We will expand Section 2 to summarize prior UFM results and replace “DNC is optimal” with the precise claim from Lemma C.1.
>
> **4a MSE outlier count**
>
> All outliers are equal and the bulk is at 0 because we are analyzing the overparameterized limit; in agreement with experiments showing reduced spectral noise as capacity grows (Sagun et al. 2017). The discrepancy between $K^2$ and $K$ is addressed in App. B.
>
> **4b DNC not globally optimal for CE**
>
> For CE, DNC is not globally optimal. However, our focus is on ubiquity rather than optimality: Garrod et al. show DNC arises frequently under standard hyperparameter settings, much like in real networks. We analyze the spectral phenomena associated with this commonly observed regime. We will add this context.
>
> **4c CE minibulk mismatch with prior empirical work**
>
> We see $(K-1)^2 - K$ rather than $K^2 - K$ due to global centering. In the CE linear UFM, the features are globally centered, giving rank K−1. This centering is incompatible with ReLU, since globally centered matrices contain negative entries. As a result, the ReLU case retains one additional global-direction spike, which changes the count from K−1 to K. This mechanism appears in Dang et al. (2024), where the extra spike makes the features a rank K orthogonal frame. We will add experiments and include the CE analogues of Thms 4.2, 4.3 in the appendix, the proofs are nearly identical to the MSE case once Dang’s tools are used.
>
> **5.1 Experiments relation to modern networks**
>
> The purpose of our experiments is to test the UFM predictions in real datasets, while staying close to the theory so that the comparison is without conflating factors. This is standard in mathematical analyses of this kind (see references in App. A). More broadly, When we say the theory explains phenomena observed in real networks, we are not just talking about our own experiments but also earlier empirical results. We agree that our experiments are supporting evidence rather than comprehensive validation. A full empirical study is beyond the scope of this work.
>
> **5.2 ReLU experiments**
>
> In revision, we will add ResNet + ReLU experiments. Preliminary results show behavior matching the ReLU UFM, with one larger feature spike enforcing non-negativity and a similar Hessian spectrum. We aim to provide preliminary plots in a follow-up response. Evidence that the UFM captures such phenomena can also be found in Sukenik et al. (2024), where the ReLU UFM and ResNet+ReLU have exact agreement.
>
> **5.3 Use of MSE and UFM-style regularization in the experiments**
>
> In the revision we will state that experiments use MSE to stay close to theory.
>
> Different regularization strengths on features and weights has no effect on the theory. Dang et al. (2023) characterizes the global minimizers for arbitrary regularization parameters: they only rescale the layer norms, leaving eigenvectors and eigenvalue structure unchanged. We will add this point to App. G.
>
> Our real-network experiments use UFM-style regularization. We are happy to add experiments with standard regularization, based on intuition of Sukenik et al. (2024) we expect this to agree with the theory.
>
> **5.4 l=3 experiments**
>
> The same patterns appear across all layers. We will add results for other layers to the revision.
>
> **Corrections**
>
> We will implement all of your suggestions.
>
> **Questions**
>
> 1. The model experiments converge to theory; we show only early epochs for readability. We will add final convergence values to the figure captions.
>
> **Limitations**
>
> We will add a limitations section for the essential assumptions that you mention.
>
> We appreciate the reviewer’s insightful feedback, which will substantially improve the paper.

---

> > ### Author Rebuttal · Reviewer_FTM4 · 2026-04-01
> >
> > Thank you for the response. Some of my concerns have been addressed, but a few points still remain, especially for the experimental setting.
> >
> > I acknowledge that the experiments in the paper are intended to test whether the UFM-based theoretical predictions carry over to real datasets. However, my concern is that, if the theory does not actually reflect what happens in real neural network training, then it becomes unclear what significance these theoretical results have. I understand that the theory is developed within the UFM architecture for the sake of mathematical tractability. Still, for such a simplification to be convincing, the resulting predictions should at least be observable in, or supported by, experiments on actual neural networks. If the phenomenon is observed only within the UFM architecture but not in real neural network training, then it is difficult for readers to accept that this simplification meaningfully captures practical training behavior.
> >
> > Moreover, explanations such as prior work only evaluating UFM or claims that UFM exhibits similar trends to real architectures are not sufficient to address this concern. The paper should be self-contained, and therefore it needs to directly demonstrate or justify within its own scope why the theoretical insights derived from UFM are relevant to real neural network training.
> >
> > Therefore, I strongly encourage the authors to include experiments on real neural networks. If the empirical results do not align with the theory, the paper should at least make this discrepancy explicit and provide some justification for why the theory remains informative for practical networks despite that mismatch.

---

> > > ### Author Response · Authors · 2026-04-06
> > >
> > > We thank the reviewer for the additional clarification and for articulating this concern so clearly. Upon reflection, we agree that further empirical validation would strengthen the manuscript, both by making it more self-contained and by providing additional support for the modelling assumptions.
> > >
> > > To address this point, in the revision we will expand the experiments currently presented in the ResNet+Linear setting in Section 5 and Appendix G to include the following architectures:
> > >
> > > * ResNet+ReLU with UFM style regularization.
> > > * Full ResNet with standard regularization.
> > >
> > > In addition, we provide preliminary results for a full ResNet-20 with a 4-layer ReLU head with residual connections, trained with standard regularization of strength $5 \times 10^{-5}$. The statistics below correspond to the first layer of the connected head. We train for 8,000 epochs; otherwise, the setup matches the real-data experiments in the main text. All quantities and plots reported below are measured at the end of training.
> > >
> > > * **Predicted Eigenvector Alignment**: The average alignment: $0.943$, with standard deviation $0.02$. Since perfect agreement corresponds to 1, this indicates that the predicted eigenvectors align very closely with those observed in training.
> > > * **Gradient Alignment $c=c’$**: The average alignment is $0.091$, with standard deviation $0.02$. Since the predicted value under perfect agreement is 0.01, this still indicates good qualitative agreement.
> > > * **Gradient Alignment $c \neq c’$**: The average alignment is $2.2 \times 10^{-4}$, with standard deviation $4.49 \times 10^{-4}$. The predicted value under perfect agreement is 0, again showing close agreement.
> > > * **Weight Eigenvalues**: we provide a plot of the top 10 spikes plus the bulk spectrum here: https://ibb.co/6cVwPX4g . The plot shows the correct number of spikes.
> > > * **Hessian Eigenvalues**: we provide a plot of the top 100 spikes plus the bulk spectrum here: https://ibb.co/CsQQFzgN . Here as well, we observe exactly the predicted number of spikes. For additional context, the average outlier eigenvalue is $5.5 \times 10^{-2}$, with standard deviation $9 \times 10^{-3}$.
> > > * **Papyan Decomposition**: We include plots of the top 10 spikes of $Hess_l - G_{cross}$ here: https://ibb.co/9ktrcpdV , and of the top 90 spikes of $Hess_l - G_{class}$ here: https://ibb.co/NvsVbn6 . As predicted, each subtraction removes exactly the expected number of outlier eigenvalues.
> > >
> > > Taken together, these results already suggest that the model provides an accurate account of the behaviour observed in real-network experiments, with some discrepancy naturally expected since the modelling assumptions do not hold exactly in practice. In the revised manuscript, we will include the full training trajectories for these quantities rather than only their final values.
> > >
> > > More broadly, while we will continue to cite prior work reporting similar phenomena, we will revise the manuscript so that those references serve only as supporting context, rather than as the primary basis for empirical validation.
> > >
> > > We thank the reviewer again for this constructive feedback, which we believe will substantially improve the manuscript.

---

### Official Review · Reviewer_HVGW · 2026-03-22

**Soundness:** 3
**Presentation:** 2
**Significance:** 2
**Originality:** 2
**Overall Recommendation:** 3
**Confidence:** 3

**Summary:**

This paper presents a formal mathematical connection between Deep Neural Collapse and the empirical low-rank Hessian and gradient spectrum phenomena of Papyan 2020 a\&b, using the Deep Unconstrained Feature model. The authors give several analytical extensions to the literature:

* closed-form equations for the spectra of layer-wise Hessians, gradients, and weights under MSE to prove that $K^2$ outliers emerge from DNC geometry
* extending these derivations to the full-network Hessian via an asymptotic double limit ($\\bar{L}, L \\to \\infty$, $\\bar{L}/L \\to 0$) and an equal-scale assumption to decouple cross-layer blocks
* generalizing the linear framework to ReLU networks by at the lowest-loss DNC solution where the ReLU behave linearly
* analytically recovering the $(K-1)^2$ "mini-bulk" eigenvalue spikes under cross entropy loss

**Compliance With Llm Reviewing Policy:**

Affirmed.

**Final Justification:**

I lean towards rejection, because: 1) In rebuttal, the authors acknowledge that the causal connection between NC and spectral structure was already established empirically by Papyan (JMLR 2020), and their contribution is its mathematical formalization, which weakens the significance. 2) The extensions to full-network Hessian and ReLU both reduce to the already-solved linear layer-wise case. 3) In the full-network case, this comes at a large cost to realism. (I do not agree with the authors' final argument that the head-depth-to-infinity assumption is "exactly the same as the UFM assumption.")

**Key Questions For Authors:**

I would appreciate replies to my points 1-3 above, in case my understanding is incorrect or there are other justifications for the assumptions made.

**Limitations:**

The assumptions in the full-network and ReLU cases are limitations that are not well justified in the paper.

**Strengths And Weaknesses:**

**Strengths**

My understanding is that this paper is making a valuable and previously-incomplete formal link between NC and low-rank, bulk-outlier spectra phenomena, using the UFM framework.

* exact, closed-form derivations for the bulk-outlier structure of the spectra in the layer-wise MSE case, including orthogonal decomposition of the class/cross-class Hessian components
* the gradient alignment result (Theorem 3.3) showing that the gradient occupies exactly the $K$-dimensional diagonal subspace of the $K^2$-dimensional eigenspace with uniform $1/K$ coefficients, which to my knowledge, is more fine-grained than prior work
* validation of theoretical predictions (such as the gradient coefficient distributions and rank convergence) on CIFAR-10 and MNIST
* analytically recovering the $(K-1)^2$ "mini-bulk" eigenvalue spikes under cross entropy loss, demonstrating this separation is a consequence of the loss function rather than the UFM assumption

**Weaknesses**

I believe the paper overclaims novelty, under-attributes prior work, and underplays the strong assumptions it makes in the full-network and ReLU extensions.

**1\. Attribution and novelty**

The paper frames its results as “identifying DNC as a unifying mechanism governing curvature, gradient alignment, and weight structure,” stating that the connection between NC and the low-dimensional spectra has been "largely overlooked." Specifically, it states the paper which introduces NC, Papyan et al (2020), only addresses the connection in a single remark. As I understand it, this misrepresents the literature: while the earlier Papyan (JMLR 2020\) didn’t yet use the term “Neural Collapse”, that paper essentially laid out the same causal connection between NC and low-dim spectra of Hessian, gradients, and weights. For example, see Papyan (JMLR 2020\) Section 1.12 and Figure 2\.  My impression is that this paper’s contribution is more about formalizing this causal relationship mathematically in the context of the UFM.

A smaller attribution issue is that the related work refers to works like Dang et al (2023) as only "tangentially related,” when their mathematical foundation is central to this paper, e.g. the use of Lemma C.1.

**2\. Unjustified strong assumptions in the full-network case (Theorem 3.6)**

To calculate the full network Hessian, the authors rely on the asymptotic double limit $\\bar{L}, L \\to \\infty$ such that $\\bar{L}/L \\to 0$. In other words, the explicitly modeled head ($L$) is infinitely deeper than the un-modeled feature-extractor backbone ($\\bar{L}$), which seems opposite to real-world networks. Furthermore, they assume the un-modeled $\\bar{L}$ backbone Hessian blocks share the exact same scale ($\\nu$) as the $L$ head. Together, these let them approximate the full Hessian by the head Hessian, trivially reducing it to the modeled head Hessian, obtained from the layer-wise Hessians. I recommend demoting Theorem 3.6, maybe to the appendix, discussing the assumptions explicitly, and softening the claim of the full Hessian as a major contribution.

**3\. Unrealistic assumptions in the ReLU case (Section 4\)**

The authors give some extension of their analysis to ReLU networks. However, Theorem 4.2 analyzes a minimum-loss DNC solution, where all pre-activations are strictly non-negative ($H\_l \= \\sigma(H\_l)$), reducing it to the linear case. This directly contradicts the empirical reality of ReLU networks, which demonstrate high sparsity. They do show (both empirically and theoretically) that among the DNC solutions of the UFM (which are known not to be global optima), the minimum-loss solutions do have this linear behavior. However, I take this more to mean that UFM does not realistically model DNC in the ReLU case, rather than as justification for the claim of a full-network result. I believe the fundamental mismatch results from the direct regularization of the features in UFM.

**Minor**

1. Several works on “gradient alignment with outlier eigenvectors” are cited at the end of the related work section without any contextualization relating them to the current work.
2. I would prefer to see more rigorous handling of the asymptotics in the proofs of Theorems 3.5 and B.2.
3. Typos: Appendix B: "Theorems 2-5" \-\> Theorems 3.2-3.5. Capitalization on lines 136 and 1010\.

---

> ### Author Rebuttal · Authors · 2026-03-31
>
> We thank the reviewer for the constructive feedback. Below we address each concern.
>
> **1a: Attribution/novelty**
>
> We did not intend to suggest that the empirical connection between NC and Hessian spectra was absent from prior work. In particular, Papyan was aware of these concepts empirically, and the connection is implicit in his previous work. Our contribution is to provide a new mathematical formalization of this relationship, and we will revise the text to make this distinction clearer.
>
> **1b: Small issue**
>
> We agree that the wording was imprecise. Our work is mathematically close to Dang et al. because both study the same model. We will rephrase this discussion to better situate our contribution within the UFM literature.
>
> **2a: Strong assumptions in Theorem 3.6**
>
> We appreciate the opportunity to clarify this assumption. Our motivation comes from the fact that, in practice, the division between “feature map” and “head” is somewhat arbitrary: although we assume a fully connected head for analytical tractability, in real networks one could place this separation at any layer.
>
> To apply the UFM framework to a real network though, we need that the feature map is sufficiently deep to produce representations within some small threshold ϵ of collapse. If this occurs at some layer $\bar{L}$, we expect later layers to remain within the same threshold, consistent with the empirical monotonic increase of DNC with depth and as justified by related UFM theory of Dang et al.
>
> From this viewpoint, if a network reaches this threshold by layer $\bar{L}$, then adding further layers should be viewed as increasing the number of subsequent layers below the same ϵ DNC threshold. This is why the model takes the limit by adding layers after the feature map, but it is our view that for real networks this is really just the same as adding layers generally, due to the arbitraryness of the separation.
>
> The $\bar{L}→∞$ limit is a device that makes the UFM assumption exact and enables a clean analysis. For finite networks, the interpretation only requires approximate collapse up to a threshold, not an exact infinite-depth limit.
>
> This remains an idealization of real training due to the modelling, which we discuss in App. A, but we do not view the Theorem 3.6 assumption as substantially stronger than the assumptions preceding it. We will add a discussion of these points to Section 3.5.
>
> **2b: same scale assumption in backbone and head blocks**
>
> We stated the assumption as exact equality of scale for simplicity, but the argument only requires the backbone and head blocks to be of the same order, so that earlier layers do not diverge relative to later ones. This can be proved for homogeneous networks with regularization. We will clarify this point in the paper and include numerical evidence from real networks.
>
> **2c: trivial reduction to the head Hessian**
>
> We agree that these steps reduce the full Hessian to the head Hessian. This reflects a structural feature of deep networks in our regime: when most later layers satisfy DNC, most layerwise Hessian blocks inherit the structure we describe, and the full Hessian does as well. We will revise the exposition to make this clearer.
>
> **3: Unrealistic assumptions in the ReLU case**
>
> We agree that a standard ReLU layer does not generally act linearly on its input, and we stress that this is not our claim. Rather, the point concerns the behavior of later ReLU layers in deep networks. In the expressive limit, these later layers play less of a role in transforming the data and more in propagating representations learned by earlier layers.
>
> When ReLU acts nonlinearly, it decreases the Frobenius norm of the representation. In homogeneous models with regularization, preserving the same representation after this decrease requires increasing parameter norms, which raises the regularization cost. Thus, when the network is incentivized to propagate a representation, it is favorable for that representation to already be nonnegative. This is also compatible with sparsity: many coordinates are zero, but this is already achieved by the feature map and remains unchanged under ReLU.
>
> This behavior is not unique to DNC or caused by the UFM. Sukenik et al. (2024) observe this effect in the ReLU UFM across a range of structures. They also observe the same effect in real networks under standard regularization. We have observed the same behavior in preliminary experiments with a ResNet + ReLU setup and will provide plots in a follow-up comment.
>
> **Minor**
> * We will add the context relating them to Section 3.3.
> * We will revise the proofs to include the relevant subleading asymptotic terms.
> * We will correct the typos.
>
> **Limitations**
>
> Thank you for highlighting this issue. We will add a limitations section that discusses these points and clarifies the scope and assumptions of our analysis.
>
> Thank you again for your valuable feedback that will improve our paper. If you have any further questions let us know.

---

> > ### Author Rebuttal · Reviewer_HVGW · 2026-04-04
> >
> > Thank you for your response.
> >
> > 1. I appreciate the clarification. Making this distinction clearer will be beneficial and address my concern about over-claimed novelty.
> >
> > 2. I still feel that the full-network-Hessian result, by taking $\bar{L}/L \to 0$, is essentially assuming what it wants to prove: it assumes the post-collapse portion of the network dominates, in which case it makes sense that the full Hessian reduces to the head Hessian. But my impression would be that, in real networks, DNC emerges progressively with depth. It could well be the case that achieving $\epsilon$-collapse always occurs at a fixed proportion of total depth, rather than at a vanishingly small one. In that case, $\bar{L}/L$  would be a constant. Therefore, I still feel that this the result should be de-emphasized.
> >
> > 3. I appreciate the authors' clarifying remarks. It is believable that ReLU behaves this way empirically in a multi-layer head after a ResNet backbone. The paper's framing still feels a bit misleading, however. The abstract says "our results hold for both linear and ReLU networks," which is technically true, but only because the ReLU is inactive in this setting. Perhaps Theorem 4.3, proving this, actually deserves more weight than Theorem 4.2.
> >
> > Overall, while the authors have clarified some aspects, I find that significance remains limited by the factors mentioned. In rebuttal, the authors acknowledge that the causal connection between NC and spectral structure was already established empirically by Papyan (JMLR 2020), and their contribution is its mathematical formalization. Many of the core results follow a systematic pattern — substituting known DNC properties into the Kronecker structure of the Hessian — that appears carefully executed but technically straightforward. It is true that the CE extension and the proof of Theorem 4.3 involve more substantial arguments. However, the extensions to full-network Hessian and ReLU both reduce to the already-solved linear layer-wise case, and in the full-network case this comes at the cost of realism, as discussed above. I maintain my score.

---

> > > ### Author Response · Authors · 2026-04-06
> > >
> > > Thank you for your additional clarification and for engaging further with our work. We address your remaining concerns below.
> > >
> > > **2 Full Hessian Results**
> > >
> > > You are correct that DNC emerges progressively with depth, and we believe this is fully consistent with our description. We clarify the reasoning below.
> > >
> > > * Our target $Y$ is fully NC collapsed. Therefore, in the expressivity limit, any globally optimal network must also be fully collapsed at the output layer in order to fit $Y$ exactly. The exception would be if the regularization were too strong, but our theorem assumes regularization decays linearly with depth; this scaling is necessary to avoid the trivial zero solution and is therefore a natural assumption in this setting.
> > > * Given this, achieving maximal collapse at the output requires each layer to contribute as much collapse as possible. In particular, the amount of collapse attainable by a fixed layer $\bar{L}$ at optimum should not depend on how many additional layers are appended after it.
> > > * Thus, if collapse to threshold $\epsilon$ is achieved by layer $\bar{L}$, and we denote by $L$ the number of layers following $\bar{L}$, then increasing the total network depth increases $L$, not $\bar{L}$.
> > > * It follows that the fraction of layers collapsed to threshold $\epsilon$ is $\frac{L}{L + \bar{L}} \to 1$ as depth increases. Therefore, maintaining $\epsilon$-collapse at a fixed proportion of the network as depth increases is incompatible with loss minimization.
> > >
> > > We view this assumption as exactly the same as the UFM assumption. The UFM assumption states that the feature map is sufficiently expressive to map the data to a wide range of feature vectors, so that it can be treated as effectively unconstrained. This is a property determined by the number of layers $\bar{L}$ of the feature map, and is unaffected by layers appended afterward. Consequently, adding more layers does not alter whether the original $\bar{L}$-layer feature map satisfies the UFM condition; it only amounts to appending additional layers after that map. As a result, whether this abstraction captures real networks depends on the validity of the UFM assumption alone. Of course, the UFM assumption itself may be questioned, which is precisely why Appendix A is devoted to highlighting the extensive empirical evidence of prior works.
> > >
> > > We also emphasize that the full UFM result is not a trivial consequence of the UFM approximation. It relies essentially on the head-Hessian result in Theorem 3.5, whose proof requires a nontrivial argument.
> > >
> > > **3 ReLU Results**
> > >
> > > We agree that this behavior arises because ReLU acts linearly in the model. However, as you note, Theorem 4.3 shows that this regime is globally optimal among DNC configurations. Consequently, this network simplicity arises due to the optimization landscape itself, not through any extra simplifying assumptions in the proof of the theoretical result.
> > >
> > > We also agree that the manuscript should be reordered to make this point clearer. In the revision, we will present Theorem 4.3 first, since it is the key theoretical step underlying Theorem 4.2, and then state Theorem 4.2 explicitly as a corollary of Theorem 4.3.
> > >
> > > To further support this point empirically, we trained a network with ResNet backbone and depth-4 ReLU head on MNIST. At convergence the proportion of feature matrix entries out of the 3rd layer, pre-ReLU  below $-10^{-3}$ was 8%, with none below $-10^{-2}$, whereas 40% of entries were above $10^{-3}$, and 20% above $10^{-2}$, so the network does reduce to approximately behaving linearly, even on real network data. We will be including such evaluations in the revision.
> > >
> > > Thank you again for your constructive feedback and for the time you invested in evaluating our manuscript.

---

### Decision · Program_Chairs · 2026-04-30

**Decision:**

Accept (regular)

**Comment:**

This paper theoretically explains why low-dimensional (bulk-outlier) structures appear in the spectra of key matrices in deep learning, including Hessian, gradient, and weight matrices. The main claim is that deep neural collapse (DNC) is the fundamental reason behind these low-dimensional spectra. The paper analyses a simplified unconstrained feature model and shows that the eigenvalues and eigenvectors of these key matrices can be constructed explicitly in terms of the class means. They also provide empirical validation for linear and ReLU networks, showing the emergence of low-dimensional spectra and feature-collapse behavior.

This paper achieves an average score of 4.

The main reasons are as follows:
- Closed-form spectral characterization and fine-grained gradient alignment analysis.
- Clear motivation and good writing/organization.

The main critiques are as follows:
- The paper may have overclaimed novelty. For example, some results have been established empirically by Papyan 2020.
- Some assumptions are strong and unrealistic.
- ReLU analysis reduces to linear case (inactive ReLU) and is not realistic.
- Mostly a mathematical formalization of known phenomena and may have limited significance.

Some of these points remain unresolved, which makes one reviewer still lean towards rejection. From my own perspective, I feel the community can benefit from having a formal treatment of the connection between low-dimensional spectra in Hessian and weights from the NC perspective. I thus recommend weak acceptance.